# WHAT HAPPENS AFTER SGD REACHES ZERO LOSS? –A MATHEMATICAL FRAMEWORK

**Zhiyuan Li**
Department of Computer Science
Princeton University
`zhiyuanli@cs.princeton.edu`

**Tianhao Wang**
Department of Statistics and Data Science
Yale University
`tianhao.wang@yale.edu`

**Sanjeev Arora**
Department of Computer Science
Princeton University
`arora@cs.princeton.edu`

## ABSTRACT

Understanding the implicit bias of Stochastic Gradient Descent (SGD) is one of the key challenges in deep learning, especially for overparametrized models, where the local minimizers of the loss function $L$ can form a manifold. Intuitively, with a sufficiently small learning rate $\eta$, SGD tracks Gradient Descent (GD) until it gets close to such manifold, where the gradient noise prevents further convergence. In such regime, Blanc et al. (2020) proved that SGD with label noise locally decreases a regularizer-like term, the sharpness of loss, $\text{tr}[\nabla^2 L]$. The current paper gives a general framework for such analysis by adapting ideas from Katzenberger (1991). It allows in principle a complete characterization for the regularization effect of SGD around such manifold—i.e., the "implicit bias"—using a stochastic differential equation (SDE) describing the limiting dynamics of the parameters, which is determined jointly by the loss function and the noise covariance. This yields some new results: (1) a *global* analysis of the implicit bias valid for $\eta^{-2}$ steps, in contrast to the local analysis of Blanc et al. (2020) that is only valid for $\eta^{-1.6}$ steps and (2) allowing *arbitrary* noise covariance. As an application, we show with arbitrary large initialization, label noise SGD can always escape the kernel regime and only requires $O(\kappa \ln d)$ samples for learning an $\kappa$-sparse overparametrized linear model in $\mathbb{R}^d$ (Woodworth et al., 2020), while GD initialized in the kernel regime requires $\Omega(d)$ samples. This upper bound is minimax optimal and improves the previous $\widetilde{O}(\kappa^2)$ upper bound (HaoChen et al., 2020).

## 1 INTRODUCTION

The implicit bias underlies the generalization ability of machine learning models trained by stochastic gradient descent (SGD). But it still remains a mystery to mathematically characterize such bias. We study SGD in the following formulation

$$x_\eta(k+1) = x_\eta(k) - \eta(\nabla L(x_\eta(k)) + \sqrt{\Xi} \cdot \sigma_{\xi_k}(x_\eta(k))) \tag{1}$$

where $\eta$ is the learning rate (LR), $L : \mathbb{R}^D \to \mathbb{R}$ is the training loss and $\sigma(x) = [\sigma_1(x), \sigma_2(x), \ldots, \sigma_\Xi(x)] \in \mathbb{R}^{D \times \Xi}$ is a deterministic noise function. Here $\xi_k$ is sampled uniformly from $\{1, 2, \ldots, \Xi\}$ and it satisfies $\mathbb{E}_{\xi_k}[\sigma_{\xi_k}(x)] = 0, \forall x \in \mathbb{R}^d$ and $k$.

It is widely believed that large LR (or equivalently, small batch size) helps SGD find better minima. For instance, some previous works argued that large noise enables SGD to select a flatter attraction basin of the loss landscape which potentially benefits generalization (Li et al., 2019c; Jastrzebski et al., 2017). However, there is also experimental evidence (Li et al., 2020b) that small LR also has equally good implicit bias (albeit with higher training time), and that is the case studied here. Presumably low LR precludes SGD jumping between different basins since under general conditions this should require $\Omega(\exp(1/\eta))$ steps (Shi et al., 2020). In other words, there should be a mechanism to reach better generalization while staying within a single basin. For deterministic GD similar mechanisms

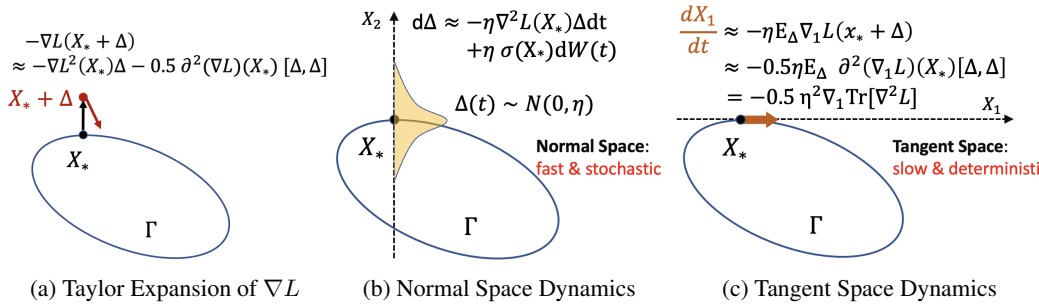

Figure 1: Illustration for limiting flow in $\mathbb{R}^2$. $\Gamma$ is an 1D manifold of minimizers of loss $L$.

have been demonstrated in simple cases (Soudry et al., 2018; Lyu & Li, 2019) and referred to as *implicit bias of gradient descent*. The current paper can be seen as study of implicit bias of Stochastic GD, which turns out to be quite different, mathematically.

Recent work (Blanc et al., 2020) shed light on this direction by analyzing effects of stochasticity in the gradient. For sufficiently small LR, SGD will reach and be trapped around some manifold of local minimizers, denoted by $\Gamma$ (see Figure 2). The effect is shown to be an implicit deterministic drift in a direction corresponding to lowering a regularizer-like term along the manifold. They showed SGD with label noise locally decreases the sharpness of loss, $\text{tr}[\nabla^2 L]$, by $\Theta(\eta^{0.4})$ in $\eta^{-1.6}$ steps. However, such an analysis is actually *local*, since the natural time scale of analysis should be $\eta^{-2}$, not $\eta^{-1.6}$.

The contribution of the current paper is a more general and global analysis of this type. We introduce a more powerful framework inspired by the classic paper (Katzenberger, 1991).

## 1.1 INTUITIVE EXPLANATION OF REGULARIZATION EFFECT DUE TO SGD

We start with an intuitive description of the implicit regularization effect described in Blanc et al. (2020). For simplification, we show it for the canonical SDE approximation (See Section B.1 for more details) of SGD (1) (Li et al., 2017; Cheng et al., 2020). Here $W(t)$ is the standard $\Xi$-dimensional Brownian motion. The only property about label noise SGD we will use is that the noise covariance $\sigma\sigma^\top(x) = \nabla^2 L(x)$ for every $x$ in the manifold $\Gamma$ (See derivation in Section 5).

$$\mathrm{d}\tilde{X}_\eta(t) = -\eta\nabla L(\tilde{X}_\eta(t))\mathrm{d}t + \eta \cdot \sigma(\tilde{X}_\eta(t))\mathrm{d}W(t). \tag{2}$$

Suppose $\tilde{X}_\eta(0)$ is already close to some local minimizer point $X_* \in \Gamma$. The goal is to show $\tilde{X}_\eta(t)$ will move in the tangent space and steadily decrease $\text{tr}[\nabla^2 L]$. At first glance, this seems impossible as the gradient $\nabla L$ vanishes around $\Gamma$, and the noise has zero mean, implying SGD should be like random walk instead of a deterministic drift. The key observation of Blanc et al. (2020) is that the local dynamics of $\tilde{X}_\eta(t)$ is completely different in tangent space and normal space — the fast random walk in normal space causes $\tilde{X}_\eta(t)$ to move slowly (with velocity $\Theta(\eta^2)$) but deterministically in certain direction. To explain this, letting $\Delta(t) = \tilde{X}_\eta(t) - X_*$, Taylor expansion of (2) gives $\mathrm{d}\Delta(t) \approx -\eta\nabla^2 L(X_*)\Delta\mathrm{d}t + \eta\sigma(X_*)\mathrm{d}W(t)$, meaning $\Delta$ is behaving like an Ornstein-Uhlenbeck (OU) process locally in the normal space. Its mixing time is $\Theta(\eta^{-1})$ and the stationary distribution is the standard multivariate gaussian in the normal space scaled by $\sqrt{\eta}$ (see Figure 1b), because noise covariance $\sigma\sigma^\top = \nabla^2 L$. Though this OU process itself doesn't form any regularization, it activates the second order Taylor expansion of $\nabla L(X_* + \Delta(t))$, i.e., $-\frac{1}{2}\partial^2(\nabla L)(X_*)[\Delta(t), \Delta(t)]$, creating a $\Theta(\eta^2)$ velocity in the tangent space. Since there is no push back force in the tangent space, the small velocity accumulates over time, and in a longer time scale of $\Omega(\eta^{-1})$, the time average of the stochastic velocity is roughly the same as the expected velocity when $\Delta$ is sampled from its stationary distribution. This simplifies the expression of the velocity in tangent space to $\frac{\eta^2}{2}\nabla_T \text{tr}[\nabla^2 L]$ (see Figure 1c), where $\nabla_T$ means the gradient is only taken in the tangent space.

However, the above approach only gives a local analysis for $O(\eta^{-1.6})$ time, where the total movement due to implicit regularization is $O(\eta^{2-1.6}) = O(\eta^{0.4})$ and thus is negligible when $\eta \to 0$. In order to get a non-trivial limiting dynamics when $\eta \to 0$, a global analysis for $\Omega(\eta^{-2})$ steps is necessary and it cannot be done by Taylor expansion with a single reference point. Recent work by Damian et al. (2021) glues analyses of multiple local phases into a global guarantee that SGD finds a $(\epsilon, \gamma)$-stationary point for the regularized loss, but still doesn't show convergence for trajectory when $\eta \to 0$ and cannot deal with general noise types, *e.g.*, noise lying in the tangent space of the manifold. The

main technical difficulty here is that it's not clear how to separate the slow and fast dynamics in different spaces and how to only take limit for the slow dynamics, especially when shifting to a new reference point in the Taylor series calculation.

## 1.2 Our Approach: Separating the Slow from the Fast

In this work, we tackle this problem via a different angle. First, since the anticipated limiting dynamics is of speed $\Theta(\eta^2)$, we change the time scaling to accelerate (2) by $\eta^{-2}$ times, which yields

$$\mathrm{d}X_\eta(t) = -\eta^{-1}\nabla L(X_\eta(t))\mathrm{d}t + \sigma(X_\eta(t))\mathrm{d}W(t). \tag{3}$$

The key idea here is that we only need to track the slow dynamic, or equivalently, some *projection* of $X$ onto the manifold $\Gamma$, $\Phi(X)$. Here $\Phi : \mathbb{R}^D \to \Gamma$ is some function to be specified and hopefully we can simplify the dynamics (3) via choosing suitable $\Phi$. To track the dynamics of $\Phi(X_\eta)$, we apply Ito's lemma (a.k.a. stochastic chain rule, see Lemma A.9) to Equation (3), which yields

$$\mathrm{d}\Phi(X_\eta(t)) = -\eta^{-1}\partial\Phi(X_\eta(t))\nabla L(X_\eta(t))\mathrm{d}t + \partial\Phi(X_\eta(t))\sigma(X_\eta(t))\mathrm{d}W(t)$$
$$+ \frac{1}{2}\sum_{i,j=1}^{D} \partial_{ij}\Phi(X_\eta(t))(\sigma(X_\eta(t))\sigma(X_\eta(t))^\top)_{ij}\mathrm{d}t.$$

Note the first term $-\eta^{-1}\partial\Phi(X_\eta)\nabla L(X_\eta)$ is going to diverge to $\infty$ when $\eta \to 0$, so a natural choice for $\Phi$ is to kill the first term. Further note $-\partial\Phi(X)\nabla L(X)$ is indeed the directional derivative of $\Phi$ at $X$ towards $-\nabla L$, killing the first term becomes equivalent to making $\Phi$ invariant under Gradient Flow (GF) of $-\nabla L(X)$! Thus it suffices to take $\Phi(X)$ to be the limit of GF starting at $X$. (Formally defined in Section 3; see Lemma C.2 for a proof of $\partial\Phi(X)\nabla L(X) \equiv 0$.)

Also intuitively $X_\eta$ will be infinitely close to $\Gamma$, i.e., $d(X_\eta(t), \Gamma) \to 0$ for any $t > 0$ as $\eta \to 0$, so we have $\Phi(X_\eta) \approx X_\eta$. Thus we can rewrite the above equation as

$$\mathrm{d}X_\eta(t) \approx \partial\Phi(X_\eta(t))\sigma(X_\eta(t))\mathrm{d}W(t) + \frac{1}{2}\sum_{i,j=1}^{D} \partial_{ij}\Phi(X_\eta(t))(\sigma(X_\eta(t))\sigma(X_\eta(t))^\top)_{ij}\mathrm{d}t, \tag{4}$$

and the solution of (4) shall converge to that of the following (in an intuitive sense):

$$\mathrm{d}X(t) = \partial\Phi(X(t))\sigma(X(t))\mathrm{d}W(t) + \frac{1}{2}\sum_{i,j=1}^{D} \partial_{ij}\Phi(X(t))(\sigma(X(t))\sigma(X(t))^\top)_{ij}\mathrm{d}t, \tag{5}$$

The above argument for SDE was first formalized and rigorously proved by Katzenberger (1991). It included an extension of the analysis to the case of asymptotic continuous dynamics (Theorem 4.1) including SGD with infinitesimal LR, but the result is weaker in this case and no convergence is shown. Another obstacle for applying this analysis is that 2nd order partial derivatives of $\Phi$ are unknown. We solve these issues in Section 4 and our main result Theorem 4.6 gives a clean and complete characterization for the implicit bias of SGD with infinitesimal LR in $\Theta(\eta^{-2})$ steps. Finally, our Corollary 5.2 shows (5) gives exactly the same regularization as $\mathrm{tr}[\nabla^2 L]$ for label noise SGD.

The **main contributions** of this paper are summarized as follows.

1. In Section 4, we propose a mathematical framework to study the implicit bias of SGD with infinitesimal LR. Our main theorem (Theorem 4.6) gives the limiting diffusion of SGD with LR $\eta$ for $\Theta(\eta^{-2})$ steps as $\eta \to 0$ and allows any covariance structure.
2. In Section 5, we give limiting dynamics of SGD with isotropic noise and label noise.
3. In Section 6, we show for any initialization, SGD with label noise achieves $O(\kappa \ln d)$ sample complexity for learning a $\kappa$-sparse overparametrized linear model (Woodworth et al., 2020). In this case, the implicit regularizer is a data-dependent weighted $\ell_1$ regularizer, meaning noise can help reduce the norm and even escape the kernel regime. The $O(\kappa \ln d)$ rate is minimax optimal (Raskutti et al., 2012) and improves over $\tilde{O}(\kappa^2)$ upper bound by HaoChen et al. (2020). In contrast, vanilla GD requires $\Omega(d)$ samples to generalize in the kernel regime.
   For technical contributions, we rigorously prove the convergence of GF for OLM (Lemma 6.3), unlike many existing implicit bias analyses which have to assume the convergence. We also prove the convergence of limiting flow to the global minimizer of the regularizer (Lemma 6.5) by a trajectory analysis via our framework. It cannot be proved by previous results (Blanc et al., 2020; Damian et al., 2021), as they only assert convergence to stationary point in the best case.

## 2    RELATED WORKS

**Loss Landscape of Overparametrized Models**  A phenomenon known as mode connectivity has been observed that local minimizers of the loss function of a neural network are connected by simple paths (Freeman & Bruna, 2016; Garipov et al., 2018; Draxler et al., 2018), especially for overparametrized models (Venturi et al., 2018; Liang et al., 2018; Nguyen et al., 2018; Nguyen, 2019). Later this phenomanon is explained under generic assumptions by Kuditipudi et al. (2019). Moreover, it has been proved that the local minimizers of an overparametrized network form a low-dimensional manifold (Cooper, 2018; 2020) which possibly has many components. Fehrman et al. (2020) proved the convergence rate of SGD to the manifold of local minimizers starting in a small neighborhood.

**Implicit Bias in Overparametrized Models**  Algorithmic regularization has received great attention in the community (Arora et al., 2018; 2019a; Gunasekar et al., 2018b;a;b; Soudry et al., 2018; Li et al., 2018; 2020a). In particular, the SGD noise is widely believed to be a promising candidate for explaining the generalization ability of modern neural networks (LeCun et al., 2012; Keskar et al., 2016; Hoffer et al., 2017; Zhu et al., 2018; Li et al., 2019a). Beyond the size of noise (Li et al., 2019c; Jastrzebski et al., 2017), the shape and class of the noise also play an important role (Wen et al., 2019; Wu et al., 2020). It is shown by HaoChen et al. (2020) that parameter-dependent noise will bias SGD towards a low-complexity local minimizer. Similar implicit bias has also been studied for overparametrized nonlinear statistical models by Fan et al. (2020). Several existing works (Vaskevicius et al., 2019; Woodworth et al., 2020; Zhao et al., 2019) have shown that for the quadratically overparametrized linear model, i.e., $w = u^{\odot 2} - v^{\odot 2}$ or $w = u \odot v$, gradient descent/flow from small initialization implicitly regularizes $\ell_1$ norm and provides better generalization when the groundtruth is sparse. This is in sharp contrast to the kernel regime, where neural networks trained by gradient descent behaves like kernel methods (Daniely, 2017; Jacot et al., 2018; Yang, 2019). This allows one to prove convergence to zero loss solutions in overparametrized settings (Li & Liang, 2018; Du et al., 2018; Allen-Zhu et al., 2019b;a; Du et al., 2019; Zou et al., 2020), where the learnt function minimizes the corresponding RKHS norm (Arora et al., 2019b; Chizat et al., 2018).

**Modelling Stochastic First-Order Methods with Itô SDE**  Apart from the discrete-time analysis, another popular approach to study SGD is through the continuous-time lens using SDE (Li et al., 2017; 2019b; Cheng et al., 2020). Such an approach is often more elegant and can provide fruitful insights like the linear scaling rule (Krizhevsky, 2014; Goyal et al., 2017) and the intrinsic learning rate (Li et al., 2020b). A recent work by Li et al. (2021) justifies such SDE approximation. Xie et al. (2020) gave a heuristic derivation explaining why SGD favors flat minima with SDE approximation. Wojtowytsch (2021) showed that the invariant distribution of the canonical SDE approximation of SGD will collapse to some manifold of minimizers and in particular, favors flat minima. By approximating SGD using a SDE with slightly modified covariance for the overparametrized linear model, Pesme et al. (2021) relates the strength of implicit regularization to training speed.

## 3    NOTATION AND PRELIMINARIES

Given loss $L$, the GF governed by $L$ can be described through a mapping $\phi : \mathbb{R}^D \times [0, \infty) \to \mathbb{R}^D$ satisfying $\phi(x, t) = x - \int_0^t \nabla L(\phi(x, s)) \mathrm{d}s$. We further denote the limiting mapping $\Phi(x) = \lim_{t \to \infty} \phi(x, t)$ whenever the limit exists. We denote $\mathbb{1}_\xi \in \mathbb{R}^\Xi$ as the one-hot vector where $\xi$-th coordinate is 1, and $\mathbb{1}$ the all 1 vector. See Appendix A for a complete clarification of notations.

### 3.1    MANIFOLD OF LOCAL MINIMIZERS

**Assumption 3.1.**  Assume that the loss $L : \mathbb{R}^D \to \mathbb{R}$ is a $\mathcal{C}^3$ function, and that $\Gamma$ is a $(D - M)$-dimensional $\mathcal{C}^2$-submanifold of $\mathbb{R}^D$ for some integer $0 \le M \le D$, where for all $x \in \Gamma$, $x$ is a local minimizer of $L$ and $\mathrm{rank}(\nabla^2 L(x)) = M$.

**Assumption 3.2.**  Assume that $U$ is an open neighborhood of $\Gamma$ satisfying that gradient flow starting in $U$ converges to some point in $\Gamma$, i.e., $\forall x \in U, \Phi(x) \in \Gamma$. (Then $\Phi$ is $\mathcal{C}^2$ on $U$ by Falconer (1983).)

**When does such a manifold exist?**  The vast overparametrization in modern deep learning is a major reason for the set of global minimizers to appear as a Riemannian manifold (possibly with multiple connected components), instead of isolated ones. Suppose all global minimizers interpolate the

training dataset, i.e., $\forall x \in \mathbb{R}^D$, $L(x) = \min_{x' \in \mathbb{R}^D} L(x')$ implies $f_i(x) = y_i$ for all $i \in [n]$, then by preimage theorem (Banyaga & Hurtubise, 2013), the manifold $\Gamma := \{x \in \mathbb{R}^D \mid f_i(x) = y_i, \forall i \in [n]\}$ is of dimension $D - n$ if the Jacobian matrix $[\nabla f_1(x), \ldots, \nabla f_n(x)]$ has rank $n$ for all $x \in \Gamma$. Note this condition is equivalent to that NTK at $x$ has full rank, which is very common in literature.

## 4 LIMITING DIFFUSION OF SGD

In Section 4.1 we first recap the main result of Katzenberger (1991). In Section 4.2 we derive the closed-form expressions of $\partial \Phi$ and $\partial^2 \Phi$. We present our main result in Section 4.3. We remark that sometimes we omit the dependency on $t$ to make things clearer.

### 4.1 RECAP OF KATZENBERGER'S THEOREM

Let $\{A_n\}_{n\geq 1}$ be a sequence of integrators, where each $A_n : \mathbb{R} \to \mathbb{R}$ is a non-decreasing function with $A_n(0) = 0$. Let $\{Z_n\}_{n\geq 1}$ be a sequence of $\mathbb{R}^{|\Xi|}$-valued stochastic processes defined on $\mathbb{R}$. Given loss function $L$ and noise covariance function $\sigma$, we consider the following stochastic process:

$$X_n(t) = X(0) + \int_0^t \sigma(X_n(s) \mathrm{d}Z_n(s) + \int_0^t -\nabla L(X_n(s)) \mathrm{d}A_n(s) \qquad (6)$$

In particular, when the integrator sequence $\{A_n\}_{n\geq 1}$ increases infinitely fast, meaning that $\forall \epsilon > 0, \inf_{t\geq 0}(A_n(t+\epsilon) - A_n(t)) \to \infty$ as $n \to \infty$, we call (6) a *Katzenberger process*.

One difficulty for directly studying the limiting dynamics of $X_n(t)$ is that the point-wise limit as $n \to \infty$ become discontinuous at $t = 0$ if $X(0) \notin \Gamma$. The reason is that clearly $\lim_{n\to\infty} X_n(0) = X(0)$, but for any $t > 0$, since $\{A_n\}_{n\geq 1}$ increases infinitely fast, one can prove $\lim_{n\to\infty} X_n(t) \in \Gamma$! To circumvent this issue, we consider $\overline{Y}_n(t) = X_n(t) - \phi(X(0), A_n(t)) + \Phi(X(0))$. Then for each $n \geq 1$, we have $Y_n(0) = \Phi(X(0))$ and $\lim_{n\to\infty} Y_n(t) = \lim_{n\to\infty} X_n(t)$. Thus $Y_n(t)$ has the same limit on $(0, \infty)$ as $X_n(t)$, but the limit of the former is further continuous at $t = 0$.

**Theorem 4.1** (Informal version of Theorem B.7, Katzenberger 1991). *Suppose the loss $L$, manifold $\Gamma$ and neighborhood $U$ satisfies Assumptions 3.1 and 3.2. Let $\{X_n\}_{n\geq 1}$ be a sequence of Katzenberger process with $\{A_n\}_{n\geq 1}, \{Z_n\}_{n\geq 1}$. Let $Y_n(t) = X_n(t) - \phi(X(0), A_n(t)) + \Phi(X_0)$. Under technical assumptions, it holds that if $(Y_n, Z_n)$ converges to some $(Y, W)$ in distribution, where $\{W(t)\}_{t\geq 0}$ is the standard Brownian motion, then $Y$ stays on $\Gamma$ and admits*

$$Y(t) = Y(0) + \int_0^t \partial \Phi(Y) \sigma(Y) \mathrm{d}W(s) + \frac{1}{2} \sum_{i,j=1}^D \int_0^t \partial_{ij} \Phi(Y)(\sigma(Y)\sigma(Y)^\top)_{ij} \mathrm{d}s. \qquad (7)$$

Indeed, SGD (1) can be rewritten into a Katzenberger process as in the following lemma.

**Lemma 4.2.** *Let $\{\eta_n\}_{n=1}^\infty$ be any positive sequence with $\lim_{n\to\infty} \eta_n = 0$, $A_n(t) = \eta_n \lfloor t/\eta_n^2 \rfloor$, and $Z_n(t) = \eta_n \sum_{k=1}^{\lfloor t/\eta_n^2 \rfloor} \sqrt{\Xi}(\mathbb{1}_{\xi_k} - \frac{1}{\Xi}\mathbb{1})$, where $\xi_1, \xi_2, \ldots \overset{i.i.d.}{\sim} \mathrm{Unif}([\Xi])$. Then with the same initialization $X_n(0) = x_{\eta_n}(0) \equiv X(0)$, $X_n(k\eta_n^2)$ defined by (6) is a Katzenberger process and is equal to $x_{\eta_n}(k)$ defined in (1) with LR equal to $\eta_n$ for all $k \geq 1$. Moreover, the counterpart of (7) is*

$$Y(t) = \Phi(X(0)) + \int_0^t \partial \Phi(Y) \sigma(Y) \mathrm{d}W(s) + \frac{1}{2} \int_0^t \partial^2 \Phi(Y)[\Sigma(Y)] \mathrm{d}s, \qquad (8)$$

*where $\Sigma \equiv \sigma\sigma^\top$ and $\{W(t)\}_{t\geq 0}$ is a $\Xi$-dimensional standard Brownian motion.*

However, there are two obstacles preventing us from directly applying Theorem 4.1 to SGD. First, the stochastic integral in (8) depends on the derivatives of $\Phi$, $\partial \Phi$ and $\partial_{ij}\Phi$, but Katzenberger (1991) did not give their dependency on loss $L$. To resolve this, we explicitly calculate the derivatives of $\Phi$ on $\Gamma$ in terms of the derivatives of $L$ in Section 4.2.

The second difficulty comes from the convergence of $(Y_n, Z_n)$ which we assume as granted for brevity in Theorem 4.1. In fact, the full version of Theorem 4.1 (see Theorem B.7) concerns the stopped version of $Y_n$ with respect to some compact $K \subset U$, i.e., $Y_n^{\mu_n(K)}(t) = Y_n(t \wedge \mu_n(K))$ where $\mu_n(K)$ is the stopping time of $Y_n$ leaving $K$. As noted in Katzenberger (1991), we need the

convergence of $\mu_n(K)$ for $Y_n^{\mu_n(K)}$ to converge, which is a strong condition and difficult to prove in our cases. We circumvent this issue by proving Theorem B.9, a user-friendly interface for the original theorem in Katzenberger (1991), and it only requires the information about the limiting diffusion. Building upon these, we present our final result as Theorem 4.6.

## 4.2 CLOSED-FORM EXPRESSION OF THE LIMITING DIFFUSION

We can calculate the derivatives of $\Phi$ by relating to those of $L$. Here the key observation is the invariance of $\Phi$ along the trajectory of GF.

**Lemma 4.3.** *For any $x \in \Gamma$, $\partial\Phi(x) \in \mathbb{R}^{D \times D}$ is the projection matrix onto tangent space $T_x(\Gamma)$.*

To express the second-order derivatives compactly, we introduce the notion of *Lyapunov operator*.

**Definition 4.4** (Lyapunov Operator). For a symmetric matrix $H$, we define $W_H = \{\Sigma \in \mathbb{R}^{D \times D} \mid \Sigma = \Sigma^\top, HH^\dagger\Sigma = \Sigma = \Sigma HH^\dagger\}$ and *Lyapunov Operator* $\mathcal{L}_H : W_H \to W_H$ as $\mathcal{L}_H(\Sigma) = H^\top\Sigma + \Sigma H$. It's easy to verify $\mathcal{L}_H^{-1}$ is well-defined on $W_H$.

**Lemma 4.5.** *Let $x$ be any point in $\Gamma$ and $\Sigma = \Sigma(x) = \sigma\sigma^\top(x) \in \mathbb{R}^{D \times D}$ be the noise covariance at $x$[1]. Then $\Sigma$ can be decomposed as $\Sigma = \Sigma_\| + \Sigma_\perp + \Sigma_{\|,\perp} + \Sigma_{\perp,\|}$, where $\Sigma_\| := \partial\Phi\Sigma\partial\Phi$, $\Sigma_\perp := (I_D - \partial\Phi)\Sigma(I_D - \partial\Phi)$ and $\Sigma_{\|,\perp} = \Sigma_{\perp,\|}^\top = \partial\Phi\Sigma(I_D - \partial\Phi)$ are the noise covariance in tangent space, normal space and across both spaces, respectively. Then it holds that*

$$\partial^2\Phi[\Sigma] = (\nabla^2 L)^\dagger\partial^2(\nabla L)\left[\Sigma_\|\right] - \partial\Phi\partial^2(\nabla L)\left[\mathcal{L}_{\nabla^2 L}^{-1}(\Sigma_\perp)\right] + 2\partial\Phi\partial^2(\nabla L)\left[(\nabla^2 L)^\dagger\Sigma_{\perp,\|}\right]. \quad (9)$$

## 4.3 MAIN RESULT

Now we are ready to present our main result. It's a direct combination of Theorem B.9 and Lemma 4.5.

**Theorem 4.6.** *Suppose the loss function $L$, the manifold of local minimizer $\Gamma$ and the open neighborhood $U$ satisfy Assumptions 3.1 and 3.2, and $x_\eta(0) = x(0) \in U$ for all $\eta > 0$. If SDE (10) has a global solution $Y$ with $Y(0) = x(0)$ and $Y$ never leaves $U$, i.e., $\mathbb{P}[Y(t) \in U, \forall t \geq 0] = 1$, then for any $T > 0$, $x_\eta(\lfloor T/\eta^2 \rfloor)$ converges in distribution to $Y(T)$ as $\eta \to 0$.*

$$dY(t) = \underbrace{\Sigma_\|^{\frac{1}{2}}(Y)dW(t)}_{\text{Tangent Noise}} + \underbrace{\frac{1}{2}\nabla^2 L(Y)^\dagger\partial^2(\nabla L)(Y)\left[\Sigma_\|(Y)\right]dt}_{\text{Tangent Noise Compensation}}$$

$$+ \frac{1}{2}\partial\Phi(Y)\left(\underbrace{\partial^2(\nabla L)(Y)\left[\nabla^2 L(Y)^\dagger\Sigma_{\perp,\|}(Y)\right]}_{\text{Mixed Regularization}} - \underbrace{\partial^2(\nabla L)(Y)\left[\mathcal{L}_{\nabla^2 L}^{-1}(\Sigma_\perp(Y))\right]}_{\text{Normal Regularization}}\right)dt, \quad (10)$$

*where $\Sigma \equiv \sigma\sigma^\top$ and $\Sigma_\|, \Sigma_\perp, \Sigma_{\perp,\|}$ are defined in Lemma 4.5.*

Based on the above theorem, the limiting dynamics of SGD can be understood as follows: (a) the **tangent noise**, $\Sigma_\|^{1/2}(Y)dW(t)$, is preserved, and the second term of (10) can be viewed as the necessary **tangent noise compensation** for the limiting dynamics to stay on $\Gamma$. Indeed, Lemma C.7 shows that the value of the second term only depends on $\Gamma$ itself, i.e., it's same for all loss $L$ which locally defines the same $\Gamma$. (b) The noise in the normal space is killed since the limiting dynamics always stay on $\Gamma$. However, its second order effect (Itô correction term) takes place as a vector field on $\Gamma$, which induces the **Noise Regularization** and **Mixed Regularization** term, corresponding to the mixed and normal noise covariance respectively.

**Remark 4.7.** *In Appendix B.4 we indeed prove a stronger version of Theorem 4.6 that the sample paths of SGD converge in distribution, i.e., let $\tilde{x}_\eta(t) = x_\eta(\lfloor t/\eta^2 \rfloor)$, then $\tilde{x}_\eta$ weakly converges to $Y$ on $[0, T]$. Moreover, we only assume the existence of a global solution for ease of presentation. As long as there exists a compact $K \subseteq \Gamma$ such that $Y$ stays in $K$ on $[0, T]$ with high probability, Theorem B.9 still provides the convergence of SGD iterates (stopped at the boundary of $K$) before time $T$ with high probability.*

## 5 IMPLICATIONS AND EXAMPLES

In this section, we derive the limiting dynamics for two notable noise types, where we fix the expected loss $L$ and the noise distribution, and only drive $\eta$ to 0. The proofs are deferred into Appendix C.3.

---

[1]For notational convenience, we drop dependency on $x$.

**Type I: Isotropic Noise.** Isotropic noise means $\Sigma(x) \equiv I_D$ for any $x \in \Gamma$ (Shi et al., 2020). The following theorem shows that the limiting diffusion with isotropic noise can be viewed as a Brownian Motion plus Riemannian Gradient Flow with respect to the pseudo-determinant of $\nabla^2 L$.

**Corollary 5.1** (Limiting Diffusion for Isotropic Noise). *If $\Sigma \equiv I_D$ on $\Gamma$, SDE (10) is then*

$$\mathrm{d}Y(t) = \underbrace{\partial\Phi(Y)\mathrm{d}W + \frac{1}{2}\nabla^2 L(Y)^\dagger \partial^2(\nabla L)(Y)\left[\partial\Phi(Y)\right]\mathrm{d}t}_{\text{Brownian Motion on Manifold}} - \underbrace{\frac{1}{2}\partial\Phi(Y)\nabla(\ln|\nabla^2 L(Y)|_+)\mathrm{d}t}_{\text{Normal Regularization}} \quad (11)$$

*where $|\nabla^2 L(Y)|_+ = \lim_{\alpha\to 0}\frac{|\nabla^2 L(Y)+\alpha I_D|}{\alpha^{D-\mathrm{rank}(\nabla^2 L(Y))}}$ is the* pseudo-determinant *of $\nabla^2 L(Y)$. $|\nabla^2 L(Y)|_+$ is also equal to the sum of log of non-zero eigenvalue values of $\nabla^2 L(Y)$.*

**Type II: Label Noise.** When doing SGD for $\ell_2$-regression on dataset $\{(z_i, y_i)\}_{i=1}^n$, adding label noise (Blanc et al., 2020; Damian et al., 2021) means replacing the true label at iteration $k$, $y_{i_k}$, by a fresh noisy label $\tilde{y}_{i_k} := y_{i_k} + \delta_k$, where $\delta_k \overset{\text{i.i.d.}}{\sim} \mathrm{Unif}\{-\delta, \delta\}$ for some constant $\delta > 0$. Then the corresponding loss becomes $\frac{1}{2}(f_{i_k}(x) - \tilde{y}_{i_k})^2$, where $f_{i_k}(x)$ is the output of the model with parameter $x$ on data $z_{i_k}$. So the label noise SGD update is

$$x_{k+1} = x_k - \eta/2 \cdot \nabla_x \left(f_{i_k}(x_k) - y_{i_k} + \delta_{i_k}\right)^2 = x_k - \eta(f_{i_k}(x_k) - y_{i_k} + \delta_k)\nabla_x f_{i_k}(x_k). \quad (12)$$

Suppose the model can achieve the global minimum of the loss $L(x) := \frac{1}{2}\mathbb{E}[(f_i(x) - \tilde{y}_i)^2]$ at $x_*$, then the model must interpolate the whole dataset, i.e., $f_i(x_*) = y_i$ for all $i \in [n]$, and thus here the manifold $\Gamma$ is a subset of $\{x \in \mathbb{R}^D \mid f_i(x) = y_i, \ \forall i \in [n]\}$. Here the key property of the label noise used in previous works is $\Sigma(x) = \frac{\delta^2}{n}\sum_{i=1}^n \nabla_x f_i(x)\nabla_x f_i(x)^\top = \delta^2 \nabla^2 L(x)$. Lately, Damian et al. (2021) further generalizes the analysis to other losses, e.g., logistic loss and exponential loss, as long as they satisfy $\Sigma(x) = c\nabla^2 L(x)$ for some constant $c > 0$.

In sharp contrast to the delicate discrete-time analysis in Blanc et al. (2020) and Damian et al. (2021), the following corollary recovers the same result but with much simpler analysis – taking derivatives is all you need. Under our framework, we no longer need to do Taylor expansion manually nor carefully control the infinitesimal variables of different orders together. It is also worth mentioning that our framework immediately gives a global analysis of $\Theta(\eta^{-2})$ steps for SGD, far beyond the local coupling analysis in previous works. In Section 6, we will see how such global analysis allows us to prove a concrete generalization upper bound in a non-convex problem, the overparametrized linear model (Woodworth et al., 2020; HaoChen et al., 2020).

**Corollary 5.2** (Limiting Flow for Label Noise). *If $\Sigma \equiv c\nabla^2 L$ on $\Gamma$ for some constant $c > 0$, SDE (10) can be simplified into (13) where the regularization is from the noise in the normal space.*

$$\mathrm{d}Y(t) = -1/4 \cdot \partial\Phi(Y(t))\nabla \mathrm{tr}[c\nabla^2 L(Y(t))]\mathrm{d}t. \quad (13)$$

# 6 PROVABLE GENERALIZATION BENEFIT WITH LABEL NOISE

In this section, we show provable benefit of label noise in generalization using our framework (Theorem B.7) in a concrete setting, the *overparametrized linear models* (OLM) (Woodworth et al., 2020). While the existing implicit regularization results for Gradient Flow often relates the generalization quality to initialization, e.g., Woodworth et al. (2020) shows that for OLM, small initialization corresponds to the *rich* regime and prefers solutions with small $\ell_1$ norm while large initialization corresponds to the *kernel* regime and prefers solutions with small $\ell_2$ norm, our result Theorem 6.1 surprisingly proves that even if an OLM is initialized in the kernel regime, label noise SGD can still help it escape and then enter the rich regime by minimizing its weighted $\ell_1$ norm. When the groundtruth is $\kappa$-sparse, this provides a $\widetilde{O}(\kappa \ln d)$ vs $\Omega(d)$ sample complexity separation between SGD with label noise and GD when both initialized in the kernel regime. Here $d$ is the dimension of the groundtruth. The lower bound for GD in the kernel regime is folklore, but for completeness, we state the result as Theorem 6.7 in Section 6.3 and append its proof in Appendix D.6.

**Theorem 6.1.** *In the setting of OLM, suppose the groundtruth is $\kappa$-sparse and $n \geq \Omega(\kappa \ln d)$ training data are sampled from either i.i.d. Gaussian or Boolean distribution. Then for any initialization $x_{init}$ (except a zero-measure set) and any $\epsilon > 0$, there exist $\eta_0, T > 0$ such that for any $\eta < \eta_0$, OLM trained with label noise SGD (12) with LR equal to $\eta$ for $\lfloor T/\eta^2 \rfloor$ steps returns an $\epsilon$-optimal solution, with probability of $1 - e^{-\Omega(n)}$ over the randomness of the training dataset.*

The proof roadmap of Theorem 6.1 is the following:

1. Show Assumption 3.1 is satisfied, i.e., the set of local minimizers, $\Gamma$, is indeed a manifold and the hessian $\nabla^2 L(x)$ is non-degenerate on $\Gamma$ (by Lemma 6.2);
2. Show Assumption 3.2 is satisfied, i.e., $\Phi(U) \subset \Gamma$ (by Lemma 6.3);
3. Show the limiting flow (13) converges to the minimizer of the regularizer (by Lemma 6.5);
4. Show the minimizer of the regularizer recovers the groundtruth (by Lemma 6.6).

Our setting is more general than HaoChen et al. (2020), which assumes $w^* \in \{0,1\}^d$ and their reparametrization can only express positive linear functions, i.e., $w = u^{\odot 2}$. Their $\widetilde{O}(\kappa^2)$ rate is achieved with a delicate three phase LR schedule, while our $O(\kappa \ln d)$ rate only uses a constant LR.

**Setting:** Let $\{(z_i, y_i)\}_{i \in [n]}$ be the training dataset where $z_1, \ldots, z_n \overset{\text{i.i.d.}}{\sim} \text{Unif}(\{\pm 1\}^d)$ or $\mathcal{N}(0, I_d)$ and each $y_i = \langle z_i, w^* \rangle$ for some unknown $w^* \in \mathbb{R}^d$. We assume that $w^*$ is $\kappa$-sparse for some $\kappa < d$. Denote $x = \binom{u}{v} \in \mathbb{R}^D = \mathbb{R}^{2d}$, and we will use $x$ and $(u, v)$ exchangeably as the parameter of functions defined on $\mathbb{R}^D$ in the sequel. For each $i \in [n]$, define $f_i(x) = f_i(u, v) = z_i^\top(u^{\odot 2} - v^{\odot 2})$. Then we fit $\{(z_i, y_i)\}_{i \in [n]}$ with an overparametrized model through the following loss function:

$$L(x) = L(u, v) = \frac{1}{n} \sum_{i=1}^n \ell_i(u, v), \quad \text{where } \ell_i(u, v) = \frac{1}{2}(f_i(u, v) - y_i)^2. \tag{14}$$

It is straightforward to verify that $\nabla^2 L(x) = \frac{4}{n} \sum_{i=1}^n \binom{z_i \odot u}{-z_i \odot v}\binom{z_i \odot u}{-z_i \odot v}^\top, \forall x \in \Gamma$. For simplicity, we define $Z = (z_1, \ldots, z_n)^\top \in \mathbb{R}^{n \times d}$ and $Y = (y_1, \ldots, y_n)^\top \in \mathbb{R}^n$. Consider the following manifold:

$$\Gamma = \left\{ x = (u^\top, v^\top)^\top \in U : Z(u^{\odot 2} - v^{\odot 2}) = Y \right\}, \quad \text{where } U = (\mathbb{R} \setminus \{0\})^D. \tag{15}$$

We verify that the above loss function $L$ and manifold $\Gamma$ satisfy Assumption 3.1 by Lemma 6.2, and that the neighborhood $U$ and $\Gamma$ satisfy Assumption 3.2 by Lemma 6.3.

**Lemma 6.2.** *Consider the loss $L$ defined in (14) and manifold $\Gamma$ defined in (15). If data is full rank, i.e., $\text{rank}(Z) = n$, then it holds that (a). $\Gamma$ is a smooth manifold of dimension $D - n$; (b). $\text{rank}(\nabla^2 L(x)) = n$ for all $x \in \Gamma$. In particular, $\text{rank}(Z) = n$ holds with probability 1 for Gaussian distribution and with probability $1 - c^d$ for Boolean distribution for some constant $c \in (0, 1)$.*

**Lemma 6.3.** *Consider the loss function $L$ defined in (14), manifold $\Gamma$ and its open neighborhood defined in (15). For gradient flow $\frac{dx_t}{dt} = -\nabla L(x_t)$ starting at any $x_0 \in U$, it holds that $\Phi(x_0) \in \Gamma$.*

**Remark 6.4.** *In previous works (Woodworth et al., 2020; Azulay et al., 2021), the convergence of gradient flow is only assumed. Recently Pesme et al. (2021) proved it for a specific initialization, i.e., $u_j = v_j = \alpha, \forall j \in [n]$ for some $\alpha > 0$. Lemma 6.3 completely removes the technical assumption.*

Therefore, by the result in the previous section, the implicit regularizer on the manifold is $R(x) = \text{tr}(\Sigma(x)) = \text{tr}(\delta^2 \nabla^2 L(x))$. Without loss of generality, we take $\delta = 1$. Hence, it follows that

$$R(x) = \frac{4}{n} \sum_{j=1}^D \left( \sum_{i=1}^n z_{i,j}^2 \right) (u_j^2 + v_j^2). \tag{16}$$

The limiting behavior of label noise SGD is described by a Riemannian gradient flow on $\Gamma$ as follows:

$$dx_t = -1/4 \cdot \partial\Phi(x_t)\nabla R(x_t)dt, \text{ with } x_0 = \Phi(x_{\text{init}}) \in \Gamma. \tag{17}$$

The goal is to show that the above limiting flow will converge to the underlying groundtruth $x^* = \binom{u^*}{v^*}$ where $(u^*, v^*) = ([w^*]_+^{\odot 1/2}, [-w^*]_+^{\odot 1/2})$.

## 6.1 Limiting Flow Converges to Minimizers of Regularizer

In this subsection we show limiting flow (13) starting from anywhere on $\Gamma$ converges to the minimizer of regularizer $R$ (by Lemma 6.5). The proof contains two parts: (a) the limiting flow converges; (b) the limit point of the flow cannot be sub-optimal stationary points. These are indeed the most technical and difficult parts of proving the $O(\kappa \ln d)$ upper bound, where the difficulty comes from the fact that the manifold $\Gamma$ is not compact, and the stationary points of the limiting flow are in fact all located on the boundary of $\Gamma$. However, the limiting flow itself is not even defined on the boundary of the manifold $\Gamma$. Even if we can extend $\partial\Phi(\cdot)\nabla R(\cdot)$ continuously to entire $\mathbb{R}^D$, the continuous extension is not everywhere differentiable.

Thus the non-compactness of $\Gamma$ brings challenges for both (a) and (b). For (a), the convergence for standard gradient flow is often for free, as long as the trajectory is bounded and the objective is

analytic or smooth and semialgebraic. The latter ensures the so-called Kurdyka-Łojasiewicz (KL) inequality (Lojasiewicz, 1963), which implies finite trajectory length and thus the convergence. However, since our flow does not satisfy those nice properties, we have to show that the limiting flow satisfies Polyak-Łojasiewicz condition (a special case of KL condition) (Polyak, 1964) via careful calculation (by Lemma D.16).

For (b), the standard analysis based on center stable manifold theorem shows that gradient descent/flow converges to strict saddle (stationary point with at least one negative eigenvalue in hessian) only for a zero-measure set of initialization (Lee et al., 2016; 2017). However, such analyses cannot deal with the case where the flow is not differentiable at the sub-optimal stationary point. To circumvent this issue, we prove the non-convergence to sub-optimal stationary points with a novel approach: we show that for any stationary point $x$, whenever there exists a descent direction of the regularizer $R$ at $x$, we can construct a potential function which increases monotonically along the flow around $x$, while the potential function is equal to $-\infty$ at $x$, leading to a contradiction. (See proof of Lemma 6.5.)

**Lemma 6.5.** *Let $\{x_t\}_{t \geq 0} \subseteq \mathbb{R}^D$ be generated by the flow defined in* (17) *with any initialization $x_0 \in \Gamma$. Then $x_\infty = \lim_{t \to \infty} x_t$ exists. Moreover, $x_\infty = x^*$ is the optimal solution of* (18).

## 6.2 MINIMIZER OF THE REGULARIZER RECOVERS THE SPARSE GROUNDTRUTH

Note $\frac{1}{n} \sum_{i=1}^n z_{i,j}^2 = 1$ when $z_{i,j} \overset{iid}{\sim} \mathrm{Unif}\{-1, 1\}$, and we can show minimizing $R(x)$ on $\Gamma$, (18), is equivalent to finding the minimum $\ell_1$ norm solution of Equation (14). Standard results in sparse recovery imply that minimum $\ell_1$ norm solution recovers with the sparse groundtruth. The gaussian case is more complicated but still can be proved with techniques from Tropp (2015).

$$\begin{aligned} \text{minimize} \quad & R(x) = \frac{4}{n} \sum_{j=1}^d \left( \sum_{i=1}^n z_{i,j}^2 \right) (u_j^2 + v_j^2), \\ \text{subject to} \quad & Z(u^{\odot 2} - v^{\odot 2}) = Zw^*. \end{aligned} \tag{18}$$

**Lemma 6.6.** *Let $z_1, \ldots, z_n \overset{i.i.d.}{\sim} \mathrm{Unif}(\{\pm 1\}^d)$ or $\mathcal{N}(0, I_d)$. Then there exist some constants $C, c > 0$ such that if $n \geq C\kappa \ln d$, then with probability at least $1 - e^{-cn}$, the optimal solution of* (18), $(\hat{u}, \hat{v})$, *is unique up to sign flips of each coordinate and recovers the groundtruth, i.e., $\hat{u}^{\odot 2} - \hat{v}^{\odot 2} = w^*$.*

## 6.3 LOWER BOUND FOR GRADIENT DESCENT IN THE KERNEL REGIME

In this subsection we show GD needs at least $\Omega(d)$ samples to learn OLM, when initialized in the kernel regime. This lower bound holds for all learning rate schedules and numbers of steps. This is in sharp contrast to the $\widetilde{O}(\kappa \ln d)$ sample complexity upper bound of SGD with label noise. Following the setting of kernel regime in (Woodworth et al., 2020), we consider the limit of $u_0 = v_0 = \alpha \mathbb{1}$, with $\alpha \to \infty$. It holds that $f_i(u_0, v_0) = 0$ and $\nabla f_i(u_0, v_0) = [\alpha z_i, -\alpha z_i]$ for each $i \in [n]$. Standard convergence analysis for NTK (Neural Tangent Kernel, Jacot et al. (2018)) shows that upon convergence, the distance traveled by parameter converges to 0, and thus the learned model shall converge in function space, so is the generalization performance. For ease of illustration, we directly consider the lower bound for test loss when the NTK is fixed throughout the training.

**Theorem 6.7.** *Assume $z_1, \ldots, z_n \overset{i.i.d.}{\sim} \mathcal{N}(0, I_d)$ and $y_i = z_i^\top w^*$, for all $i \in [n]$. Define the loss with linearized model as $L(x) = \sum_{i=1}^n (f_i(x_0) + \langle \nabla f_i(x_0), x - x_0 \rangle - y_i)^2$, where $x = \binom{u}{v}$ and $x_0 = \binom{u_0}{v_0} = \alpha \binom{\mathbb{1}}{\mathbb{1}}$. Then for any groundtruth $w^*$, any learning rate schedule $\{\eta_t\}_{t \geq 1}$, and any fixed number of steps $T$, the expected $\ell_2$ loss of $x(T)$ is at least $(1 - \frac{n}{d}) \|w^*\|_2^2$, where $x(T)$ is the $T$-th iterate of GD on $L$, i.e., $x(t + 1) = x(t) - \eta_t \nabla L(x(t))$, for all $t \geq 0$.*

## 7 CONCLUSION AND FUTURE WORK

We propose a mathematical framework to study the implicit bias of SGD with infinitesimal LR. We show that with arbitrary noise covariance, $\Theta(\eta^{-2})$ steps of SGD converge to a limiting diffusion on certain manifold of local minimizer, as the LR $\eta \to 0$. For specific noise types, this allows us to recover and strengthen results regarding implicit bias in previous works with much simpler analysis. In particular, we show a sample complexity gap between label noise SGD and GD in the kernel regime for a overparametrized linear model, justifying the generalization benefit of SGD. For the future work, we believe our framework can be applied to analyze the implicit bias of SGD in more complex models towards better understanding of the algorithmic regularization induced by stochasticity. It will be valuable to extend our method to other stochastic optimization algorithms, e.g., ADAM, SGD with momentum.

ACKNOWLEDGEMENT

We thank Yangyang Li for pointing us to Katzenberger (1991). We also thank Wei Zhan and Jason Lee for helpful discussions.

The authors acknowledge support from NSF, ONR, Simons Foundation, Schmidt Foundation, Mozilla Research, Amazon Research, DARPA and SRC. ZL is also supported by Microsoft Research PhD Fellowship.

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

## A   PRELIMINARIES ON STOCHASTIC PROCESSES

We first clarify the notations in this paper. For any integer $k$, we denote $\mathcal{C}^k$ as the set of the $k$ times continuously differentiable functions. We denote $a \wedge b = \min\{a, b\}$. For any vector $u, v$ and $\alpha \in \mathbb{R}$, we define $[u \odot v]_i = u_i v_i$ and $[v^{\odot \alpha}]_i = v_i^{\alpha}$. For any matrix $A$, we denote its pseudo inverse by $A^{\dagger}$. For mapping $F : \mathbb{R}^D \to \mathbb{R}^D$, we denote the *Jacobian* of $F$ at $x$ by $\partial F(x) \in \mathbb{R}^{D \times D}$ where the $(i, j)$-th entry is $\partial_j F_i(x)$. We also use $\partial F(x)[u]$ and $\partial^2 F(x)[u, v]$ to denote the first and second order directional derivative of $F$ at $x$ along the derivation of $u$ (and $v$). We abuse the notation of $\partial^2 F$ by viewing it a linear mapping defined on $\mathbb{R}^D \otimes \mathbb{R}^D \cong \mathbb{R}^{D^2}$, in the sense that $\partial^2 F(x)[\Sigma] = \sum_{i,j=1}^{D} \partial^2 F(x)[e_i, e_j] \Sigma_{ij}$, for any $\Sigma \in \mathbb{R}^{D \times D}$. For any submanifold $\Gamma \subset \mathbb{R}^D$ and $x \in \Gamma$, we denote by $T_x(\Gamma)$ the tangent space of $\Gamma$ at $x$ and $T_x^{\perp}(\Gamma)$ the normal space of $\Gamma$ at $x$.

Next, we review a few basics of stochastic processes that will be useful for proving our results, so that our paper will be self-contained. We refer the reader to classics like Karatzas & Shreve (2014); Billingsley (2013); Pollard (2012) for more systematic derivations.

Throughout the rest of this section, let $\mathcal{E}$ be a Banach space equipped with norm $\|\cdot\|$, e.g., $(\mathbb{R}, |\cdot|)$ and $(\mathbb{R}^D, \|\cdot\|_2)$.

### A.1   CÀDLÀGFUNCTION AND METRIC

**Definition A.1** (Càdlàgfunction). Let $T \in [0, \infty]$. A function $g : [0, T) \to E$ is *càdlàg* if for all $t \in [0, T)$ it is right-continuous at $t$ and its left limit $g(t-)$ exists. Let $\mathcal{D}_{\mathcal{E}}[0, T)$ be the set of all càdlàgfunction mapping $[0, T)$ into $\mathcal{E}$. We also use $\mathcal{D}_{\mathcal{E}}[0, T)$ to denote the set of all continuous function mapping $[0, T)$ into $\mathcal{E}$. By definition, $\mathcal{C}_{\mathcal{E}}[0, T) \subset \mathcal{D}_{\mathcal{E}}[0, T)$.

**Definition A.2** (Continuity modulus). For any function $f : [0, \infty) \to \mathcal{E}$ and any interval $I \subseteq [0, \infty)$, we define

$$\omega(f; I) = \sup_{s, t \in I} \|f(s) - f(t)\|.$$

For any $N \in \mathbb{N}$ and $\theta > 0$, we further define the continuity modulus of continuous $f$ as

$$\omega_N(f, \theta) = \sup_{0 \le t \le t + \theta \le N} \{\omega(f; [t, t + \theta])\}.$$

Moreover, the continuity modulus of càdlàg$f \in \mathcal{D}_{\mathcal{E}}[0, \infty)$ is defined as

$$\omega'_N(f, \theta) = \inf\left\{\max_{i \le r} \omega(f; [t_{i-1}, t_i]) : 0 \le t_0 < \cdots < t_r = N, \inf_{i < r}(t_i - t_{i-1}) \ge \theta\right\}.$$

**Definition A.3** (Jump). For any $g \in \mathcal{D}_{\mathcal{E}}[0, T)$, we define the jump of $g$ at $t$ to be

$$\Delta g(t) = g(t) - g(t-).$$

For any $\delta > 0$, we define $h_{\delta} : [0, \infty) \to [0, \infty)$ by

$$h_{\delta}(r) = \begin{cases} 0 & \text{if } r \le \delta \\ 1 - \delta/r & \text{if } r \ge \delta \end{cases}.$$

We then further define $J_{\delta} : \mathcal{D}_{\mathbb{R}^D}[0, \infty) \to \mathcal{D}_{\mathbb{R}^D}[0, \infty)$ (Katzenberger, 1991) as

$$J_{\delta}(g)(t) = \sum_{0 < s \le t} h_{\delta}(\|\Delta g(s)\|) \Delta g(s). \tag{19}$$

**Definition A.4** (Skorohod metric on $\mathcal{D}_{\mathcal{E}}[0, \infty)$). For each finite $T > 0$ and each pair of functions $f, g \in \mathcal{D}_{\mathcal{E}}[0, \infty)$, define $d_T(f, g)$ as the infimum of all those values of $\delta$ for which there exist grids $0 \le t_0 < t_1 < \cdots < t_m$ and $0 < s_0 < s_1 < \cdots < \cdots < s_m$, with $t_k, s_k \ge T$, such that $|t_i - s_i| \le \delta$ for $i = 0, \ldots, k$, and

$$\|f(t) - g(s)\| \le \delta \qquad \text{if } (t, s) \in [t_i, t_{i+1}) \times [s_i, s_{i+1})$$

for $i = 0, \ldots, k - 1$. The *Skorohod metric* on $\mathcal{D}_{\mathcal{E}}[0, \infty)$ is defined to be

$$d(f, g) = \sum_{T=1}^{\infty} 2^{-T} \min\{1, d_T(f, g)\}.$$

A.2 STOCHASTIC PROCESSES AND STOCHASTIC INTEGRAL

Let $(\Omega, \mathcal{F}, \{\mathcal{F}_t\}_{t\geq 0}, \mathbb{P})$ be a filtered probability space.

**Definition A.5** (Cross variation). Let $X$ and $Y$ be two $\{\mathcal{F}_t\}_{t\geq 0}$-adapted stochastic processes such that $X$ has sample paths in $\mathcal{D}_{\mathbb{R}^{D\times e}}[0, \infty)$ and $Y$ has samples paths in $\mathcal{D}_{\mathbb{R}^e}[0, \infty)$, then the cross variation of $X$ and $Y$ on $(0, t]$, denoted by $[X, Y](t)$, is defined to be the limit of

$$\sum_{i=0}^{m-1} (X(t_{i+1}) - X(t_i))(Y(t_{i+1}) - Y(t_i))$$

in probability as the mesh size of $0 = t_0 < t_1 < \cdots < t_m = t$ goes to 0, if it exists. Moreover, for $Y$ itself, we write

$$[Y] = \sum_{i=1}^e [Y_i, Y_i]$$

**Definition A.6** (Martingale). Let $\{X(t)\}_{t\geq 0}$ be a $\{\mathcal{F}_t\}_{t\geq 0}$-adapted stochastic process. If for all $0 \leq s \leq t$, it holds that

$$\mathbb{E}[X(t) \mid \mathcal{F}_s] = X(s),$$

then $X$ is called a martingale.

**Definition A.7** (Local martingale). Let $\{X(t)\}_{t\geq 0}$ be a $\{\mathcal{F}_t\}_{t\geq 0}$-adapted stochastic process. If there exists a sequence of $\{\mathcal{F}_t\}_{t\geq 0}$-stopping time, $\{\tau_k\}_{k\geq 0}$, such that

- $\mathbb{P}[\tau_k < \tau_{k+1}] = 1, \mathbb{P}[\lim_{k\to\infty} \tau_k = \infty] = 1$,

- and $\{X^{\tau_k}(t)\}_{t\geq 0}$ is a $\{\mathcal{F}_t\}_{t\geq 0}$-adapted martingale,

then $X$ is called a local martingale.

**Definition A.8** (Semimartingale). Let $\{X(t)\}_{t\geq 0}$ be a $\{\mathcal{F}_t\}_{t\geq 0}$-adapted stochastic process. If there exists a local martingale $\{M(t)\}_{t\geq 0}$ and a càdlàg $\{\mathcal{F}_t\}_{t\geq 0}$-adapted process $\{A(t)\}_{t\geq 0}$ with bounded total variation that $X(t) = M(t) + A(t)$, then $X$ is called a semimartingale.

**Lemma A.9** (Itô's Lemma). *Let $\{X(t)\}_{t\geq 0}$ be defined through the following Itô drift-diffusion process:*

$$\mathrm{d}X(t) = \mu(t)\mathrm{d}t + \sigma(t)\mathrm{d}W(t).$$

*where $\{W(t)\}_{t\geq 0}$ is the standard Brownian motion. Then for any twice differentiable function $f$, it holds that*

$$\mathrm{d}f(t, X(t)) = \left(\frac{\partial f}{\partial t} + (\nabla_x f)^\top \mu_t + \frac{1}{2}\operatorname{tr}[\sigma^\top \nabla_x^2 f \sigma]\right)\mathrm{d}t + (\nabla_x f)^\top \sigma(t)\mathrm{d}W(t).$$

A.3 WEAK CONVERGENCE FOR STOCHASTIC PROCESSES

Let $(\mathcal{D}_{\mathcal{E}}[0, \infty), \mathcal{A}, d)$ be a metric space equipped with a $\sigma$-algebra $\mathcal{A}$ and the Skorohod metric defined in the previous subsection.

Let $\{X_n\}_{n\geq 0}$ be a sequence of stochastic processes on a sequence of probability spaces $\{(\Omega_n, \mathcal{F}_n, \mathbb{P}_n)\}_{n\geq 0}$ such that each $X_n$ has sample paths in $\mathcal{D}_{\mathcal{E}}[0, \infty)$. Also, let $X$ be a stochastic process on $(\Omega, \mathcal{F}, \mathbb{P})$ with sample paths on $\mathcal{D}_{\mathcal{E}}[0, \infty)$.

**Definition A.10** (Weak convergence). A sequence of stochastic process $\{X_n\}_{n\geq 0}$ is said to *converge in distribution* or *weakly converge* to $X$ (written as $X_n \Rightarrow X$) if and only if for all $\mathcal{A}$-measurable, bounded, and continuous function $f : \mathcal{D}_{\mathcal{E}}[0, \infty) \to \mathbb{R}$, it holds that

$$\lim_{n\to\infty} \mathbb{E}[f(X_n)] = \mathbb{E}[f(X)]. \tag{20}$$

Though we define weak convergence for a countable sequence of stochastic processes, but it is still valid if we index the stochastic processes by real numbers, e.g., $\{X_\eta\}_{\eta\geq 0}$, and consider the weak convergence of $X_\eta$ as $\eta \to 0$. This is because the convergence in (20) is for a sequence of real numbers, which is also well-defined if we replace $\lim_{n\to\infty}$ by $\lim_{\eta\to 0}$.

**Definition A.11** ($\delta$-Prohorov distance). *Let $\delta > 0$. For any two probability measures $P$ and $Q$ on a metric space with metric $d$, let $(X, Y)$ be a coupling such that $P$ is the marginalized law of $X$ and $Q$ that of $Y$. We define*

$$\rho^\delta(P, Q) = \inf\{\epsilon > 0 : \exists (X, Y), \mathbb{P}[d(X, Y) \geq \epsilon] \leq \delta\}.$$

Note this distance is not a metric because it does not satisfy triangle inequality.

**Definition A.12** (Prohorov metric). *For any two probability measures $P$ and $Q$ on a metric space with metric $d$, let $(X, Y)$ be a coupling such that $P$ is the marginalized law of $X$ and $Q$ that of $Y$. Denote the marginal laws of $X$ and $Y$ by $\mathcal{L}(X)$ and $\mathcal{L}(Y)$ respectively. We define the Prohorov metric as*

$$\rho(P, Q) = \inf\{\epsilon > 0 : \exists (X, Y), \mathcal{L}(X) = P, \mathcal{L}(Y) = Q, \mathbb{P}[d(X, Y) \geq \epsilon] \leq \epsilon\}.$$

It can be shown that $X_n \Rightarrow X$ is equivalent to $\lim_{n \to \infty} \rho(X_n, X) = 0$.

**Theorem A.13** (Skorohod Representation Theorem). *Suppose $P_n, n = 1, 2, \ldots$ and $P$ are probability measures on $\mathcal{E}$ such that $P_n \Rightarrow P$. Then there is a probability space $(\Omega, \mathcal{F}, \mathbb{P})$ on which are defined $\mathcal{E}$-valued random variables $X_n, n = 1, 2, \ldots$ and $X$ with distributions $P_n$ and $P$ respectively, such that $\lim_{n \to \infty} X_n = X$ a.s.*

The main convergence result in Katzenberger (1991) (Theorem B.7) are in the sense of Skorohod metric in Definition A.4, which is harder to understand and use compared to the more common *uniform metric* (Definition A.14). However, convergence in Skorohod metric and uniform metric indeed coincide with each other when the limit is in $\mathcal{C}_{\mathbb{R}^D}[0, \infty)$, i.e., the continuous functions.

**Definition A.14** (Uniform metric on $\mathcal{D}_{\mathcal{E}}[0, \infty)$). *For each finite $T > 0$ and each pair of functions $f, g \in \mathcal{D}_{\mathcal{E}}[0, T]$, the uniform metric is defined to be*

$$d_U(f, g; T) = \sup_{t \in [0, T)} \|f(t) - g(t)\|.$$

The *uniform metric* on $\mathcal{D}_{\mathcal{E}}[0, \infty)$ is defined to be

$$d_U(f, g) = \sum_{T=1}^{\infty} 2^{-T} \min\{1, d_U(f, g; T)\}.$$

**Lemma A.15** (Problem 7, Section 5, Pollard (2012)). *If $X_n \Rightarrow X$ in the Skorohod sense, and $X$ has sample paths in $\mathcal{C}_{\mathbb{R}^D}[0, \infty)$, then $X_n \Rightarrow X$ in the uniform metric.*

**Remark A.16.** *We shall note the uniform metric defined above is weaker than $\sup_{t \in [0, \infty)} \|f(t) - g(t)\|$. Convergence in the uniform metric on $[0, \infty]$ defined in Definition A.14 is equivalent to convergence in the uniform metric on each compact set $[0, T]$ for $T \in \mathbb{N}^+$. The same holds for the Skorohod topology.*

## B  LIMITING DIFFUSION OF SGD

In this section, we give a complete derivation of the limiting diffusion of SGD. Here we use $\Rightarrow$ to denote the convergence in distribution. For any $U \subseteq \mathbb{R}^D$, we denote by $\mathring{U}$ its interior. For linear space $S$, we use $S^\perp$ to denote its orthogonal complement.

First, as mentioned in Assumption 3.2, we verify that the mapping $\Phi$ is $\mathcal{C}^2$ in Lemma B.1. In Appendix B.1 we discuss how different time scalings could affect the coefficients in SDE (2) and (3). Then we check the necessary conditions for applying the results in Katzenberger (1991) in Appendix B.2 and recap the corresponding theorem for the asymptotically continuous case in Appendix B.3. Finally, we provide a user-friendly interface for Katzenberger's theorem in Appendix B.4.

**Lemma B.1** (Implication of Falconer (1983)). *Under Assumption 3.2, $\Phi$ is $\mathcal{C}^2$ on $U$.*

*Proof of Lemma B.1.* Applying Theorem 5.1 of Falconer (1983) with $f(\cdot) = \phi(\cdot, 1)$ suffices. $\qquad \square$

## B.1 Approximating SGD by SDE

Let's first clarify how we derive the SDEs, (2) and (3), that approximate SGD (1) under different time scalings. Recall $W(t)$ is $\Xi$-dimensional Brownian motion and that $\sigma(X) : \mathbb{R}^D \to \mathbb{R}^{D \times \Xi}$ is a deterministic noise function. As proposed by Li et al. (2017), one approach to approximate SGD (1) by SDE is to consider the following SDE:

$$\mathrm{d}X(t) = -\nabla L(X(t))\mathrm{d}t + \sqrt{\eta}\sigma(X(t))\mathrm{d}W(t),$$

where the time correspondence is $t = k\eta$, i.e., $X(k\eta) \approx x_\eta(k)$.

Now rescale the above SDE by considering $\tilde{X}(t) = X(t\eta)$, which then yields

$$\begin{aligned}
\mathrm{d}\tilde{X}(t) = \mathrm{d}X(t\eta) &= -\nabla L(X(t\eta))\mathrm{d}(t\eta) + \sqrt{\eta}\sigma(X(t\eta))\mathrm{d}W(t\eta) \\
&= -\eta\nabla L(X(t\eta))\mathrm{d}t + \sqrt{\eta}\sigma(X(t\eta))\mathrm{d}W(t\eta).
\end{aligned}$$

Now we define $W'(t) = \frac{1}{\sqrt{\eta}}W(t\eta)$, and it's easy to verify that $W'(t)$ is also a $\Xi$-dimensional brownian motion, which means $W' \overset{d}{=} W$, i.e., $W$ and $W'$ have the same sample paths in $\mathcal{C}_{\mathbb{R}^d}[0, \infty)$. Thus

$$\begin{aligned}
\mathrm{d}\tilde{X}(t) &= -\eta\nabla L(X(t\eta))\mathrm{d}t + \eta\sigma(X(t\eta))\mathrm{d}W'(t) \\
&= -\eta\nabla L(\tilde{X}(t))\mathrm{d}t + \eta\sigma(\tilde{X}(t))\mathrm{d}W'(t),
\end{aligned}$$

where the time correspondence is $t = k$, i.e., $\tilde{X}(k) \approx x_\eta(k)$. The above SDE is exactly the same as (2).

Then, to accelerate the above SDE by $\eta^{-2}$ times, let's define $\bar{X}(t) = \tilde{X}(t/\eta^2)$. Then it follows that

$$\begin{aligned}
\mathrm{d}\bar{X}(t) = \mathrm{d}\tilde{X}(t/\eta^2) &= -\eta\nabla L(\tilde{X}(t/\eta^2))\mathrm{d}t/\eta^2 + \eta\sigma(\tilde{X}(t/\eta^2))\mathrm{d}W(t/\eta^2) \\
&= -\frac{1}{\eta}\nabla L(\bar{X}(t))\mathrm{d}t + \sigma(\bar{X}(t))\mathrm{d}\left(\eta W(t/\eta^2)\right)
\end{aligned}$$

Again note that $\eta W(t/\eta^2) \overset{d}{=} W(t)$ in sample paths and thus is also a $\Xi$-Brownian motion. Here the time correspondence is $t = k\eta^2$, i.e., evolving for constant time with the above SDE approximates $\Omega(1/\eta^2)$ steps of SGD. In this way, we derive SDE (3) in the main context.

## B.2 Necessary Conditions

Below we collect the necessary conditions imposed on $\{Z_n\}_{n\geq 1}$ and $\{A_n\}_{n\geq 1}$ in Katzenberger (1991). Recall that we consider the following stochastic process

$$X_n(t) = X(0) + \int_0^t \sigma(X_n(s))\mathrm{d}Z_n(s) - \int_0^t \nabla L(X_n(s))\mathrm{d}A_n(s).$$

For any stopping time $\tau$, the stopped process is defined as $X_n^\tau(t) = X_n(t \wedge \tau)$. For any compact $K \subset U$, we define the stopping time of $X_n$ leaving $K$ as $\lambda_n(K) = \inf\{t \geq 0 \mid X_n(t-) \notin \mathring{K}$ or $X_n(t) \notin \mathring{K}\}$.

**Condition B.2.** The integrator sequence $\{A_n\}_{n\geq 1}$ is *asymptotically continuous*: $\sup_{t>0} |A_n(t) - A_n(t-)| \Rightarrow 0$ where $A_n(t-) = \lim_{s \to t-} A_n(s)$ is the left limit of $A_n$ at $t$.

**Condition B.3.** The integrator sequence $\{A_n\}_{n\geq 1}$ *increases infinitely fast*: $\forall \epsilon > 0, \inf_{t\geq 0}(A_n(t+\epsilon) - A_n(t)) \Rightarrow \infty$.

**Condition B.4** (Eq.(5.1), Katzenberger 1991). For every $T > 0$, as $n \to \infty$, it holds that

$$\sup_{0 < t \leq T \wedge \lambda_n(K)} \|\Delta Z_n(t)\|_2 \Rightarrow 0.$$

**Condition B.5** (Condition 4.2, Katzenberger 1991). For each $n \geq 1$, let $Y_n$ be a $\{\mathcal{F}_t^n\}$-semimartingale with sample paths in $\mathcal{D}_{\mathbb{R}^D}[0, \infty)$. Assume that for some $\delta > 0$ (allowing $\delta = \infty$) and every $n \geq 1$ there exist stopping times $\{\tau_n^m \mid m \geq 1\}$ and a decomposition of $Y_n - J_\delta(Y_n)$

into a local martingale $M_n$ plus a finite variation process $F_n$ such that $\mathbb{P}[\tau_n^m \leq m] \leq 1/m$, $\{[M_n](t \wedge \tau_n^m) + T_{t \wedge \tau_n^m}(F_n)\}_{n \geq 1}$ is uniformly integrable for every $t \geq 0$ and $m \geq 1$, and

$$\lim_{\gamma \to 0} \limsup_{n \to \infty} \mathbb{P}\left[\sup_{0 \leq t \leq T}(T_{t+\gamma}(F_n) - T_t(F_n)) > \epsilon\right] = 0,$$

for every $\epsilon > 0$ and $T > 0$, where $T_t(\cdot)$ denotes total variation on the interval $[0, t]$.

**Lemma B.6.** *For SGD iterates defined using the notation in Lemma 4.2, the sequences $\{A_n\}_{n \geq 1}$ and $\{Z_n\}_{n \geq 1}$ satisfy Condition B.2, B.3, B.4 and B.5.*

*Proof of Lemma B.6.* Condition B.2 is obvious from the definition of $\{A_n\}_{n \geq 1}$.

Next, for any $\epsilon > 0$ and $t \in [0, T]$, we have

$$A_n(t + \epsilon) - A_n(t) = \eta_n \cdot \left\lfloor \frac{t + \epsilon}{\eta_n^2} \right\rfloor - \eta_n \cdot \left\lfloor \frac{t}{\eta_n^2} \right\rfloor \geq \frac{t + \epsilon - \eta_n^2}{\eta_n} - \frac{t}{\eta_n} = \frac{\epsilon - \eta_n^2}{\eta_n},$$

which implies that $\inf_{0 \leq t \leq T}(A_n(t + \epsilon) - A_n(t)) > \epsilon/(2\eta_n)$ for small enough $\eta_n$. Then taking $n \to \infty$ yields the Condition B.3.

For Condition B.4, note that

$$\Delta Z_n(t) = \begin{cases} \eta_n \sqrt{\Xi}(\mathbb{1}_{\xi_k} - \frac{1}{\Xi}\mathbb{1}) & \text{if } t = k \cdot \eta_n^2, \\ 0 & \text{otherwise.} \end{cases}$$

Therefore, we have $\|\Delta Z_n(t)\|_2 \leq 2\eta_n\sqrt{\Xi}$ for all $t > 0$. This implies that $\|\Delta Z_n(t)\|_2 \to 0$ uniformly over $t > 0$ as $n \to \infty$, which verifies Condition B.4.

We proceed to verify Condition B.5. By the definition of $Z_n$, we know that $\{Z_n(t)\}_{t \geq 0}$ is a jump process with independent increments and thus is a martingale. Therefore, by decomposing $Z_n = M_n + F_n$ with $M_n$ being a local martingale and $F_n$ a finite variation process, we must have $F_n = 0$ and $M_n$ is $Z_n$ itself. It then suffices to show that $[M_n](t \wedge \tau_n^m)$ is uniformly integrable for every $t \geq 0$ and $m \geq 1$. Since $M_n$ is a pure jump process, we have

$$[M_n](t \wedge \tau_n^m) = \sum_{0 < s \leq t \wedge \tau_n^m} \|\Delta M_n(s)\|_2^2 \leq \sum_{0 < s \leq t} \|\Delta M_n(s)\|_2^2$$

$$= \sum_{k=1}^{\lfloor t/\eta_n^2 \rfloor} \left\| \eta_n\sqrt{\Xi}\left(\mathbb{1}_{\xi_k} - \frac{1}{\Xi}\mathbb{1}\right) \right\|_2^2 \leq 4\Xi \sum_{k=1}^{\lfloor t/\eta_n^2 \rfloor} \eta_n^2 \leq 4\Xi t.$$

This implies that $[M_\eta](t \wedge \tau_\eta^m)$ is universally bounded by $4t$, and thus $[M_\eta](t \wedge \tau_\eta^m)$ is uniformly integrable. This completes the proof. $\qquad\square$

**Lemma 4.2.** *Let $\{\eta_n\}_{n=1}^\infty$ be any positive sequence with $\lim_{n \to \infty} \eta_n = 0$, $A_n(t) = \eta_n \lfloor t/\eta_n^2 \rfloor$, and $Z_n(t) = \eta_n \sum_{k=1}^{\lfloor t/\eta_n^2 \rfloor} \sqrt{\Xi}(\mathbb{1}_{\xi_k} - \frac{1}{\Xi}\mathbb{1})$, where $\xi_1, \xi_2, \ldots \overset{i.i.d.}{\sim} \text{Unif}([\Xi])$. Then with the same initialization $X_n(0) = x_{\eta_n}(0) \equiv X(0)$, $X_n(k\eta_n^2)$ defined by (6) is a Katzenberger process and is equal to $x_{\eta_n}(k)$ defined in (1) with LR equal to $\eta_n$ for all $k \geq 1$. Moreover, the counterpart of (7) is*

$$Y(t) = \Phi(X(0)) + \int_0^t \partial\Phi(Y)\sigma(Y)\mathrm{d}W(s) + \frac{1}{2}\int_0^t \partial^2\Phi(Y)[\Sigma(Y)]\mathrm{d}s, \tag{8}$$

*where $\Sigma \equiv \sigma\sigma^\top$ and $\{W(t)\}_{t \geq 0}$ is a $\Xi$-dimensional standard Brownian motion.*

*Proof of Lemma 4.2.* For any $n \geq 1$, it suffices to show that given $X_n(k\eta_n^2) = x_{\eta_n}(k)$, we further have $X_n((k+1)\eta_n^2) = x_{\eta_n}(k+1)$. By the definition of $X_n(t)$, we have

$$X_n((k+1)\eta_n^2) - X_n(k\eta_n^2)$$
$$= -\int_{k\eta_n^2}^{(k+1)\eta_n^2} \nabla L(X_n(t))\mathrm{d}A_n(t) + \int_{k\eta_n^2}^{(k+1)\eta_n^2} \sigma(X_n(t))\mathrm{d}Z_n(t)$$
$$= -\nabla L(X_n(k\eta_n^2))(A_n((k+1)\eta_n^2) - A_n(k\eta_n^2)) + \sigma(X_n(k\eta_n^2))(Z_n((k+1)\eta_n^2) - Z_n(k\eta_n^2))$$
$$= -\eta_n\nabla L(X_n(k\eta_n^2)) + \eta_n\sqrt{\Xi}\sigma_{\xi_k}(X_n(k\eta_n^2))$$
$$= -\eta_n\nabla L(x_{\eta_n}(k)) + \eta_n\sqrt{\Xi}\sigma_{\xi_k}(x_{\eta_n}(k)) = x_{\eta_n}(k+1) - x_{\eta_n}(k)$$

where the second equality is because $A_n(t)$ and $Z_n(t)$ are constant on interval $[k\eta_n^2, (k+1)\eta_n^2)$. This confirms the alignment between $\{X_n(k\eta_n^2)\}_{k\geq 1}$ and $\{x_{\eta_n}(k)\}_{k\geq 1}$.

For the second claim, note that $\sigma(x)\mathbb{E}Z_n(t) \equiv 0$ for all $x \in \mathbb{R}^D, t \geq 0$ (since the noise has zero-expectation) and that $\{Z_n(t) - \mathbb{E}Z_n(t)\}_{t\geq 0}$ will converge in distribution to a Brownian motion by the classic functional central limit theorem (see, for example, Theorem 4.3.5 in Whitt (2002)). Thus, the limiting diffusion of $X_n$ as $n \to \infty$ can be obtained by substituting $Z$ with the standard Brownian motion $W$ in (22). This completes the proof. $\qquad\square$

### B.3 KATZENBERGER'S THEOREM FOR ASYMPTOTICALLY CONTINUOUS CASE

The full Katzenberger's theorem deals with a more general case, which only requires the sequence of intergrators to be *asymptotically continuous*, thus including SDE (3) and SGD (1) with $\eta$ goes to 0.

To describe the results in Katzenberger (1991), we first introduce some definitions. For each $n \geq 1$, let $(\Omega^n, \mathcal{F}^n, \{\mathcal{F}_t^n\}_{t\geq 0}, \mathbb{P})$ be a filtered probability space, $Z_n$ an $\mathbb{R}^e$-valued cadlag $\{\mathcal{F}_t^n\}$-semimartingale with $Z_n(0) = 0$ and $A_n$ a real-valued cadlag $\{\mathcal{F}_t^n\}$-adapted nondecreasing process with $A_n(0) = 0$. Let $\sigma_n : U \to \mathbb{M}(D, e)$ be continuous with $\sigma_n \to \sigma$ uniformly on compact subsets of $U$. Let $X_n$ be an $\mathbb{R}^D$-valued cadlag $\{\mathcal{F}_t^n\}$-semimartingale satisfying, for all compact $K \subset U$,

$$X_n(t) = X(0) + \int_0^t \sigma(X_n)\mathrm{d}Z_n + \int_0^t -\nabla L(X_n)\mathrm{d}A_n \tag{21}$$

for all $t \leq \lambda_n(K)$ where $\lambda_n(K) = \inf\{t \geq 0 \mid X_n(t-) \notin \mathring{K} \text{ or } X_n(t) \notin \mathring{K}\}$ is the stopping time of $X_n$ leaving $K$.

**Theorem B.7** (Theorem 6.3, Katzenberger 1991). *Suppose $X(0) \in U$, Assumptions 3.1 and 3.2, Condition B.2, B.3, B.4 and B.5 hold. For any compact $K \subset U$, define $\mu_n(K) = \inf\{t \geq 0 \mid Y_n(t-) \notin \mathring{K} \text{ or } Y_n(t) \notin \mathring{K}\}$, then the sequence $\{(Y_n^{\mu_n(K)}, Z_n^{\mu_n(K)}, \mu_n(K))\}$ is relatively compact in $\mathcal{D}_{\mathbb{R}^{D\times e}}[0, \infty) \times [0, \infty)$. If $(Y, Z, \mu)$ is a limit point of this sequence under the skorohod metric (Definition A.4), then $(Y, Z)$ is a continuous semimartingale, $Y(t) \in \Gamma$ for every $t \geq 0$ a.s., $\mu \geq \inf\{t \geq 0 \mid Y(t) \notin \mathring{K}\}$ a.s. and $Y(t)$ admits*

$$Y(t) = Y(0) + \int_0^{t\wedge\mu} \partial\Phi(Y(s))\sigma(Y(s))\mathrm{d}Z(s)$$

$$+ \frac{1}{2}\sum_{i,j=1}^D \sum_{k,l=1}^e \int_0^{t\wedge\mu} \partial_{ij}\Phi(Y(s))\sigma(Y(s))_{ik}\sigma(Y(s))_{jl}\mathrm{d}[Z_k, Z_l](s). \tag{22}$$

We note that by Lemma A.15, convergence in distribution under skorohod metric is equivalent to convergence in distribution under uniform metric Definition A.14, therefore in the rest of the paper we will only use the uniform metric in the rest of the paper, e.g., whenever we mention Prohorov metric and $\delta$-Prohorov distance, the underlying metric is the uniform metric.

### B.4 A USER-FRIENDLY INTERFACE FOR KATZENBERGER'S THEOREM

Based on the Lemma B.6, we can immediately apply Theorem B.7 to obtain the following limiting diffusion of SGD.

**Theorem B.8.** *Let the manifold $\Gamma$ and its open neighborhood $U$ satisfy Assumptions 3.1 and 3.2. Let $K \subset U$ be any compact set and fix some $x_0 \in K$. Consider the SGD formulated in Lemma 4.2 where $X_{\eta_n}(0) \equiv x_0$. Define*

$$Y_{\eta_n}(t) = X_{\eta_n}(t) - \phi(X_{\eta_n}(0), A_{\eta_n}(t)) + \Phi(X_{\eta_n}(0))$$

*and $\mu_{\eta_n}(K) = \min\{t \in \mathbb{N} \mid Y_{\eta_n}(t) \notin \mathring{K}\}$. Then the sequence $\{(Y_{\eta_n}^{\mu_{\eta_n}(K)}, Z_{\eta_n}, \mu_{\eta_n}(K))\}_{n\geq 1}$ is relatively compact in $\mathcal{D}_{\mathbb{R}^D \times \mathbb{R}^n}[0, \infty) \times [0, \infty]$. Moreover, if $(Y, Z, \mu)$ is a limit point of this sequence, it holds that $Y(t) \in \Gamma$ a.s for all $t \geq 0$, $\mu \geq \inf\{t \geq 0 \mid Y(t) \notin \mathring{K}\}$ and $Y(t)$ admits*

$$Y(t) = \int_{s=0}^{t\wedge\mu} \partial\Phi(Y(s))\sigma(Y(s))\mathrm{d}W(s) + \int_{s=0}^{t\wedge\mu} \frac{1}{2}\sum_{i,j=1}^D \partial_{ij}\Phi(Y(s))(\sigma(Y(s))\sigma(Y(s))^\top)_{ij}\mathrm{d}s \tag{23}$$

where $\{W(s)\}_{s\geq 0}$ is the standard Brownian motion and $\sigma(\cdot)$ is as defined in Lemma 4.2.

However, the above theorem is hard to parse and cannot be directly applied if we want to further study the implicit bias of SGD through this limiting diffusion. Therefore, we develop a user-friendly interface to it in below. In particular, Theorem 4.6 is the a special case of Theorem B.9. In Theorem 4.6, we replace $\partial\Phi(Y(t))\sigma(Y(t))$ with $\Sigma_\parallel^{\frac{1}{2}}(Y(t))$ to simplify the equation, since $\partial\Phi(Y(t))\sigma(Y(t))(\partial\Phi(Y(t))\sigma(Y(t)))^\top = \Sigma_\parallel(Y(t))$ and thus this change doesn't affect the distribution of the sample paths of the solution.

**Theorem B.9.** *Under the same setting as Theorem B.8, we change the integer index back to $\eta > 0$ with a slight abuse of notation. For any stopping time $\mu$ and stochastic process $\{Y(t)\}_{t\geq 0}$ such that $\mu \geq \inf\{t \geq 0 \mid Y(t) \notin \mathring{K}\}$, $Y(0) = \Phi(x_0)$ and that $(Y, \mu)$ satisfy Equation (23) for some standard Brownian motion $W$. For any compact set $K \subseteq U$ and $T > 0$, define $\mu(K) = \inf\{t \geq 0 \mid Y(t) \notin \mathring{K}\}$ and $\delta = \mathbb{P}(\mu(K) \leq T)$. Then for any $\epsilon > 0$, it holds for all sufficiently small LR $\eta$ that:*

$$\rho^{2\delta}(Y_\eta^{\mu_\eta(K)\wedge T}, Y^{\mu(K)\wedge T}) \leq \epsilon, \tag{24}$$

*which means there is a coupling between the distribution of the stopped processes $Y_\eta^{\mu_\eta(K)\wedge T}$ and $Y^{\mu(K)\wedge T}$, such that the uniform metric between them is smaller than $\epsilon$ with probability at least $1 - 2\delta$. In other words, $\lim_{\eta\to 0} \rho^{2\delta}(Y_\eta^{\mu_\eta(K)\wedge T}, Y^{\mu(K)\wedge T}) = 0$.*

*Moreover, when $\{Y(t)\}_{t\geq 0}$ is a global solution to the following limiting diffusion*

$$Y(t) = \int_{s=0}^t \partial\Phi(Y(s))\sigma(Y(s))\mathrm{d}W(s) + \int_{s=0}^t \frac{1}{2}\sum_{i,j=1}^D \partial_{ij}\Phi(Y(s))(\sigma(Y(s))\sigma(Y(s))^\top)_{ij}\mathrm{d}s$$

*and $Y$ never leaves $U$, i.e. $\mathbb{P}[\forall t \geq 0, Y(t) \in U] = 1$, it holds that $Y_\eta^T$ converges in distribution to $Y^T$ as $\eta \to 0$ for any fixed $T > 0$.*

For clarity, we break the proof of Theorem B.9 into two parts, devoted to the two claims respectively.

*Proof of the first claim of Theorem B.9.* First, Theorem B.8 guarantees there exists a stopping time $\tilde{\mu}$ and a stochastic process $\{\widetilde{Y}(t)\}_{t\geq 0}$ such that

  1. $(\widetilde{Y}, \tilde{\mu})$ satisfies Equation (23);

  2. $\widetilde{Y} \in \Gamma$ *a.s.*;

  3. $\tilde{\mu} \geq \tilde{\mu}(K) := \inf\{t \geq 0 \mid \widetilde{Y}(t) \notin \mathring{K}\}$.

The above conditions imply that $\widetilde{Y}^{\tilde{\mu}(K)} \in \Gamma$ *a.s.*. Since the coefficients in Equation (23) are locally Lipschitz, we claim that $(\widetilde{Y}^{\tilde{\mu}(K)}, \tilde{\mu}(K)) \overset{d}{=} (Y^{\mu(K)}, \mu(K))$. To see this, note that for any compact $K \subseteq U$, the noise function $\sigma, \partial\Phi$ and $\partial^2\Phi$ are all Lipschitz on $K$, thus we can extend their definitions to $\mathbb{R}^D$ such that the resulting functions are still locally Lipschitz. Based on this extension, applying classic theorem on weak uniqueness (e.g., Theorem 1.1.10, Hsu 2002) to the extended version of Equation (23) yields the equivalence in law. Thus we only need to prove the first claim for $\widetilde{Y}$.

Let $\mathcal{E}_T$ be the event such that $\tilde{\mu}(K) > T$ on $\mathcal{E}_T$. Then restricted on $\mathcal{E}_T$, we have $\widetilde{Y}(T \wedge \tilde{\mu}) = \widetilde{Y}(T \wedge \tilde{\mu}(K))$ as $\tilde{\mu} \geq \tilde{\mu}(K)$ holds a.s. We first prove the claim for any convergent subsequence of $\{Y_\eta\}_{\eta > 0}$.

Now, let $\{\eta_m\}_{m\geq 1}$ be a sequence of LRs such that $\eta_m \to 0$ and $Y_{\eta_m}^{\mu_{\eta_m}(K)} \Rightarrow \widetilde{Y}^{\tilde{\mu}}$ as $m \to \infty$. By applying the Skorohod representation theorem, we can put $\{Y_{\eta_m}\}_{m\geq 1}$ and $\widetilde{Y}$ under the same probability space such that $Y_{\eta_m}^{\mu_{\eta_m}(K)} \to \widetilde{Y}^{\tilde{\mu}}$ a.s. in the Skorohod metric, or equivalently the uniform metric (since $\widetilde{Y}^{\tilde{\mu}}$ is continuous) i.e.,

$$d_U(Y_{\eta_m}^{\mu_{\eta_m}(K)}, \widetilde{Y}^{\tilde{\mu}}) \to 0, a.s.,$$

which further implies that for any $\epsilon > 0$, there exists some $N > 0$ such that for all $m > N$,

$$\mathbb{P}\left[d_U(Y_{\eta_m}^{\mu_{\eta_m}(K)\wedge T}, \widetilde{Y}^{\tilde{\mu}\wedge T}) \geq \epsilon\right] \leq \delta.$$

Restricted on $\mathcal{E}_T$, we have $d_U(Y_{\eta_m}^{\mu_{\eta_m}(K)\wedge T}, \widetilde{Y}^{\tilde{\mu}\wedge T}) = d_U(Y_{\eta_m}^{\mu_{\eta_m}(K)\wedge T}, \widetilde{Y}^{\tilde{\mu}(K)\wedge T})$, and it follows that for all $m > N$,

$$\begin{aligned}
\mathbb{P}\left[d_U(Y_{\eta_m}^{\mu_{\eta_m}(K)\wedge T}, \widetilde{Y}^{\tilde{\mu}(K)\wedge T}) \geq \epsilon\right] &\leq \mathbb{P}\left[\{d_U(Y_{\eta_m}^{\mu_{\eta_m}(K)\wedge T}, \widetilde{Y}^{\tilde{\mu}(K)\wedge T}) \geq \epsilon\} \cap \mathcal{E}_T\right] + \mathbb{P}[\mathcal{E}_T^c] \\
&= \mathbb{P}\left[\{d_U(Y_{\eta_m}^{\mu_{\eta_m}(K)\wedge T}, \widetilde{Y}^{\tilde{\mu}\wedge T}) \geq \epsilon\} \cap \mathcal{E}_T\right] + \mathbb{P}[\mathcal{E}_T^c] \\
&\leq \mathbb{P}\left[d_U(Y_{\eta_m}^{\mu_{\eta_m}(K)\wedge T}, \widetilde{Y}^{\tilde{\mu}\wedge T}) \geq \epsilon\right] + \mathbb{P}[\mathcal{E}_T^c] \\
&\leq 2\delta.
\end{aligned}$$

By the definition of the Prohorov metric in Definition A.12, we then get $\rho^{2\delta}(Y_{\eta_m}^{\mu_{\eta_m}(K)\wedge T}, \widetilde{Y}^{\tilde{\mu}(K)\wedge T}) \leq \epsilon$ for all $m > N$. Therefore, we have

$$\lim_{m\to\infty} \rho^{2\delta}(Y_{\eta_m}^{\mu_{\eta_m}(K)\wedge T}, \widetilde{Y}^{\tilde{\mu}(K)\wedge T}) = 0.$$

Now we claim that it indeed holds that $\lim_{\eta\to 0} \rho^{2\delta}(Y_\eta^{\mu_\eta(K)\wedge T}, \widetilde{Y}^{\tilde{\mu}(K)\wedge T}) = 0$. We prove this by contradiction. Suppose otherwise, then there exists some $\epsilon > 0$ such that for all $\eta_0 > 0$, there exists some $\eta < \eta_0$ with $\rho^{2\delta}(Y_\eta^{\mu_\eta(K)\wedge T}, \widetilde{Y}^{\tilde{\mu}(K)\wedge T}) > \epsilon$. Consequently, there is a sequence $\{\eta_m\}_{m\geq 1}$ satisfying $\lim_{m\to\infty} \eta_m = 0$ and $\rho^{2\delta}(Y_{\eta_m}^{\mu_{\eta_m}(K)}, \widetilde{Y}^{\tilde{\mu}(K)\wedge T}) > \epsilon$ for all $m$. Since $\{(Y_{\eta_m}^{\mu_{\eta_m}(K)\wedge T}, Z_{\eta_m}, \mu_{\eta_m}(K))\}_{m\geq 1}$ is relatively compact, there exists a subsequence (WLOG, assume it is the original sequence itself) converging to $(\widetilde{Y}^{\tilde{\mu}\wedge T}, W, \tilde{\mu})$ in distribution. However, repeating the exactly same argument as above, we would have $\rho^{2\delta}(Y_{\eta_m}^{\mu_{\eta_m}(K)\wedge T}, \widetilde{Y}^{\tilde{\mu}(K)\wedge T}) \leq \epsilon$ for all sufficiently large $m$, which is a contradiction. This completes the proof. $\qquad\square$

*Proof of the second claim of Theorem B.9.* We will first show there exists a sequence of compact set $\{K_m\}_{m\geq 1}$ such that $\cup_{m=1}^\infty K_m = U$ and $K_m \subseteq K_{m+1}$. For $m \in \mathbb{N}^+$, we define $H_m = U \setminus (B_{1/m}(0) + \mathbb{R}^D \setminus U)$ and $K_m = \overline{H_m} \cap B_m(0)$. By definition it holds that $\forall m < m', H_m \subseteq H_{m'}$ and $K_m \subseteq K_{m'}$. Moreover, since $K_m$ is bounded and closed, $K_m$ is compact for every $m$. Now we claim $\cup_{m=1}^\infty K_m = U$. Note that $\cup_{m=1}^\infty K_m = \cup_{m=1}^\infty \overline{H_m} \cap B_m(0) = \cup_{m=1}^\infty \overline{H_m}$. $\forall x \in U$, since $U$ is open, we know $d_U(x, \mathbb{R}^D \setminus U) > 0$, thus there exists $m_0 \in \mathbb{N}^+$, such that $\forall m \geq m_0$, $x \notin (B_{1/m}(0) + \mathbb{R}^D \setminus U)$ and thus $x \in H_m$, which implies $x \in \cup_{m=1}^\infty \overline{H_m}$. On the other hand, $\forall x \in \mathbb{R}^D \setminus U$, it holds that $x \in (B_{1/m}(0) + \mathbb{R}^D \setminus U)$ for all $m \in \mathbb{N}^+$, thus $x \notin H_m \subset K_m$.

Therefore, since $Y \in U$ and is continuous almost surely, random variables $\lim_{m\to\infty} \mu(K_m) = \infty$ a.s., which implies $\mu(K_m)$ converges to $\infty$ in distribution, i,e,, $\forall \delta > 0, T > 0, \exists m \in \mathbb{N}^+$, such that $\forall K \supseteq K_m$, it holds $\mathbb{P}[\mu(K) \leq T] \leq \delta$.

Now we will show for any $T > 0$ and $\epsilon > 0$, there exists $\eta_0$ such that $\rho^\epsilon(Y^T, Y_\eta^T) \leq \epsilon$ for all $\eta \leq \eta_0$. Fixing any $T > 0$, for any $\epsilon > 0$, let $\delta = \frac{\epsilon}{4}$, then from above we know exists compact set $K$, such that $\mathbb{P}(\mu(K) \leq T) \leq \delta$. We further pick $K' = K + B_{2\epsilon'}(0)$, where $\epsilon'$ can be any real number satisfying $0 < \epsilon' < \epsilon$ and $K' \subseteq U$. Such $\epsilon'$ exists since $U$ is open. Note $K \subseteq K'$, we have $\mathbb{P}(\mu(K') \leq T) \leq \mathbb{P}(\mu(K) \leq T) \leq \delta$. Thus by the first claim of Theorem B.9, there exists $\eta_0 > 0$, such that for all $\eta \leq \eta_0$, we have $\rho^{2\delta}(Y_\eta^{\mu_\eta(K')\wedge T}, Y^{\mu(K')\wedge T}) \leq 2^{-\lceil T\rceil}\epsilon'$.

Note that $\rho^\delta(Y^{\mu(K)\wedge T}, Y^{\mu(K')\wedge T}) = 0$, so we have for all $\eta \leq \eta_0$,

$$\rho^{3\delta}(Y^{\mu(K)\wedge T}, Y_\eta^{\mu_\eta(K')\wedge T}) \leq 2^{-\lceil T\rceil}\epsilon'.$$

By the definition of $\delta$-Prohorov distance in Definition A.11, we can assume $(Y^{\mu(K)\wedge T}, Y_\eta^{\mu_\eta(K')\wedge T})$ is already the coupling such that $\mathbb{P}\left[d_U(Y^{\mu(K)\wedge T}, Y_\eta^{\mu_\eta(K')\wedge T}) \geq 2^{-\lceil T\rceil}\epsilon'\right] \leq 3\delta$. Below we want to show $\rho^{3\delta}(Y^{\mu(K)\wedge T}, Y_\eta^T) \leq 2^{-\lceil T\rceil}\epsilon'$. Note that for all $t \geq 0$, $Y^{\mu(K)\wedge T}(t) \in K$, thus we know if

$\mu_\eta(K') \leq T$, then

$$d_U(Y^{\mu(K)\wedge T}, Y_\eta^{\mu_\eta(K')\wedge T}) \geq 2^{-\lceil T\rceil} \left\| Y^{\mu(K)\wedge T}(\mu_\eta(K')) - Y_\eta^{\mu_\eta(K')\wedge T}(\mu_\eta(K')) \right\|_2$$
$$\geq 2^{-\lceil T\rceil} d_U(K, \mathbb{R}^d/K')$$
$$\geq 2^{-\lceil T\rceil} \epsilon'.$$

On the other hand, if $\mu_\eta(K') > T$, then $Y_\eta^T = Y_\eta^{\mu_\eta(K')\wedge T}$. Thus we can conclude that $d_U(Y^{\mu(K)\wedge T}, Y_\eta^T) \geq 2^{-\lceil T\rceil}\epsilon'$ implies $d_U(Y^{\mu(K)\wedge T}, Y_\eta^{\mu_\eta(K')\wedge T}) \geq 2^{-\lceil T\rceil}\epsilon'$. Therefore, we further have

$$\mathbb{P}\left[d_U(Y^{\mu(K)\wedge T}, Y_\eta^T) \geq 2^{-\lceil T\rceil}\epsilon'\right] \leq \mathbb{P}\left[d_U(Y^{\mu(K)\wedge T}, Y_\eta^{\mu_\eta(K')\wedge T}) \geq 2^{-\lceil T\rceil}\epsilon'\right] \leq 3\delta,$$

that is,

$$\rho^{3\delta}(Y^{\mu(K)\wedge T}, Y_\eta^T) \leq 2^{-\lceil T\rceil}\epsilon'.$$

Finally, since $\rho^\delta(Y^T, Y^{\mu(K)\wedge T}) = 0$, we have for all $\eta \leq \eta_0$,

$$\rho^\epsilon(Y^T, Y_\eta^T) = \rho^{4\delta}(Y^T, Y_\eta^T) \leq \rho^{3\delta}(Y^{\mu(K)\wedge T}, Y_\eta^T) + \rho^\delta(Y^T, Y^{\mu(K)\wedge T}) \leq 2^{-\lceil T\rceil}\epsilon' + 0 \leq \epsilon,$$

which completes the proof. □

Now, we provide the proof of Theorem 4.6 as a direct application of Theorem B.9.

*Proof of Theorem 4.6.* We first prove that $Y$ never leaves $\Gamma$, i.e., $\mathbb{P}[Y(t) \in \Gamma, \forall t \geq 0] = 1$. By the result of Theorem B.8, we know that for each compact set $K \subset \Gamma$, $Y^{\mu(K)}$ stays on $\Gamma$ almost surely, where $\mu(K) := \inf\{t \geq 0 \mid \widetilde{Y}(t) \notin \mathring{K}\}$ is the earliest time that $Y$ leaves $K$. In other words, for all compact set $K \subset \Gamma$, $\mathbb{P}[\exists t \geq 0, Y(t) \notin \Gamma, Y(t) \in K] = 0$. Let $\{K_m\}_{m\geq 1}$ be any sequence of compact sets such that $\cup_{m\geq 1}K_m = U$ and $K_m \subset U$, e.g., the ones constructed in the proof of the second claim of Theorem B.9. Therefore, we have

$$\mathbb{P}[\exists t \geq 0, Y(t) \notin \Gamma] = \mathbb{P}[\exists t \geq 0, Y(t) \notin \Gamma, Y(t) \in U] \leq \sum_{m=1}^{\infty} \mathbb{P}[\exists t \geq 0, Y(t) \notin \Gamma, Y(t) \in K_m] = 0,$$

which means $Y$ always stays on $\Gamma$.

Then recall the decomposition of $\Sigma = \Sigma_\parallel + \Sigma_\perp + \Sigma_{\parallel,\perp} + \Sigma_{\perp,\parallel}$ as defined in Lemma 4.5. Since $Y$ never leaves $\Gamma$, by Lemma 4.5, we can rewrite Equation (10) as

$$dY(t) = \Sigma_\parallel^{1/2}dW(t) + \partial^2\Phi(Y(t))[\Sigma(Y(t))]dt$$
$$= \partial\Phi(Y(t))\sigma(Y(t))dW(t) + \frac{1}{2}\sum_{i,j=1}^{D} \partial_{ij}\Phi(Y(t))(\sigma(Y(t))\sigma(Y(t))^\top)_{ij}dt$$

where the second equality follows from the definition that $\Sigma_\parallel = \partial\Phi\Sigma\partial\Phi = \partial\Phi\sigma\sigma^\top\partial\Phi$. This coincides with the formulation of the limiting diffusion in Theorem B.9. Therefore, further combining Lemma 4.2 and the second part of Theorem B.9, we obtain the desired result. □

**Remark B.10.** *Our result suggests that for tiny LR $\eta$, SGD dynamics have two phases. In Phase I of $\Theta(1/\eta)$ steps, the SGD iterates move towards the manifold $\Gamma$ of local minimizers along GF. Then in Phase II which is of $\Theta(1/\eta^2)$ steps, the SGD iterates stay close to $\Gamma$ and diffuse approximately according to (10). See Figure 2 for an illustration of this two-phase dynamics. However, since the length of Phase I gets negligible compared to that of Phase II when $\eta \to 0$, Theorem 4.6 only reflects the time scaling of Phase II.*

## C EXPLICIT FORMULA OF THE LIMITING DIFFUSION

In this section, we demonstrate how to compute the derivatives of $\Phi$ by relating to those of the loss function $L$, and then present the explicit formula of the limiting diffusion.

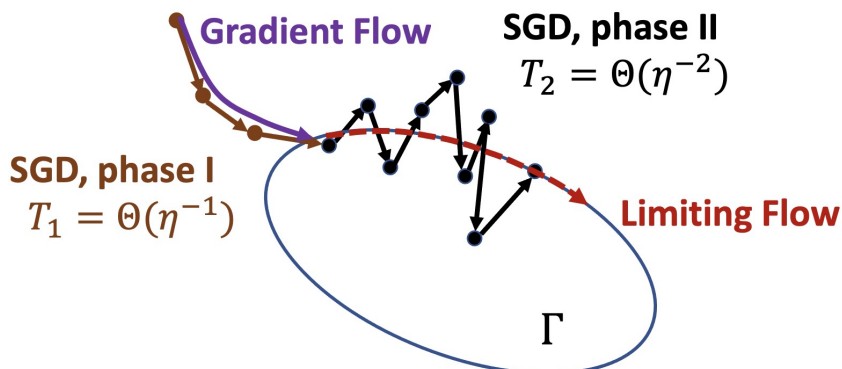

Figure 2: Illustration for two-phase dynamics of SGD with the same example as in Figure 1 . $\Gamma$ is an 1D manifold of minimizers of loss $L$.

## C.1 EXPLICIT EXPRESSION OF THE DERIVATIVES

For any $x \in \Gamma$, we choose an orthonormal basis of $T_x(\Gamma)$ as $\{v_1, \ldots, v_{D-M}\}$. Let $\{v_{D-M+1}, \ldots, v_D\}$ be an orthonormal basis of $T_x^\perp(\Gamma)$ so that $\{v_i\}_{i \in [D]}$ is an orthonormal basis of $\mathbb{R}^D$.

**Lemma C.1.** *For any $x \in \Gamma$ and any $v \in T_x(\Gamma)$, it holds that $\nabla^2 L(x)v = 0$.*

*Proof.* For any $x \in T_x(\Gamma)$, let $\{x(t)\}_{t \geq 0}$ be a parametrized smooth curve on $\Gamma$ such that $x(0) = x$ and $\frac{dx(t)}{dt}\big|_{t=0} = v$. Then $\nabla L(x_t) = 0$ for all $t$. Thus $0 = \frac{d\nabla L(x_t)}{dt}\big|_{t=0} = \nabla^2 L(x)v$. $\qquad\square$

**Lemma C.2.** *For any $x \in \mathbb{R}^D$, it holds that $\partial\Phi(x)\nabla L(x) = 0$ and*

$$\partial^2\Phi(x)[\nabla L(x), \nabla L(x)] = -\partial\Phi(x)\nabla^2 L(x)\nabla L(x).$$

*Proof.* Fixing any $x \in \mathbb{R}^D$, let $\frac{dx(t)}{dt} = -\nabla L(x(t))$ be initialized at $x(0) = x$. Since $\Phi(x(t)) = \Phi(x)$ for all $t \geq 0$, we have

$$\frac{d}{dt}\Phi(x(t)) = -\partial\Phi(x(t))\nabla L(x(t)) = 0.$$

Evaluating the above equation at $t = 0$ yields $\partial\Phi(x)\nabla L(x) = 0$. Moreover, take the second order derivative and we have

$$\frac{d^2}{dt^2}\Phi(x_t) = -\partial^2\Phi(x(t))\left[\frac{dx(t)}{dt}, \nabla L(x(t))\right] - \partial\Phi(x(t))\nabla^2 L(x(t))\frac{dx(t)}{dt} = 0.$$

Evaluating at $t = 0$ completes the proof. $\qquad\square$

Now we can prove Lemma 4.3, restated in below.

**Lemma 4.3.** *For any $x \in \Gamma$, $\partial\Phi(x) \in \mathbb{R}^{D \times D}$ is the projection matrix onto tangent space $T_x(\Gamma)$.*

*Proof of Lemma 4.3.* For any $v \in T_x(\Gamma)$, let $\{v(t), t \geq 0\}$ be a parametrized smooth curve on $\Gamma$ such that $v(0) = x$ and $\frac{dv(t)}{dt}\big|_{t=0} = v$. Since $v(t) \in \Gamma$ for all $t \geq 0$, we have $\Phi(v(t)) = v(t)$, and thus

$$\frac{dv(t)}{dt}\bigg|_{t=0} = \frac{d}{dt}\Phi(v(t))\bigg|_{t=0} = \partial\Phi(x)\frac{dv(t)}{dt}\bigg|_{t=0}.$$

This implies that $\partial\Phi(x)v = v$ for all $v \in T_x(\Gamma)$.

Next, for any $u \in T_x^\perp(\Gamma)$ and $t \geq 0$, consider expanding $\nabla L(x + t\nabla^2 L(x)^\dagger u)$ at $t = 0$:

$$\nabla L\left(x + t\nabla^2 L(x)^\dagger u\right) = \nabla^2 L(x) \cdot t\nabla^2 L(x)^\dagger u + o(t)$$
$$= tu + o(t)$$

where the second equality follows from the assumption that $\nabla^2 L(x)$ is full-rank when restricted on $T_x^\perp(\Gamma)$. Then since $\partial\Phi$ is continuous, it follows that

$$\lim_{t\to 0}\frac{\partial\Phi(x+t\nabla^2 L(x)^\dagger u)\nabla L(x+t\nabla^2 L(x)^\dagger u)}{t} = \lim_{t\to 0}\partial\Phi(x+t\nabla^2 L(x)^\dagger)(u+o(1))$$
$$= \partial\Phi(x)u.$$

By Lemma C.2, we have $\partial\Phi(x+t(\nabla^2 L(x))^\dagger u)\nabla L(x+t(\nabla^2 L(x))^\dagger u) = 0$ for all $t > 0$, which then implies that $\partial\Phi(x)u = 0$ for all $u \in T_x^\perp(\Gamma)$.

Therefore, under the basis $\{v_i, \dots, v_N\}$, $\partial\Phi(x)$ is given by

$$\partial\Phi(x) = \begin{pmatrix} I_{D-M} & 0 \\ 0 & 0 \end{pmatrix} \in \mathbb{R}^{D\times D},$$

that is, the projection matrix onto $T_x(\Gamma)$. $\qquad\square$

**Lemma C.3.** *For any $x \in \Gamma$, it holds that $\partial\Phi(x)\nabla^2 L(x) = 0$.*

*Proof.* It directly follows from Lemma C.1 and Lemma 4.3. $\qquad\square$

Next, we proceed to compute the second-order derivatives.

**Lemma C.4.** *For any $x \in \Gamma$, $u \in \mathbb{R}^D$ and $v \in T_x(\Gamma)$, it holds that*

$$\partial^2\Phi(x)[v,u] = -\partial\Phi(x)\partial^2(\nabla L)(x)[v,\nabla^2 L(x)^\dagger u] - \nabla^2 L(x)^\dagger\partial^2(\nabla L)(x)[v,\partial\Phi(x)u].$$

*Proof of Lemma C.4.* Consider a parametrized smooth curve $\{v(t)\}_{t\geq 0}$ on $\Gamma$ such that $v(0) = x$ and $\frac{dv(t)}{dt}\big|_{t=0} = v$. We define $P(t) = \partial\Phi(v(t))$, $P^\perp(t) = I_D - P(t)$ and $H(t) = \nabla^2 L(v(t))$ for all $t \geq 0$. By Lemma C.1 and 4.3, we have

$$P^\perp(t)H(t) = H(t)P^\perp(t) = H(t), \tag{25}$$

Denote the derivative of $P(t)$, $P^\perp(t)$ and $H(t)$ with respect to $t$ as $P'(t)$, $(P^\perp)'(t)$ and $H'(t)$. Then differentiating with respect to $t$, we have

$$(P^\perp)'(t)H(t) = H'(t) - P^\perp(t)H'(t) = P(t)H'(t). \tag{26}$$

Then combining (25) and (26) and evaluating at $t = 0$, we have

$$P'(0)H(0) = -(P^\perp)'(0)H(0) = -P(0)H'(0) \tag{27}$$

We can decompose $P'(0)$ and $H(0)$ as follows

$$P'(0) = \begin{pmatrix} P'_{11}(0) & P'_{12}(0) \\ P'_{21}(0) & P'_{22}(0) \end{pmatrix}, \quad H(0) = \begin{pmatrix} 0 & 0 \\ 0 & H_{22}(0) \end{pmatrix}, \tag{28}$$

where $P'_{11}(0) \in \mathbb{R}^{(D-M)\times(D-M)}$ and $H_{22}$ is the hessian of $L$ restricted on $T_x^\perp(\Gamma)$. Also note that

$$P(0)H'(0)P^\perp(0) = \begin{pmatrix} I_{D-M} & 0 \\ 0 & 0 \end{pmatrix}\begin{pmatrix} H'_{11}(0) & H'_{12}(0) \\ H'_{21}(0) & H'_{22}(0) \end{pmatrix}\begin{pmatrix} 0 & 0 \\ 0 & I_M \end{pmatrix}$$
$$= \begin{pmatrix} 0 & H'_{12}(0) \\ 0 & 0 \end{pmatrix},$$

and thus by (28) we have

$$P'(0)H(0) = \begin{pmatrix} 0 & P'_{12}(0)H_{22}(0) \\ 0 & P'_{22}(0)H_{22}(0) \end{pmatrix} = \begin{pmatrix} 0 & -H'_{12}(0) \\ 0 & 0 \end{pmatrix}.$$

This implies that we must have $P'_{22}(0) = 0$ and $P'_{12}(0)H_{22}(0) = H'_{12}(0)$. Similarly, by taking transpose in (28), we also have $H_{22}(0)P'_{21}(0) = -H'_{21}(0)$.

It then remains to determine the value of $P'_{11}(0)$. Note that since $P(t)P(t) = P(t)$, we have $P'(t)P(t) + P(t)P'(t) = P'(t)$, evaluating at $t = 0$ yields

$$2P'_{11}(0) = P'_{11}(0).$$

Therefore, we must have $P'_{11}(0) = 0$. Combining the above results, we obtain

$$P'(0) = -P(0)H'(0)H(0)^\dagger - H(0)^\dagger H'(0)P(0).$$

Finally, recall that $P(t) = \partial\Phi(v(t))$, and thus

$$P'(0) = \frac{\mathrm{d}}{\mathrm{d}t}\partial\Phi(v(t))\Big|_{t=0} = \partial^2\Phi(x)[v].$$

Similarly, we have $H'(0) = \partial^2(\nabla L)(x)[v]$, and it follows that

$$\partial^2\Phi(x)[v] = -\partial\Phi(x)\partial^2(\nabla L)(x)[v]\nabla^2 L(x)^\dagger - \nabla^2 L(x)^\dagger \partial^2(\nabla L)(x)[v]\partial\Phi(x).$$

$\square$

**Lemma C.5.** *For any $x \in \Gamma$ and $u \in T_x^\perp(\Gamma)$, it holds that*

$$\partial^2\Phi(x)[uu^\top + \nabla^2 L(x)^\dagger uu^\top \nabla^2 L(x)] = -\partial\Phi(x)\partial^2(\nabla L)(x)[\nabla^2 L(x)^\dagger uu^\top].$$

*Proof of Lemma C.5.* For any $u \in T_x^\perp(\Gamma)$, we define $u(t) = x + t\nabla^2 L(x)^\dagger u$ for $t \geq 0$. By Taylor approximation, we have

$$\nabla L(u(t)) = t\nabla^2 L(x)\nabla^2 L(x)^\dagger u + o(t) = tu + o(t) \tag{29}$$

and

$$\nabla^2 L(u(t)) = \nabla^2 L(x) + t\partial^2(\nabla L)(x)[\nabla^2 L(x)^\dagger u] + o(t). \tag{30}$$

Combine (29) and (30) and apply Lemma C.2, and it follows that

$$\begin{aligned}
0 &= \partial^2\Phi(u(t))[\nabla L(u(t)), \nabla L(u(t))] + \partial\Phi(u(t))\nabla^2 L(u(t))\nabla L(u(t)) \\
&= t^2\partial^2\Phi(u(t))[u + o(1)](u + o(1)) + t^2\partial\Phi(u(t))\partial^2(\nabla L)(x)[\nabla^2 L(x)^\dagger u](u + o(1)) \\
&\quad + t^2\frac{\partial\Phi(u(t))}{t}\nabla^2 L(x)(u + o(1)) \\
&= t^2\partial^2\Phi(u(t))[u + o(1)](u + o(1)) + t^2\partial\Phi(u(t))\partial^2(\nabla L)(x)[\nabla^2 L(x)^\dagger u](u + o(1)) \\
&\quad + t^2\frac{\partial\Phi(u(t)) - \partial\Phi(x)}{t}\nabla^2 L(x)(u + o(1))
\end{aligned}$$

where the last equality follows from Lemma C.3. Dividing both sides by $t^2$ and letting $t \to 0$, we get

$$\partial^2\Phi(x)[u]u + \partial\Phi(x)\partial^2(\nabla L)(x)[\nabla^2 L(x)^\dagger u]u + \partial^2\Phi(x)[\nabla^2 L(x)^\dagger u]\nabla^2 L(x)u = 0.$$

Rearranging the above equation completes the proof. $\square$

With the notion of Lyapunov Operator in Definition 4.4, Lemma C.5 can be further simplified into Lemma C.6.

**Lemma C.6.** *For any $x \in \Gamma$ and $\Sigma \in \mathrm{span}\{uu^\top \mid u \in T_x^\perp(\Gamma)\}$,*

$$\langle \partial^2\Phi(x), \Sigma \rangle = -\partial\Phi(x)\partial^2(\nabla L)(x)[\mathcal{L}_{\nabla^2 L(x)}^{-1}(\Sigma)]. \tag{31}$$

*Proof of Lemma C.6.* Let $A = uu^\top + \nabla^2 L(x)^\dagger uu^\top \nabla^2 L(x)$ and $B = \nabla^2 L(x)^\dagger uu^\top$. The key observation is that $A + A^\top = \mathcal{L}_{\nabla^2 L(x)}(B + B^\top)$. Therefore, by Lemma C.5, it holds that

$$\partial^2\Phi(x)[\mathcal{L}_{\nabla^2 L(x)}(B+B^\top)] = \partial^2\Phi(x)[A+A^\top] = 2\partial\Phi(x)\partial^2(\nabla L)(x)[B] = \partial\Phi(x)\partial^2(\nabla L)(x)[B+B^\top].$$

Since $\nabla^2 L(x)^\dagger$ is full-rank when restricted to $T_x^\perp(\Gamma)$, we have $\mathrm{span}\{\nabla^2 L(x)^\dagger uu^\top + uu^\top \nabla^2 L(x)^\dagger \mid u \in T_x^\perp(\Gamma)\} = \mathrm{span}\{uu^\top \mid u \in T_x^\perp(\Gamma)\}$. Thus by the linearity of above equation, we can replace $B + B^\top$ by any $\Sigma \in \mathrm{span}\{uu^\top \mid u \in T_x^\perp(\Gamma)\}$, resulting in the desired equation. $\square$

Then Lemma 4.5 directly follows from Lemma C.4 and C.5.

## C.2 Tangent Noise Compensation only Dependends on the Manifold Itself

Here we show that the second term of (10), i.e., the **tangent noise compensation** for the limiting dynamics to stay on $\Gamma$, only depends on $\Gamma$ itself.

**Lemma C.7.** *For any $x \in \Gamma$, suppose there exist a neighborhood $U_x$ of $x$ and two loss functions $L$ and $L'$ that define the same manifold $\Gamma$ locally in $U_x$, i.e., $\Gamma \cap U_x = \{x \mid \nabla L(x) = 0\} = \{x \mid \nabla L'(x) = 0\}$. Then for any $v \in T_x(\Gamma)$, it holds that $(\nabla^2 L(x))^\dagger \partial^2 (\nabla L)(x) [v, v] = (\nabla^2 L'(x))^\dagger \partial^2 (\nabla L')(x) [v, v]$.*

*Proof of Lemma C.7.* Let $\{v(t)\}_{t \geq 0}$ be a smooth curve on $\Gamma$ with $v(0) = x$ and $\frac{\mathrm{d}v(t)}{\mathrm{d}t}\big|_{t=0} = v$. Since $v(t)$ stays on $\Gamma$, we have $\nabla L(v(t)) = 0$ for all $t \geq 0$. Taking derivative for two times yields $\partial^2(\nabla L)(v(t))[\frac{\mathrm{d}v(t)}{\mathrm{d}t}, \frac{\mathrm{d}v(t)}{\mathrm{d}t}] + \nabla^2 L(v(t)) \frac{\mathrm{d}^2 v(t)}{\mathrm{d}t^2} = 0$. Evaluating it at $t = 0$ and multiplying both sides by $\nabla^2 L(x)^\dagger$, we get

$$\nabla^2 L(x)^\dagger \partial^2(\nabla L)(x)[v, v] = -\nabla^2 L(x)^\dagger \nabla^2 L(x) \frac{\mathrm{d}^2 v(t)}{\mathrm{d}t^2}\bigg|_{t=0} = -\partial \Phi(x) \frac{\mathrm{d}^2 v(t)}{\mathrm{d}t^2}\bigg|_{t=0}.$$

Since $\partial \Phi(x)$ is the projection matrix onto $T_x(\Gamma)$ by Lemma 4.3, it does not depend on $L$, so analogously we also have $\nabla^2 L'(x)^\dagger \partial^2(\nabla L')(x)[v, v] = -\partial \Phi(x) \frac{\mathrm{d}^2 v(t)}{\mathrm{d}t^2}\big|_{t=0}$ as well. The proof is thus completed. Note that $\partial \Phi(x) \frac{\mathrm{d}^2 v(t)}{\mathrm{d}t^2}\big|_{t=0}$ is indeed the second fundamental form for $v$ at $x$, and the value won't change if we choose another parametric smooth curve with a different second-order time derivative. (See Chapter 6 in Do Carmo (2013) for a reference.) $\square$

## C.3 Proof of results in Section 5

Now we are ready to give the missing proofs in Section 5 which yield explicit formula of the limiting diffusion for label noise and isotropic noise.

**Corollary 5.1** (Limiting Diffusion for Isotropic Noise). *If $\Sigma \equiv I_D$ on $\Gamma$, SDE (10) is then*

$$\mathrm{d}Y(t) = \underbrace{\partial \Phi(Y)\mathrm{d}W + \frac{1}{2}\nabla^2 L(Y)^\dagger \partial^2(\nabla L)(Y)[\partial \Phi(Y)]\,\mathrm{d}t}_{\text{Brownian Motion on Manifold}} - \underbrace{\frac{1}{2}\partial \Phi(Y)\nabla(\ln |\nabla^2 L(Y)|_+)\mathrm{d}t}_{\text{Normal Regularization}} \quad (11)$$

*where $|\nabla^2 L(Y)|_+ = \lim_{\alpha \to 0} \frac{|\nabla^2 L(Y) + \alpha I_D|}{\alpha^{D - \mathrm{rank}(\nabla^2 L(Y))}}$ is the pseudo-determinant of $\nabla^2 L(Y)$. $|\nabla^2 L(Y)|_+$ is also equal to the sum of log of non-zero eigenvalue values of $\nabla^2 L(Y)$.*

*Proof of Corollary 5.1.* Set $\Sigma_\parallel = \partial \Phi$, $\Sigma_\perp = I_D - \partial \Phi$ and $\Sigma_{\perp,\parallel} = \Sigma_{\parallel,\perp} = 0$ in the decomposition of $\Sigma$ by Lemma 4.5, and we need to show $\partial \Phi \nabla(\ln |\Sigma|_+) = \partial^2(\nabla L)[(\nabla^2 L)^\dagger]$.

Holbrook (2018) shows that the gradient of pseudo-inverse determinant satisfies $\nabla |A|_+ = |A|_+ A^\dagger$. Thus we have for any vector $v \in \mathbb{R}^D$, $\langle v, \nabla \ln |\nabla^2 L|_+ \rangle = \left\langle \frac{|\nabla^2 L|_+ \nabla^2 L}{|\nabla^2 L|_+}, \partial^2(\nabla L)[v] \right\rangle = \langle \nabla^2 L, \partial^2(\nabla L)[v] \rangle = \partial^2(\nabla L)[v, \nabla^2 L] = \langle v, \partial^2(\nabla L)[(\nabla^2 L)^\dagger] \rangle$, which completes the proof. $\square$

**Corollary 5.2** (Limiting Flow for Label Noise). *If $\Sigma \equiv c\nabla^2 L$ on $\Gamma$ for some constant $c > 0$, SDE (10) can be simplified into (13) where the regularization is from the noise in the normal space.*

$$\mathrm{d}Y(t) = -1/4 \cdot \partial \Phi(Y(t))\nabla \mathrm{tr}[c\nabla^2 L(Y(t))]\mathrm{d}t. \quad (13)$$

*Proof of Corollary 5.2.* Since $\Sigma = c\nabla^2 L$, here we have $\Sigma_\perp = \Sigma$ and $\Sigma_\parallel, \Sigma_{\perp,\parallel}, \Sigma_{\parallel,\perp} = 0$. Thus it suffices to show that $2\partial^2(\nabla L)\left[\mathcal{L}_{\nabla^2 L}^{-1}(\Sigma_\perp)\right] = \nabla \mathrm{tr}[\nabla^2 L]$. Note that for any $v \in \mathbb{R}^D$,

$$v^\top \nabla \mathrm{tr}[\nabla^2 L] = \langle I_D, \partial^2(\nabla L)[v] \rangle = \langle I_D - \partial \Phi, \partial^2(\nabla L)[v] \rangle, \quad (32)$$

where the second equality is because the the tangent space of symmetric rank-$n$ matrices at $\nabla^2 L$ is $\{A\nabla^2 L + \nabla^2 L A^\top \mid A \in \mathbb{R}^{D \times D}\}$, and every element in this tangent space has zero inner-product with $\partial \Phi$ by Lemma 4.3. Also note that $\mathcal{L}_{\nabla^2 L}^{-1}(\nabla^2 L) = \frac{1}{2}(I_D - \partial \Phi)$, thus $\langle I_D - \partial \Phi, \partial^2(\nabla L)[v] \rangle = 2\langle \mathcal{L}_{\nabla^2 L}^{-1}(\nabla^2 L), \partial^2(\nabla L)[v] \rangle = 2v^\top \partial^2(\nabla L)[\mathcal{L}_{\nabla^2 L}^{-1}(\nabla^2 L)]$. $\square$

### C.4    EXAMPLE: $k$-PHASE MOTOR

We also give an example with rigorous proof where the implicit bias induced by noise in the normal space cannot be characterized by a fixed regularizer, which was first discovered by Damian et al. (2021) but was only verified via experiments.

Note the normal regularization in both cases of label noise and isotropic noise induces Riemmanian gradient flow against some regularizer, it's natural to wonder if the limiting flow induced by the normal noise can always be characterized by certain regularizer. Interestingly, Damian et al. (2021) answers this question negatively via experiments in their Section E.2. We adapt their example into the following one, and rigorously prove the limiting flow moves around a cycle at a constant speed and never stops using our framework.

Suppose dimension $D = k + 2 \geq 5$. For each $x \in \mathbb{R}^D$, we decompose $x = \begin{pmatrix} x_{1:2} \\ x_{3:D} \end{pmatrix}$ where $x_{1:2} \in \mathbb{R}^2$ and $x_{3:D} \in \mathbb{R}^{D-2}$. Let $Q_\theta \in \mathbb{R}^{2 \times 2}$ be the rotation matrix of angle $\theta$, i.e., $Q_\theta = \begin{pmatrix} \cos\theta & -\sin\theta \\ \sin\theta & \cos\theta \end{pmatrix}$ and the loss $L(x) := \frac{1}{8}(\|x_{1:2}\|_2^2 - 1)^2 + \frac{1}{2}\sum_{j=3}^{D}(2 + \langle Q_\alpha^{j-3}v, x_{1:2}\rangle)x_j^2$, where $\alpha = \frac{2\pi}{D-2}$ and $v$ is any vector in $\mathbb{R}^2$ with unit norm. Here the manifold is given by $\Gamma := \{x \mid L(x) = 0\} = \{x \in \mathbb{R}^D \mid x_1^2 + x_2^2 = 1, x_j = 0, \forall j = 3, \ldots, D\}$.

The basic idea is that we can add noise in the 'auxiliary dimensions' for $j = 3, \ldots, D$ to get the regularization force on the circle $\{x_1^2 + x_2^2 = 1\}$, and the goal is to make the vector field induced by the normal regularization always point to the same direction, say anti-clockwise. However, this cannot be done with a single auxiliary dimension because from the analysis for label noise, we know when $\mathcal{L}_{\nabla^2 L}^{-1}(\Sigma_\perp)$ is identity, the *normal regularization* term in Equation (10) has 0 path integral along the unit circle and thus it must have both directions. The key observation here is that we can align the magnitude of noise with the strength of the regularization to make the path integral positive. By using $k \geq 3$ auxiliary dimensions, we can further ensure the normal regularization force is anti-clockwise and of constant magnitude, which is reminiscent of how a three-phase induction motor works.

**Lemma C.8.** *Let $\Sigma \in \mathbb{R}^{D \times D}$ be given by $\Sigma_{ij}(x) = (1 + \langle Q_\alpha^{j-3}v, Q_{-\pi/2}x_{1:2}\rangle)(2 + \langle Q_\alpha^{j-3}v, x_{1:2}\rangle)$, if $i = j \geq 3$ or 0 otherwise, then the solution of SDE (10) is the following (33), which implies that $Y(t)$ moves anti-clockwise with a constant angular speed of $(D-2)/2$.*

$$Y_{1:2}(t) = Q_{t(D-2)/2}Y_{1:2}(0) \quad and \quad Y_{3:D}(t) \equiv 0. \tag{33}$$

*Proof of Lemma C.8.* Note that for any $x \in \Gamma$, it holds that

$$\left(\nabla^2 L(x)\right)_{ij} = \begin{cases} 2 + \langle Q_\alpha^{j-3}v, x_{1:2}\rangle & \text{if } i = j \geq 3, \\ x_i x_j & \text{if } i, j \in \{1, 2\}, \\ 0 & \text{otherwise.} \end{cases} \tag{34}$$

Then clearly $\Sigma$ only brings about noise in the normal space, and specifically, it holds that $\mathcal{L}_{\nabla^2 L(x)}^{-1}(\Sigma(x)) = \text{diag}(0, 0, 1 + \langle Q_\alpha^0 v, Q_{-\pi/2}x_{1:2}\rangle, \ldots, 1 + \langle Q_\alpha^{D-3}v, Q_{-\pi/2}x_{1:2}\rangle)$. Further note that, by the special structure of the hessian in (34) and Lemma C.3, for any $x \in \Gamma$, we have $\partial\Phi(x) = (x_2, -x_1, 0, \ldots, 0)^\top(x_2, -x_1, 0, \ldots, 0) = \begin{pmatrix} Q_{-\pi/2}x_{1:2} \\ 0 \end{pmatrix}\begin{pmatrix} Q_{-\pi/2}x_{1:2} \\ 0 \end{pmatrix}^\top$. Combining these

facts, the dynamics of the first two coordinates in SDE (10) can be simplified into

$$
\begin{aligned}
\frac{\mathrm{d}x_{1:2}(t)}{\mathrm{d}t} &= -\left(\frac{1}{2}\partial\Phi(x(t))\partial^2(\nabla L)(x(t))[\mathcal{L}_{\nabla^2 L}^{-1}(\Sigma(x(t)))]\right)_{1:2} \\
&= -\frac{1}{2}Q_{-\pi/2}x_{1:2}x_{1:2}^\top Q_{-\pi/2}^\top \sum_{j=3}^{D}\left(1+\left\langle Q_\alpha^{j-3}v, Q_{-\pi/2}x_{1:2}\right\rangle\right)\nabla_{1:2}(\partial_{jj}L)(x) \\
&= -\frac{1}{2}Q_{-\pi/2}x_{1:2}\left\langle Q_{-\pi/2}x_{1:2}, \sum_{j=3}^{D}\left(1+\left\langle Q_\alpha^{j-3}v, Q_{-\pi/2}x_{1:2}\right\rangle\right)Q_\alpha^{j-3}v\right\rangle \\
&= -\frac{1}{2}Q_{-\pi/2}x_{1:2}\left(\left\langle Q_{-\pi/2}x_{1:2}, \sum_{j=3}^{D}Q_\alpha^{j-3}v\right\rangle + \sum_{j=3}^{D}\left\langle Q_\alpha^{j-3}v, Q_{-\pi/2}x_{1:2}\right\rangle^2\right) \\
&= -\frac{1}{2}Q_{-\pi/2}x_{1:2}\left(0 + \frac{D-2}{2}\left\|Q_{-\pi/2}x_{1:2}\right\|_2^2\right) = \frac{D-2}{2}Q_{\pi/2}x_{1:2},
\end{aligned}
$$

where the second to the last equality follows from the property of $Q_\alpha$ and the last equality follows from the fact that $\|x_{1:2}\|_2^2 = 1$ for all $x \in \Gamma$. Note we require $k \geq 3$ (or $D \geq 5$) to allow $\sum_{j=3}^{D}\left\langle Q_\alpha^{j-3}v, Q_{-\pi/2}x_{1:2}\right\rangle^2 = \frac{D-2}{2}\left\|Q_{-\pi/2}x_{1:2}\right\|_2^2$. On the other hand, we have $\frac{\mathrm{d}x_{3:D}(t)}{\mathrm{d}t} = 0$ as $\partial\Phi$ kills the movement on that component.

The proof is completed by noting that the solution of $x_{1:2}$ is

$$
x_{1:2}(t) = \exp\left(t \cdot \frac{D-2}{2}Q_{\pi/2}\right)x_{1:2}(0),
$$

and by Lemma C.9,

$$
\exp\left(t \cdot \frac{D-2}{2}Q_{\pi/2}\right) = (\exp(Q_{\pi/2}))^{\frac{t(D-2)}{2}} = Q_1^{\frac{t(D-2)}{2}} = Q_{\frac{t(D-2)}{2}}.
$$

$\square$

**Lemma C.9.** $\exp\left(\begin{pmatrix} 0 & -1 \\ 1 & 0 \end{pmatrix}\right) = \begin{pmatrix} \cos 1 & -\sin 1 \\ \sin 1 & \cos 1 \end{pmatrix}$.

*Proof.* By definition, for matrix $A = \begin{pmatrix} 0 & -1 \\ 1 & 0 \end{pmatrix}$, $\exp(A) = \sum_{t=0}^{\infty}\frac{A^t}{t!}$. Note $A_2 = -I$, $A^3 = -A$ and $A^4 = I$, and by using this pattern, we can easily check that

$$
\sum_{t=0}^{\infty}\frac{A^t}{t!} = \begin{pmatrix} \sum_{i=0}^{\infty}(-1)^i\frac{1}{(2i)!} & -\sum_{i=0}^{\infty}(-1)^i\frac{1}{(2i+1)!} \\ \sum_{i=0}^{\infty}(-1)^i\frac{1}{(2i+1)!} & \sum_{i=0}^{\infty}(-1)^i\frac{1}{(2i)!} \end{pmatrix} = \begin{pmatrix} \cos 1 & -\sin 1 \\ \sin 1 & \cos 1 \end{pmatrix}.
$$

$\square$

# D    PROOF OF RESULTS IN SECTION 6

In this section, we present the missing proofs in Section 6 regarding the overparametrized linear model.

For convenience, for any $p, r \geq 0$ and $u \in \mathbb{R}^D$, we denote by $B_r^p(u)$ the $\ell_p$ norm ball of radius $r$ centered at $u$. We also denote $v_{i:j} = (v_i, v_{i+1}, \ldots, v_j)^\top$ for $i, j \in [D]$.

## D.1    PROOF OF THEOREM 6.1

In this subsection, we provide the proof of Theorem 6.1.

**Theorem 6.1.** *In the setting of OLM, suppose the groundtruth is $\kappa$-sparse and $n \geq \Omega(\kappa \ln d)$ training data are sampled from either i.i.d. Gaussian or Boolean distribution. Then for any initialization $x_{init}$ (except a zero-measure set) and any $\epsilon > 0$, there exist $\eta_0, T > 0$ such that for any $\eta < \eta_0$, OLM trained with label noise SGD (12) with LR equal to $\eta$ for $\lfloor T/\eta^2 \rfloor$ steps returns an $\epsilon$-optimal solution, with probability of $1 - e^{-\Omega(n)}$ over the randomness of the training dataset.*

*Proof of Theorem 6.1.* First, by Lemma 6.6, it holds with probability at least $1 - e^{-\Omega(n)}$ that the solution to (18), $x_*$, is unique up to and satisfies $|x_*| = \psi(w_*)$. Then on this event, for any $\epsilon > 0$, by Lemma 6.5, there exists some $T > 0$ such that $x_T$ given by the Riemannian gradient flow (17) satisfies that $x_T$ is an $\epsilon/2$-optimal solution of the OLM. For this $T$, by Theorem 4.6, we know that the $\lfloor T/\eta^2 \rfloor$-th SGD iterate, $x_\eta(\lfloor T/\eta^2 \rfloor)$, satisfies $\|x_\eta(\lfloor T/\eta^2 \rfloor) - x_T\|_2 \leq \epsilon/2$ with probability at least $1 - e^{-\Omega(n)}$ for all sufficiently small $\eta > 0$, and thus $x_\eta(\lfloor T/\eta^2 \rfloor)$ is an $\epsilon$-optimal solution of the OLM. Finally, the validity of applying Theorem 4.6 is guaranteed by Lemma 6.2 and 6.3. This completes the proof. $\square$

In the following subsections, we provide the proofs of all the components used in the above proof.

## D.2 PROOF OF LEMMA 6.2

Recall that for each $i \in [n]$ $f_i(x) = f(u,v) = z_i^\top(u^{\odot 2} - v^{\odot 2})$, $\nabla f_i(x) = 2\binom{z_i \odot u}{z_i \odot v}$, and $K(x) = (K_{ij}(x))_{i,j \in [n]}$ where each $K_{ij}(x) = \langle \nabla f_i(x), \nabla f_j(x) \rangle$. Then

$$\nabla^2 \ell_i(x) = 2 \begin{pmatrix} z_i \odot u \\ -z_i \odot v \end{pmatrix} \left( (z_i \odot u)^\top \quad -(z_i \odot v)^\top \right) + (f_i(u,v) - y_i) \cdot \mathrm{diag}(z_i, z_i).$$

So for any $x \in \Gamma$, it holds that

$$\nabla^2 L(x) = \frac{2}{n} \sum_{i=1}^n \begin{pmatrix} z_i \odot u \\ -z_i \odot v \end{pmatrix} \left( (z_i \odot u)^\top \quad -(z_i \odot v)^\top \right). \tag{35}$$

**Lemma D.1.** *For any fixed $x \in \mathbb{R}^D$, suppose $\{\nabla f_i(x)\}_{i \in [n]}$ is linearly independent, then $K(x)$ is full-rank.*

*Proof of Lemma D.1.* Suppose otherwise, then there exists some $\lambda \in \mathbb{R}^n$ such that $\lambda \neq 0$ and $\lambda^\top K(x)\lambda = 0$. However, note that

$$\lambda^\top K(x)\lambda = \sum_{i,j=1}^n \lambda_i \lambda_j K_{ij}(x)$$

$$= \sum_{i,j=1}^n \lambda_i \lambda_j \langle \nabla f_i(x), \nabla f_j(x) \rangle$$

$$= \left\| \sum_{i=1}^n \lambda_i \nabla f_i(x) \right\|_2^2,$$

which implies that $\sum_{i=1}^n \lambda_i \nabla f_i(x) = 0$. This is a contradiction since by assumption $\{\nabla f_i(x)\}_{i \in [n]}$ is linearly independent. $\square$

**Lemma 6.2.** *Consider the loss $L$ defined in (14) and manifold $\Gamma$ defined in (15). If data is full rank, i.e., $\mathrm{rank}(Z) = n$, then it holds that (a). $\Gamma$ is a smooth manifold of dimension $D - n$; (b). $\mathrm{rank}(\nabla^2 L(x)) = n$ for all $x \in \Gamma$. In particular, $\mathrm{rank}(Z) = n$ holds with probability 1 for Gaussian distribution and with probability $1 - c^d$ for Boolean distribution for some constant $c \in (0,1)$.*

*Proof of Lemma 6.2.* (1) By preimage theorem (Banyaga & Hurtubise, 2013), it suffices to check the jacobian $[\nabla f_1(x), \ldots, \nabla f_n(x)] = 2[\binom{z_1 \odot u}{-z_1 \odot v}, \ldots, \binom{z_n \odot u}{-z_n \odot v}]$ is full rank. Similarly, for the second claim, due to (35). it is also equivalent to show that $\{\binom{z_i \odot u}{-z_i \odot v}\}_{i \in [n]}$ is of rank $n$.

Since $\binom{u}{v} \in \Gamma \subset U$, each coordinate is non-zero, thus we only need to show that $\{z_i\}_{i \in [n]}$ is of rank $n$. This happens with probability 1 in the Gaussian case, and probability at least $1 - c^d$ for some constant $c \in (0,1)$ by Kahn et al. (1995). This completes the proof. $\square$

### D.3 PROOF OF LEMMA 6.3

We first establish some auxiliary results. The following lemma shows the PL condition along the trajectory of gradient flow.

**Lemma D.2.** *Along the gradient flow generated by* $-\nabla L$*, it holds that* $\|\nabla L(x(t))\|^2 \geq \frac{16}{n}\lambda_{\min}(ZZ^\top) \cdot \min_{i \in [d]}|u_i(0)v_i(0)|L(x(t)), \forall t \geq 0$.

To prove Lemma D.2, we need the following invariance along the gradient flow.

**Lemma D.3.** *Along the gradient flow generated by* $-\nabla L$*,* $u_j(t)v_j(t)$ *stays constant for all* $j \in [d]$*. Thus,* $\text{sign}(u_j(t)) = \text{sign}(u_j(0))$ *and* $\text{sign}(v_j(t)) = \text{sign}(v_j(0))$ *for any* $j \in [d]$*.*

*Proof of Lemma D.3.*

$$
\begin{aligned}
\frac{\partial}{\partial t}(u_j(t)v_j(t)) &= \frac{\partial u_j(t)}{\partial t} \cdot v_j(t) + u_j(t) \cdot \frac{\partial v_j(t)}{\partial t} \\
&= \nabla_u L(u(t), v(t))_j \cdot v_j(t) + u_j(t) \cdot \nabla_v L(u(t), v(t))_j \\
&= \frac{2}{n}\sum_{i=1}^n (f_i(u(t), v(t)) - y_i)z_{i,j}u_j(t)v_j(t) - \frac{2u_j(t)}{n}\sum_{i=1}^n (f_i(u(t), v(t)) - y_i)z_{i,j}v_j(t) \\
&= 0.
\end{aligned}
$$

Therefore, any sign change of $u_j(t), v_j(t)$ would enforce $u_j(t) = 0$ or $v_j(t) = 0$ for some $t > 0$ since $u_j(t), v_j(t)$ are continuous in time $t$. This immediately leads to a contradiction to the invariance of $u_j(t)v_j(t)$. $\square$

We then can prove Lemma D.2.

*Proof of Lemma D.2.* Note that

$$
\begin{aligned}
\|\nabla L(x)\|_2^2 &= \frac{1}{n^2}\sum_{i,j=1}^n (f_i(x) - y_i)(f_j(x) - y_j)\langle \nabla f_i(x), \nabla f_j(x)\rangle \\
&\geq \frac{1}{n^2}\sum_{i=1}^n (f_i(x) - y_i)^2 \lambda_{\min}(K(x)) \\
&= \frac{2}{n}L(x)\lambda_{\min}(K(x)),
\end{aligned}
$$

where $K(x)$ is a $n \times n$ p.s.d. matrix with $K_{ij}(x) = \langle \nabla f_i(x), \nabla f_j(x)\rangle$. Below we lower bound $\lambda_{\min}(K(x))$, the smallest eigenvalue of $K(x)$. Note that $K_{ij}(x(t)) = 4\sum_{h=1}^d z_{i,h}z_{j,h}((u_h(t))^2 + (v_h(t))^2)$, and we have

$$
K(x(t)) = 4Z\text{diag}((u(t))^{\odot 2} + (v(t))^{\odot 2})Z^\top \succeq 8Z\text{diag}(|u(t) \odot v(t)|)Z^\top
$$

$$
\overset{(*)}{=} 8Z\text{diag}(|u(0) \odot v(0)|)Z^\top \succeq 8\min_{i \in [d]}|u_i(0)v_i(0)|ZZ^T
$$

where $(*)$ is by Lemma D.3. Thus $\lambda_{\min}(K(x(t)) \geq 8\min_{i \in [d]}|u_i(0)v_i(0)|\lambda_{\min}(ZZ^T)$ for all $t \geq 0$, which completes the proof. $\square$

We also need the following characterization of the manifold $\Gamma$.

**Lemma D.4.** *All the stationary points in* $U$ *are global minimizers, i.e.,* $\Gamma = \{x \in U \mid \nabla L(x) = 0\}$*.*

*Proof of Lemma D.4.* Since $\Gamma$ is the set of local minimizers, each $x$ in $\Gamma$ must satisfy $\nabla L(x) = 0$. The other direction is proved by noting that $\text{rank}(\{z_i\}_{i \in [n]}) = n$, which implies $\text{rank}(\{\nabla f_i(x)\}_{i \in [n]}) = n$. $\square$

Now, we are ready to prove Lemma 6.3 which is restated below.

**Lemma 6.3.** *Consider the loss function $L$ defined in (14), manifold $\Gamma$ and its open neighborhood defined in (15). For gradient flow $\frac{dx_t}{dt} = -\nabla L(x_t)$ starting at any $x_0 \in U$, it holds that $\Phi(x_0) \in \Gamma$.*

*Proof of Lemma 6.3.* It suffices to prove gradient flow $\frac{dx(t)}{dt} = -\nabla L(x(t))$ converges when $t \to \infty$, as long as $x(0) \in U$. Whenever it converges, it must converge to a stationary point in $U$. The proof will be completed by noting that all stationary point of $L$ in $U$ belongs to $\Gamma$ (Lemma D.4).

Below we prove $\lim_{t\to\infty} x(t)$ exists. Denote $C = \frac{16}{n} \min_{i\in[d]} |u_i(0)v_i(0)|\lambda_{\min}(ZZ^\top)$, then it follows from Lemma D.2 that

$$\left\| \frac{dx(t)}{dt} \right\| = \|\nabla L(x(t))\| \leq \frac{\|\nabla L(x(t))\|_2^2}{\sqrt{CL(x(t))}} = \frac{-\frac{dL(x(t))}{dt}}{\sqrt{L(x(t))}} = -\frac{1}{2\sqrt{C}} \frac{d\sqrt{L(x(t))}}{dt}.$$

Thus the total GF trajectory length is bounded by $\int_{t=0}^{\infty} \left\| \frac{dx(t)}{dt} \right\| dt \leq \int_{t=0}^{\infty} -\frac{1}{2\sqrt{C}} \frac{d\sqrt{L(x(t))}}{ddt} dt \leq \frac{L(x(0))}{2\sqrt{C}}$, where the last inequality uses that $L$ is non-negative over $\mathbb{R}^D$. Therefore, the GF must converge. □

## D.4 PROOF OF RESULTS IN SECTION 6.2

Without loss of generality, we will assume $\sum_{i=1} z_{i,j}^2 > 0$ for all $j \in [d]$, because otherwise we can just delete the unused coordinate, since there won't be any update in the parameter corresponding to that coordinate. Moreover, in both gaussian and boolean setting, it can be shown that with probability 1, $\sum_{i=1} z_{i,j}^2 > 0$ for all $j \in [d]$.

To study the optimal solution to (18), we consider the corresponding $d$-dimensional convex program in terms of $w \in \mathbb{R}^d$, which has been studied in Tropp (2015):

$$\text{minimize} \quad R(w) = \frac{4}{n} \sum_{j=1}^{d} \left( \sum_{i=1}^{n} z_{i,j}^2 \right) |w_j|, \tag{36}$$
$$\text{subject to} \quad Zw = Zw^*.$$

Here we slightly abuse the notation of $R$ and the parameter dimension will be clear from the context. We can relate the optimal solution to (18) to that of (36) via a canonical parametrization defined as follows.

**Definition D.5** (Canonical Parametrization). *For any $w \in \mathbb{R}^d$, we define $\binom{u}{v} = \psi(w) = ([w^\top]_+^{\odot 1/2}, [-w^\top]_+^{\odot 1/2})^\top$ as the canonical parametrization of $w$. Clearly, it holds that $u^{\odot 2} - v^{\odot 2} = w$.*

Indeed, we can show that if (36) has a unique optimal solution, it immediately follows that the optimal solution to (18) is also unique up to sign flips of each coordinate, as summarized in the lemma below.

**Lemma D.6.** *Suppose the optimal solution to (36) is unique and equal to $w^*$. Then the optimal solution to (18) is also unique up to sign flips of each coordinate. In particular, one of them is given by $(\tilde{u}^*, \tilde{v}^*) = \psi(w^*)$, that is, the canonical parametrization of $w^*$.*

*Proof of Lemma D.6.* Let $(\hat{u}, \hat{v})$ be any optimal solution of (18) and we define $\hat{w} = \hat{u}^{\odot 2} - \hat{v}^{\odot 2}$, which is also feasible to (36). By the optimality of $w^*$, we have

$$\sum_{j=1}^{d} \left( \sum_{i=1}^{n} z_{i,j}^2 \right) |w_j^*| \leq \sum_{j=1}^{d} \left( \sum_{i=1}^{n} z_{i,j}^2 \right) |\hat{w}_j| \leq \sum_{j=1}^{d} \left( \sum_{i=1}^{n} z_{i,j}^2 \right) (\hat{u}_j^2 + \hat{v}_j^2). \tag{37}$$

On the other hand, $(\tilde{u}^*, \tilde{v}^*) = \psi(w^*)$ is feasible to (18). Thus, it follows from the optimality of $(\hat{u}, \hat{v})$ that

$$\sum_{j=1}^{d} \left( \sum_{i=1}^{n} z_{i,j}^2 \right) (\hat{u}_j^2 + \hat{v}_j^2) \leq \sum_{j=1}^{d} \left( \sum_{i=1}^{n} z_{i,j}^2 \right) ((\tilde{u}_j^*)^2 + (\tilde{v}_j^*)^2) = \sum_{j=1}^{d} \left( \sum_{i=1}^{n} z_{i,j}^2 \right) |w_j^*|. \tag{38}$$

Combining (37) and (38) yields

$$\sum_{j=1}^{d}\left(\sum_{i=1}^{n}z_{i,j}^2\right)(\hat{u}_j^2+\hat{v}_j^2)=\sum_{j=1}^{d}\left(\sum_{i=1}^{n}z_{i,j}^2\right)|w_j^*|=\sum_{j=1}^{d}\left(\sum_{i=1}^{n}z_{i,j}^2\right)|\hat{u}_j^2-\hat{v}_j^2| \tag{39}$$

which implies that $\hat{u}^{\odot 2}-\hat{v}^{\odot 2}$ is also an optimal solution of (36). Since $w^*$ is the unique optimal solution to (36), we have $\hat{u}^{\odot 2}-\hat{v}^{\odot 2}=w^*$. Moreover, by (39), we must have $\hat{u}^{\odot 2}=[w^*]_+$ and $\hat{u}^{\odot 2}=[w^*]_+$, otherwise the equality would not hold. This completes the proof. $\square$

Therefore, the unique optimality of (18) can be reduced to that of (36). In the sequel, we show that the latter holds for both Boolean and Gaussian random vectors. We divide Lemma 6.6 into to Lemma D.8 and D.7 for clarity.

**Lemma D.7** (Boolean Case). *Let $z_1,\ldots,z_n \overset{i.i.d.}{\sim} \mathrm{Unif}(\{\pm 1\}^d)$. There exist some constants $C,c>0$ such that if the sample size $n$ satisfies*

$$n \geq C[\kappa\ln(d/\kappa)+\kappa]$$

*then with probability at least $1-e^{-cn^2}$, the optimal solution of (18), $(\hat{u},\hat{v})$, is unique up to sign flips of each coordinate and recovers the groundtruth, i.e., $\hat{u}^{\odot 2}-\hat{v}^{\odot 2}=w^*$.*

*Proof of Lemma D.7.* By the assumption that $z_1,\ldots,z_n \overset{i.i.d.}{\sim} \mathrm{Unif}(\{\pm 1\}^d)$, we have $\sum_{i=1}^{n}z_{i,j}^2=n$ for all $j\in[d]$. Then (36) is equivalent to the following optimization problem:

$$\begin{aligned}\text{minimize}\quad & g(w)=\|w\|_1,\\ \text{subject to}\quad & Zw=Z((u^*)^{\odot 2}-(v^*)^{\odot 2}).\end{aligned} \tag{40}$$

This model exactly fits the Example 6.2 in Tropp (2015) with $\sigma=1$ and $\alpha=1/\sqrt{2}$. Then applying Equation (4.2) and Theorem 6.3 in Tropp (2015), (40) has a unique optimal solution equal to $(u^*)^{\odot 2}-(v^*)^{\odot 2}$ with probability at least $1-e^{-ch^2}$ for some constant $c>0$, given that the sample size satisfies

$$n\geq C(\kappa\ln(d/\kappa)+\kappa+h)$$

for some absolute constant $C>0$. Choosing $h=\frac{n}{2C}$ and then adjusting the choices of $C,c$ appropriately yield the desired result. Finally, applying Lemma D.6 finishes the proof. $\square$

The Gaussian case requires more careful treatment.

**Lemma D.8** (Gaussian Case). *Let $z_1,\ldots,z_n \overset{i.i.d.}{\sim} \mathcal{N}(0,I_d)$. There exist some constants $C,c>0$ such that if the sample size satisfies*

$$n\geq C\kappa\ln d,$$

*then with probability at least $1-(2d+1)e^{-cn}$, the optimal solution of (18), $(\hat{u},\hat{v})$, is unique up to sign flips of each coordinate of $\hat{u}$ and $\hat{v}$ and recovers the groundtruth, i.e., $\hat{u}^{\odot 2}-\hat{v}^{\odot 2}=w^*$.*

*Proof of Lemma D.8.* Since $z_1,\ldots,z_n \overset{i.i.d.}{\sim} \mathcal{N}(0,I_d)$, we have

$$\mathbb{P}\left[\sum_{i=1}^{n}z_{i,j}^2\in[n/2,3n/2],\forall j\in[d]\right]\geq 1-2de^{-cn}$$

for some constant $c>0$, and we denote this event by $\mathcal{E}_n$. Therefore, on $\mathcal{E}_n$, we have

$$2\sum_{j=1}^{D}(u_j^2+v_j^2)\leq R(x)\leq 6\sum_{j=1}^{D}(u_j^2+v_j^2)$$

or equivalently,

$$2(\|u^{\odot 2}\|_1+\|v^{\odot 2}\|_1)\leq R(x)\leq 6(\|u^{\odot 2}\|_1+v^{\odot 2}\|_1).$$

Define $w^* = (u^*)^{\odot 2} - (v^*)^{\odot 2}$, and (36) is equivalent to the following convex optimization problem

$$\text{minimize} \quad g(w) = \frac{4}{n} \sum_{j=1}^{d} \left( \sum_{i=1}^{n} z_{i,j}^2 \right) |w_j + w_j^*|, \tag{41}$$

$$\text{subject to} \quad Zw = 0.$$

The point $w = 0$ is feasible for (41), and we claim that this is the unique optimal solution when $n$ is large enough. In detail, assume that there exists a non-zero feasible point $w$ for (41) in the descent cone (Tropp, 2015) $\mathcal{D}(g, w^*)$ of $g$, then

$$\lambda_{\min}(Z; \mathcal{D}(g, w^*)) \leq \frac{\|Zw\|_2}{\|w\|_2} = 0$$

where the equality follows from that $w$ is feasible. Therefore, we only need to show that $\lambda_{\min}(Z; \mathcal{D}(g, x^*))$ is bounded from below for sufficiently large $n$.

On $\mathcal{E}_n$, it holds that $g$ belongs to the following function class

$$\mathcal{G} = \left\{ h : \mathbb{R}^d \to \mathbb{R} \mid h(w) = \sum_{j=1}^{d} v_j |w_j|, v \in \Upsilon \right\} \text{ with } \Upsilon = \{v \in \mathbb{R}^d : v_j \in [2, 6], \forall j \in [d]\}.$$

We identify $g_v \in \mathcal{G}$ with $v \in \Upsilon$, then $\mathcal{D}(g, w^*) \subseteq \cup_{v \in \Upsilon} \mathcal{D}(g_v, w^*)) := \mathcal{D}_\Upsilon$, which further implies that

$$\lambda_{\min}(Z; \mathcal{D}(g, w^*)) \geq \lambda_{\min}(Z; \mathcal{D}_\Upsilon).$$

Recall the definition of minimum conic singular value (Tropp, 2015):

$$\lambda_{\min}(Z; \mathcal{D}_\Upsilon) = \inf_{p \in \mathcal{D}_\Upsilon \cap \mathcal{S}^{d-1}} \sup_{q \in \mathcal{S}^{n-1}} \langle q, Zp \rangle.$$

where $\mathcal{S}^{n-1}$ denotes the unit sphere in $\mathbb{R}^n$. Applying the same argument as in Tropp (2015) yields

$$\mathbb{P}\left[ \lambda_{\min}(Z; \mathcal{D}_\Upsilon) \geq \sqrt{n-1} - w(\mathcal{D}_\Upsilon) - h \right] \geq 1 - e^{-h^2/2}.$$

Take the intersection of this event with $\mathcal{E}_n$, and we obtain from a union bound that

$$\lambda_{\min}(Z; \mathcal{D}(g, w^*)) \geq \sqrt{n-1} - w(\mathcal{D}_\Upsilon) - h \tag{42}$$

with probability at least $1 - e^{-h^2/2} - 2de^{-cn}$. It remains to determine $w(\mathcal{D}_\Upsilon)$, which is defined as

$$w(\mathcal{D}_\Upsilon) = \mathbb{E}_{z \sim \mathcal{N}(0, I_d)} \left[ \sup_{p \in \mathcal{D}_\Upsilon \cap \mathcal{S}^{d-1}} \langle z, p \rangle \right] = \mathbb{E}_{z \sim \mathcal{N}(0, I_d)} \left[ \sup_{v \in \Upsilon} \sup_{p \in \mathcal{D}(g_v, x^*) \cap \mathcal{S}^{d-1}} \langle z, p \rangle \right]. \tag{43}$$

Without loss of generality, we assume that $w^* = (w_1^*, \ldots, w_\kappa^*, 0, \ldots, 0)^\top$ with $w_1^*, \ldots, w_\kappa^* > 0$, otherwise one only needs to specify the signs and the nonzero set of $w^*$ in the sequel. For any $v \in \Upsilon$ and any $p \in \mathcal{D}(g_v, w^*) \cap \mathcal{S}^{d-1}$, there exists some $\tau > 0$ such that $g_v(w^* + \tau \cdot p) \leq g_v(w^*)$, i.e.,

$$\sum_{j=1}^{d} v_j |w_j^* + \tau p_j| \leq \sum_{j=1}^{d} v_j |w_j^*|$$

which further implies that

$$\tau \sum_{j=\kappa+1}^{d} v_j |p_j| \leq \sum_{j=1}^{\kappa} v_j (|w_j^*| - |w_j^* - \tau p_j|) \leq \tau \sum_{j=1}^{\kappa} v_j |p_j|$$

where the second inequality follows from the triangle inequality. Then since each $v_j \in [2, 6]$, it follows that

$$\sum_{j=\kappa+1}^{d} |p_j| \leq 3 \sum_{j=1}^{\kappa} |p_j|.$$

Note that this holds for all $\xi \in \Xi$ simultaneously. Now let us denote $p_{1:\kappa} = (p_1, \ldots, p_\kappa) \in \mathbb{R}^\kappa$ and $p_{(\kappa+1):d} = (p_{\kappa+1}, \ldots, p_d) \in \mathbb{R}^{d-\kappa}$, and similarly for other $d$-dimensional vectors. Then for all $p \in \mathcal{D}_\Upsilon \cap \mathcal{S}^{d-1}$, by Cauchy-Schwartz inequality, we have

$$\|p_{(\kappa+1):d}\|_1 \le 3\|p_{1:\kappa}\|_1 \le 3\sqrt{\kappa}\|p_{1:\kappa}\|_2.$$

Thus, for any $z \in \mathbb{R}^d$ and any $p \in \mathcal{D}_\Upsilon \cap \mathcal{S}^{d-1}$, it follows that

$$
\begin{aligned}
\langle z, p \rangle &= \langle z_{1:\kappa}, p_{1:\kappa} \rangle + \langle z_{(\kappa+1):d}, p_{(\kappa+1):d} \rangle \\
&\le \|z_{1:\kappa}\|_2 \|p_{1:\kappa}\|_2 + \|p_{(\kappa+1):d}\|_1 \cdot \max_{j \in \{\kappa+1,\ldots,d\}} |z_j| \\
&\le \|z_{1:\kappa}\|_2 \|p_{1:\kappa}\|_2 + 3\sqrt{\kappa} \|p_{1:\kappa}\|_2 \cdot \max_{j \in \{\kappa+1,\ldots,d\}} |z_j| \\
&\le \|z_{1:\kappa}\|_2 + 3\sqrt{\kappa} \cdot \max_{j \in \{\kappa+1,\ldots,d\}} |z_j|
\end{aligned}
$$

where the last inequality follows from the fact that $p \in \mathcal{S}^{d-1}$. Therefore, combine the above inequality with (43), and we obtain that

$$
\begin{aligned}
w(\mathcal{D}_\Upsilon) &\le \mathbb{E}\left[ \|z_{1:\kappa}\|_2 + 3\sqrt{\kappa} \cdot \max_{j \in \{\kappa+1,\ldots,d\}} |z_j| \right] \\
&\le \sqrt{\kappa} + 3\sqrt{\kappa} \cdot \mathbb{E}\left[ \max_{j \in \{\kappa+1,\ldots,d\}} |z_j| \right].
\end{aligned}
\tag{44}
$$

where the second inequality follows from the fact that $\mathbb{E}[\|z_{1:\kappa}\|_2] \le \sqrt{\mathbb{E}[\|z_{1:\kappa}\|_2^2]} = \sqrt{\kappa}$. To bound the second term in (44), applying Lemma D.9, it follows from (44) that

$$w(\mathcal{D}_\Upsilon) \le \sqrt{\kappa} + 3\sqrt{2\kappa \ln(2(d-\kappa))}. \tag{45}$$

Therefore, combining (45) and (42), we obtain

$$\lambda_{\min}(Z; \mathcal{D}(g, w^*)) \ge \sqrt{n-1} - \sqrt{\kappa} - 3\sqrt{2\kappa \ln(2(d-\kappa))} - h.$$

Therefore, choosing $h = \sqrt{n-1}/2$, as long as $n$ satisfies that $n \ge C(\kappa \ln d)$ for some constant $C > 0$, we have $\lambda_{\min}(Z; \mathcal{D}(g, w^*)) > 0$ with probability at least $1 - (2d+1)e^{-cn}$. Finally, the uniqueness of the optimal solution to (18) in this case follows from Lemma D.6. $\square$

**Lemma D.9.** *Let $z \sim \mathcal{N}(0, I_d)$, then it holds that $\mathbb{E}\left[\max_{i \in [d]} |z_i|\right] \le \sqrt{2\ln(2d)}$.*

*Proof of Lemma D.9.* Denote $M = \max_{i \in [d]} |z_i|$. For any $\lambda > 0$, by Jensen's inequality, we have

$$e^{\lambda \cdot \mathbb{E}[M]} \le \mathbb{E}\left[e^{\lambda M}\right] = \mathbb{E}\left[\max_{i \in [d]} e^{\lambda |z_i|}\right] \le \sum_{i=1}^d \mathbb{E}\left[e^{\lambda |z_i|}\right].$$

Note that $\mathbb{E}[e^{\lambda |z_i|}] \le 2 \cdot \mathbb{E}[e^{\lambda z_i}]$. Thus, by the expression of the Gaussian moment generating function, we further have

$$e^{\lambda \cdot \mathbb{E}[M]} \le 2 \sum_{i=1}^d \mathbb{E}\left[e^{\lambda z_i}\right] = 2d e^{\lambda^2/2},$$

from which it follows that

$$\mathbb{E}[M] \le \frac{\ln(2d)}{\lambda} + \frac{\lambda}{2}.$$

Choosing $\lambda = \sqrt{2\ln(2d)}$ yields the desired result. $\square$

## D.5 PROOF OF LEMMA 6.5

Instead of studying the convergence of the Riemannian gradient flow directly, it is more convenient to consider it in the ambient space $\mathbb{R}^D$. To do so, we define a Lagrange function $\mathcal{L}(x; \lambda) = R(x) + \sum_{i=1}^n \lambda_i(f_i(x) - y_i)$ for $\lambda \in \mathbb{R}^n$. Based on this Lagrangian, we can continuously extend $\partial\Phi(x)\nabla R(x)$ to the whole space $\mathbb{R}^D$. In specific, we can find a continuous function $F : \mathbb{R}^D \to \mathbb{R}^D$ such that $F(\cdot)|_\Gamma = \partial\Phi(\cdot)\nabla R(\cdot)$. Such an $F$ can be implicitly constructed via the following lemma.

**Lemma D.10.** *The $\ell_2$ norm has a unique minimizer among $\{\nabla_x\mathcal{L}(x; \lambda) \mid \lambda \in \mathbb{R}^n\}$ for any fixed $x \in \mathbb{R}^D$. Thus we can define $F : \mathbb{R}^D \to \mathbb{R}^D$ by $F(x) = \mathrm{argmin}_{g \in \{\nabla_x\mathcal{L}(x;\lambda)|\lambda\in\mathbb{R}^n\}} \|g\|_2$. Moreover, it holds that $\langle F(x), \nabla f_i(x) \rangle = 0$ for all $i \in [n]$.*

*Proof of Lemma D.10.* Fix any $x \in \mathbb{R}^D$. Note that $\{\nabla_x\mathcal{L}(x; \lambda) \mid \lambda \in \mathbb{R}^n\}$ is the subspace spanned by $\{\nabla f_i(x)\}_{i\in[n]}$ shifted by $\nabla R(x)$, thus there is unique minimizer of the $\ell_2$ norm in this set. This implies that $F(x) = \mathrm{argmin}_{g \in \{\nabla_x\mathcal{L}(x;\lambda)|\lambda\in\mathbb{R}^n\}} \|g\|_2$ is well-defined.

To show the second claim, denote $h(\lambda) = \|\nabla_x\mathcal{L}(x; \lambda)\|_2^2/2$, which is a quadratic function of $\lambda \in \mathbb{R}^n$. Then we have

$$\nabla h(\lambda) = \begin{pmatrix} \langle\nabla R(x), \nabla f_1(x)\rangle \\ \vdots \\ \langle\nabla R(x), \nabla f_n(x)\rangle \end{pmatrix} + \begin{pmatrix} \sum_{i=1}^n \lambda_i\langle\nabla f_1(x), \nabla f_i(x)\rangle \\ \vdots \\ \sum_{i=1}^n \lambda_i\langle\nabla f_n(x), \nabla f_i(x)\rangle \end{pmatrix} = \begin{pmatrix} \langle\nabla R(x), \nabla f_1(x)\rangle \\ \vdots \\ \langle\nabla R(x), \nabla f_n(x)\rangle \end{pmatrix} + K(x)\lambda.$$

For any $\lambda$ such that $\nabla_x\mathcal{L}(x; \lambda) = F(x)$, we must have $\nabla h(\lambda) = 0$ by the definition of $F(x)$, which by the above implies

$$(K(x)\lambda)_i = -\langle\nabla R(x), \nabla f_i(x)\rangle \qquad \text{for all } i \in [n].$$

Therefore, we further have

$$\langle F(x), \nabla f_i(x)\rangle = \langle\nabla R(x), \nabla f_i(x)\rangle + \sum_{j=1}^n \lambda_j\langle\nabla f_i(x), \nabla f_j(x)\rangle = \langle\nabla R(x), \nabla f_i(x)\rangle + (K(x)\lambda)_i = 0$$

for all $i \in [n]$. This finishes the proof. □

Hence, with any initialization $x(0) \in \Gamma$, the limiting flow (17) is equivalent to the following dynamics

$$\frac{\mathrm{d}x(t)}{\mathrm{d}t} = -\frac{1}{4}F(x(t)). \tag{46}$$

Thus Lemma 6.5 can be proved by showing that the above $x(t)$ converges to $x^*$ as $t \to \infty$. We first present a series of auxiliary results in below.

**Lemma D.11** (Implications for $F(x) = 0$)**.** *Let $F : \mathbb{R}^D \to \mathbb{R}^D$ be as defined in Lemma D.10. For any $x = \binom{u}{v} \in \mathbb{R}^D$ such that $F(x) = 0$, it holds that for each $j \in [d]$, either $u_j = 0$ or $v_j = 0$.*

*Proof.* Since $F(x) = 0$, it holds for all $j \in [d]$ that,

$$0 = \frac{\partial R}{\partial u_j}(x) + \sum_{i=1}^n \lambda_i(x)\frac{\partial f_i}{\partial u_j}(x) = 2u_j\left[\frac{4}{n}\sum_{i=1}^n z_{i,j}^2 + \sum_{i=1}^n \lambda_i(x)z_{i,j}\right],$$

$$0 = \frac{\partial R}{\partial v_j}(x) + \sum_{i=1}^n \lambda_i(x)\frac{\partial f_i}{\partial v_j}(x) = 2v_j\left[\frac{4}{n}\sum_{i=1}^n z_{i,j}^2 - \sum_{i=1}^n \lambda_i(x)z_{i,j}\right].$$

If there exists some $j \in [d]$ such that $u_j \neq 0$ and $v_j \neq 0$, then it follows from the above two identities that

$$\sum_{i=1}^n z_{i,j}^2 = 0$$

which happens with probability 0 in both the Boolean and Gaussian case. Therefore, we must have $u_j = 0$ or $v_j = 0$ for all $j \in [d]$. □

**Lemma D.12.** *Let $F : \mathbb{R}^D \to \mathbb{R}^D$ be as defined in Lemma D.10. Then $F$ is continuous on $\mathbb{R}^D$.*

*Proof.* **Case I.** We first consider the simpler case of any fixed $x^* \in U = (\mathbb{R} \setminus \{0\})^D$, assuming that $K(x^*)$ is full-rank. Lemma D.10 implies that for any $\lambda \in \mathbb{R}^n$ such that $\nabla_x \mathcal{L}(x^*; \lambda) = F(x^*)$, we have

$$K(x^*)\lambda = -[\nabla f_1(x) \ldots \nabla f_n(x)]^\top \nabla R(x).$$

Thus such $\lambda$ is unique and given by

$$\lambda(x^*) = -K(x^*)^{-1}[\nabla f_1(x) \ldots \nabla f_n(x)]^\top \nabla R(x).$$

Since $K(x)$ is continuous around $x^*$, there exists a sufficiently small $\delta > 0$ such that for any $x \in B_\delta(x^*)$, $K(x)$ is full-rank, which further implies that $K(x)^{-1}$ is also continuous in $B_\delta(x)$. Therefore, by the above characterization of $\lambda$, we see that $\lambda(x)$ is continuous for $x \in B_\delta(x^*)$, and so is $F(x) = \nabla R(x) + \sum_{i=1}^n \lambda_i(x) \nabla f_i(x)$.

**Case II.** Next, we consider all general $x^* \in \mathbb{R}^D$. Here for simplicity, we reorder the coordinates as $x = (u_1, v_1, u_2, v_2, \ldots, u_d, v_d)$ with a slight abuse of notation. Without loss of generality, fix any $x^*$ such that for some $q \in [d]$, $(u_i(0))^2 + (v_i(0))^2 > 0$ for all $i = 1, \ldots, q$ and $u_i^* = v_i^* = 0$ for all $i = q+1, \ldots, d$. Then $\nabla R(x^*)$ and $\{\nabla f_i(x^*)\}_{i \in [n]}$ only depend on $\{z_{i,j}\}_{i \in [n], j \in [q]}$, and for all $i \in [n]$, it holds that

$$(\nabla R(x^*))_{(2q+1):D} = (\nabla f_i(x^*))_{(2q+1):D} = 0.$$

Note that if we replace $\{\nabla f_i(x)\}_{i \in [n]}$ by any fixed and invertible linear transform of itself, it would not affect the definition of $F(x)$. In specific, we can choose an invertible matrix $Q \in \mathbb{R}^{n \times n}$ such that, for some $q' \in [q]$, $(\tilde{z}_1, \ldots, \tilde{z}_n) = (z_1, \ldots, z_n)Q$ satisfies that $\{\tilde{z}_{i,1:q}\}_{i \in [q']}$ is linearly independent and $\tilde{z}_{i,1:q} = 0$ for all $i = q'+1, \ldots, n$. We then consider $\left[\nabla \tilde{f}_1(x), \ldots, \nabla \tilde{f}_n(x)\right] = [\nabla f_1(x), \ldots, \nabla f_n(x)] Q$ and the corresponding $F(x)$. For notational simplicity, we assume that $Q$ can be chosen as the identity matrix, so that $(z_1, \ldots, z_n)$ itself satisfies the above property, and we repeat it here for clarity

$$\{z_{i,1:q}\}_{i \in [q']} \text{ is linearly independent and } \tilde{z}_{i,1:q} = 0 \text{ for all } i = q'+1, \ldots, n. \tag{47}$$

This further implies that

$$(\nabla f_i(x))_{1:(2q)} = 0, \quad \text{for all } i \in \{q'+1, \ldots, n\} \text{ and } x \in \mathbb{R}^D. \tag{48}$$

In the sequel, we use $\lambda$ for $n$-dimensional vectors and $\bar{\lambda}$ for $q'$-dimensional vectors. Denote[2]

$$\lambda(x) \in \operatorname*{argmin}_{\lambda \in \mathbb{R}^n} \left\| \nabla R(x) + \sum_{i=1}^n \lambda_i \nabla f_i(x) \right\|_2,$$

$$\bar{\lambda}(x) \in \operatorname*{argmin}_{\bar{\lambda} \in \mathbb{R}^{q'}} \left\| \left( \nabla R(x) + \sum_{i=1}^{q'} \bar{\lambda}_i \nabla(f_i(x)) \right)_{1:(2q)} \right\|_2.$$

Then due to (47) and (48), we have

$$\left\| \left( \nabla R(x^*) + \sum_{i=1}^{q'} \bar{\lambda}_i(x^*) \nabla f_i(x^*) \right)_{1:(2q)} \right\|_2 = \left\| \nabla R(x^*) + \sum_{i=1}^n \lambda_i(x) \nabla f_i(x^*) \right\|_2 = \|F(x^*)\|_2. \tag{49}$$

---

[2] We do not care about the specific choice of $\lambda(x)$ or $\bar{\lambda}(x)$ when there are multiple candidates, and we only need their properties according to Lemma D.10, so they can be arbitrary. Also, the minimum of $\ell_2$-norm of an affine space can always be attained so argmin exists.

On the other hand, for any $x \in \mathbb{R}^D$, by (48), we have

$$
\left\| \left( \nabla R(x) + \sum_{i=1}^{q'} \bar{\lambda}_i(x) \nabla f_i(x) \right)_{1:(2q)} \right\|_2 = \min_{\lambda \in \mathbb{R}^n} \left\| \left( \nabla R(x) + \sum_{i=1}^{n} \lambda_i(x) \nabla f_i(x) \right)_{1:(2q)} \right\|_2
$$

$$
\leq \left\| \left( \nabla R(x) + \sum_{i=1}^{n} \lambda_i(x) \nabla f_i(x) \right)_{1:(2q)} \right\|_2 = \| F_{1:(2q)}(x) \|_2
$$

$$
\leq \| F(x) \|_2 \leq \left\| \nabla R(x) + \sum_{i=1}^{n} \lambda_i(x^*) \nabla f_i(x) \right\|_2 \tag{50}
$$

where the first and third inequalities follow from the definition of $F(x)$. Let $x \to x^*$, by the continuity of $\nabla R(x)$ and $\{\nabla f_i(x)\}_{i \in [n]}$, we have

$$
\lim_{x \to x^*} \left\| \nabla R(x) + \sum_{i=1}^{n} \lambda_i(x^*) \nabla f_i(x) \right\|_2 = \left\| \nabla R(x^*) + \sum_{i=1}^{n} \lambda_i(x^*) \nabla f_i(x^*) \right\|_2 \tag{51}
$$

Denote $\tilde{K}(x) = (\tilde{K}_{ij}(x))_{(i,j) \in [q']^2} = (\langle \nabla f_i(x)_{1:(2q)}, \nabla f_i(x)_{1:(2q)} \rangle)_{(i,j) \in [q']^2}$. By applying the same argument as in **Case I**, since $\tilde{K}(x^*)$ is full-rank, it also holds that $\lim_{x \to x^*} \bar{\lambda}(x) = \bar{\lambda}(x^*)$, and thus

$$
\lim_{x \to x^*} \left\| \left( \nabla R(x) + \sum_{i=1}^{q'} \bar{\lambda}_i(x) \nabla f_i(x)_{1:(2q)} \right) \right\|_2 = \left\| \left( \nabla R(x) + \sum_{i=1}^{q'} \bar{\lambda}_i(x^*) \nabla f_i(x^*) \right)_{1:(2q)} \right\|_2. \tag{52}
$$

Combing (49), (50), (51) and (52) yields

$$
\lim_{x \to x^*} \| F_{1:(2q)}(x) \|_2 = \lim_{x \to x^*} \min_{\lambda \in \mathbb{R}^n} \left\| \left( \nabla R(x) + \sum_{i=1}^{n} \lambda_i \nabla f_i(x) \right)_{1:(2q)} \right\|_2 = \| F(x^*) \|_2. \tag{53}
$$

Moreover, since $\| F_{(2q+1):D}(x) \|_2 = \sqrt{\| F(x) \|_2^2 - \| F_{1:(2q)}(x) \|_2^2}$, we also have

$$
\lim_{x \to x^*} \| F_{(2q+1):D}(x) \|_2 = 0. \tag{54}
$$

It then remains to show that $\lim_{x \to x^*} F_{1:(2q)}(x) = F_{1:(2q)}(x^*)$, which directly follows from $\lim_{x \to x^*} \lambda_{1:q'}(x) = \lambda_{1:q'}(x^*) = \bar{\lambda}(x^*)$.

Now, for any $\epsilon > 0$, due to the convergence of $\bar{\lambda}(x)$ and that $\tilde{K}(x^*) \succ 0$, we can pick a sufficiently small $\delta_1$ such that for some constant $\alpha > 0$ and all $x \in B_{\delta_1}(x^*)$, it holds that $\| \bar{\lambda}(x) - \bar{\lambda}(x^*) \|_2 \leq \epsilon/2$ and

$$
\left\| \left( \nabla R(x) + \sum_{i=1}^{q'} \bar{\lambda}_i \nabla f_i(x) \right)_{1:(2q)} \right\|_2^2 \geq \left\| \left( \nabla R(x) + \sum_{i=1}^{q'} \bar{\lambda}_i(x) \nabla f_i(x) \right)_{1:(2q)} \right\|_2^2 + \alpha \| \bar{\lambda} - \bar{\lambda}(x) \|_2^2. \tag{55}
$$

for all $\bar{\lambda} \in \mathbb{R}^p$, where the inequality follows from the strong convexity. Meanwhile, due to (48), we have

$$
\lim_{x \to x^*} \left\| \left( \nabla R(x) + \sum_{i=1}^{q'} \lambda_i(x) \nabla f_i(x) \right)_{1:(2q)} \right\|_2 = \lim_{x \to x^*} \left\| \left( \nabla R(x) + \sum_{i=1}^{n} \lambda_i(x) \nabla f_i(x) \right)_{1:(2q)} \right\|_2
$$

$$
= \left\| \left( \nabla R(x) + \sum_{i=1}^{q'} \bar{\lambda}_i(x^*) \nabla f_i(x^*) \right)_{1:(2q)} \right\|_2
$$

$$
= \lim_{x \to x^*} \left\| \left( \nabla R(x) + \sum_{i=1}^{q'} \bar{\lambda}_i(x) \nabla f_i(x) \right)_{1:(2q)} \right\|_2.
$$

where the second equality follows from (53) and the second equality is due to (52). Therefore, we can pick a sufficiently small $\delta_2$ such that

$$\left\|\left(\nabla R(x) + \sum_{i=1}^{q'} \lambda_i(x)\nabla f_i(x)\right)_{1:(2q)}\right\|_2 \le \left\|\left(\nabla R(x) + \sum_{i=1}^{q'} \bar{\lambda}_i(x)\nabla f_i(x)\right)_{1:(2q)}\right\|_2 + \frac{\alpha\epsilon^2}{4} \quad (56)$$

for all $x \in B_{\delta_2}(x^*)$. Setting $\delta = \min(\delta_1, \delta_2)$, it follows from (55) and (56) that

$$\|\lambda_{1:q'}(x) - \bar{\lambda}(x)\|_2 \le \frac{\epsilon}{2}, \quad \text{for all } x \in B_\delta(x^*).$$

Recall that we already have $\|\bar{\lambda}(x) - \bar{\lambda}(x^*)\| \le \epsilon/2$, and thus

$$\|\lambda_{1:q'}(x) - \lambda(x^*)_{1:q'}\|_2 = \|\lambda_{1:q'}(x) - \bar{\lambda}(x^*)\|_2 \le \|\lambda_{1:q'}(x) - \bar{\lambda}(x)\|_2 + \|\bar{\lambda}(x) - \bar{\lambda}(x^*)\|_2 \le \epsilon$$

for all $x \in B_\delta(x^*)$. Therefore, we see that $\lim_{x \to x^*} \lambda_{1:q'}(x) = \lambda(x^*)_{1:q'}$.

Finally, it follows from the triangle inequality that

$$\|F(x) - F(x^*)\|_2 \le \left\|\left(F(x) - F(x^*)\right)_{1:(2q)}\right\|_2 + \|F_{(2q+1):D}(x)\|_2 + \underbrace{\|F_{(2q+1):D}(x^*)\|_2}_{0}$$

$$= \left\|\left(\nabla R(x) + \sum_{i=1}^{q'} \lambda_i(x)\nabla f_i(x) - \nabla R(x^*) - \sum_{i=1}^{q'} \lambda_i(x^*)\nabla f_i(x^*)\right)_{1:(2q)}\right\|_2 + \|F_{(2q+1):D}(x)\|_2$$

$$\le \left\|\sum_{i=1}^{q'} \lambda_i(x)\nabla f_i(x) - \lambda_i(x^*)\nabla f_i(x^*)\right\|_2 + \|\nabla R(x) - \nabla R(x^*)\|_2 + \|F_{(2q+1):D}(x)\|_2$$

where, as $x \to x^*$, the first term vanishes by the convergence of $\lambda_{1:q'}(x)$ and the continuity of each $\nabla f_i(x)$, the second term converges to 0 by the continuity of $\nabla R(x)$ and the third term vanishes by (54). Therefore, we conclude that

$$\lim_{x \to x^*} F(x) = F(x^*),$$

that is, $F$ is continuous. $\qquad \square$

**Lemma D.13.** *For any initialization $x^* \in \Gamma$, the Riemmanian Gradient Flow (17) (or equivalently, (46)) is defined on $[0, \infty)$.*

*Proof of Lemma D.13.* Let $[0, T)$ be the right maximal interval of existence of the solution of Riemannian gradient glow and suppose $T \ne \infty$. Since $R(x(t))$ is monotone decreasing, thus $R(x(t))$ is upper bounded by $R(x(0))$ and therefore $\|\nabla R(x(t))\|$ is also upper bounded. Since $\left\|\frac{dx(t)}{dt}\right\|_2 \le \|\nabla R(x(t))\|_2$ for any $t < T$, the left limit $x(T-) := \lim_{\tau \to T^-} x(\tau)$ must exist. By Corollary 1, Perko (2001), $x(T-)$ belongs to boundary of $U$, i.e., $u_j(T-) = 0$ or $v_j(T-) = 0$ for some $j \in [d]$ by Lemma D.11. By the definition of the Riemannian gradient flow in (17), we have

$$\frac{\mathrm{d}}{\mathrm{d}t}(u_j(t)v_j(t)) = \begin{pmatrix} v_j(t)e_j^\top & u_j(t)e_j^\top \end{pmatrix} \frac{\mathrm{d}x(t)}{\mathrm{d}t}$$

$$= -\frac{1}{4}\begin{pmatrix} v_j(t)e_j^\top & u_j(t)e_j^\top \end{pmatrix} F(x(t)).$$

By the expression of $F(x(t)) = \nabla R(x(t)) + \sum_{i=1}^n \lambda_i(x(t))\nabla f_i(x(t))$, we then have

$$\frac{\mathrm{d}}{\mathrm{d}t}(u_j(t)v_j(t)) = -\left[\frac{2}{n}\sum_{i=1}^n z_{i,j}^2 + \frac{1}{2}\sum_{i=1}^n \lambda_i(x(t))z_{i,j}\right]u_j(t)v_j(t) - \left[\frac{2}{n}\sum_{i=1}^n z_{i,j}^2 - \frac{1}{2}\sum_{i=1}^n \lambda_i(x(t))z_{i,j}\right]u_j(t)v_j(t)$$

$$= -\left(\frac{4}{n}\sum_{i=1}^n z_{i,j}^2\right)u_j(t)v_j(t).$$

Denote $s_j = \frac{4}{n}\sum_{i=1}^n z_{i,j}^2$. It follows that $|u_j(t)v_j(t)| = |u_j(0)v_j(0)|e^{-s_j t}$ for all $t \in [0, T)$. Taking the limit we have $|u_j(T-)v_j(T-)| \ge |u_j(0)v_j(0)|e^{-s_j T} > 0$. Contradiction with $T \ne \infty$! $\qquad \square$

Before showing that $F$ satisfies the PL condition, we need the following two intermediate results. Given two points $u$ and $v$ in $\mathbb{R}^d$, we say $u$ weakly dominate $v$ (written as $u \leq v$) if and only if $u_i \leq v_i$, for all $i \in [d]$. Given two subsets $A$ and $B$ of $\mathbb{R}^D$, we say $A$ weakly dominates $B$ if and only if for any point $v$ in $B$, there exists a point $u \in A$ such that $u \leq v$.

**Lemma D.14.** *For some $q \in [D]$, let $S$ be any $q$-dimensional subspace of $\mathbb{R}^D$ and $P = \{u \in \mathbb{R}^D \mid u_i \geq 0, \forall i \in [D]\}$. Let $u_\star$ be an arbitrary point in $P$ and $Q = P \cap (u_\star + S)$. Then there exists a radius $r > 0$, such that $B_r^1(0) \cap Q$ weakly dominates $Q$, where $B_r^1(0)$ is the $\ell_1$-norm ball of radius $r$ centered at $0$.*

*As a direct implication, for any continuous function $f : P \to \mathbb{R}$, which is coordinate-wise non-decreasing, $\min_{x \in U} f(x)$ can always be achieved.*

*Proof of Lemma D.14.* We will prove by induction on the environment dimension $D$. For the base case of $D = 1$, either $S = \{0\}$ or $S = \mathbb{R}$, and it is straight-forward to verify the desired for both scenarios.

Suppose the proposition holds for $D - 1$, below we show it holds for $D$. For each $i \in [D]$, we apply the proposition with $D - 1$ to $Q \cap \{u \in P \mid u_i = 0\}$ (which can be seen as a subset of $\mathbb{R}^{D-1}$), and let $r_i$ be the corresponding $\ell_1$ radius. Set $r = \max_{i \in [D]} r_i$, and we show that choosing the radius to be $r$ suffices.

For any $v \in Q$, we take a random direction in $S$, denoted by $\omega$. If $\omega \geq 0$ or $\omega \leq 0$, we denote by $y$ the first intersection (i.e., choosing the smallest $\lambda$) between the line $\{v - \lambda|\omega|\}_{\lambda \geq 0}$ and the boundary of $U$, i.e., $\cup_{i=1}^D \{z \in \mathbb{R}^D \mid z_i = 0\}$. Clearly $y \leq v$. By the induction hypothesis, there exists a $u \in B_r^1(0) \cap Q$ such that $u \leq y$. Thus $u \leq v$ and meets our requirement.

If $\omega$ has different signs across its coordinates, we take $y_1, y_2$ to be the first intersections of the line $\{v - \lambda|\omega|\}_{\lambda \in \mathbb{R}}$ and the boundary of $U$ in directions of $\lambda > 0$ and $\lambda < 0$, respectively. Again by the induction hypothesis, there exist $u_1, u_2 \in B_r^1(0) \cap Q$ such that $u_1 \leq y_1$ and $u_2 \leq y_2$. Since $v$ lies in the line connecting $u_1$ and $u_2$, there exists some $h \in [0, 1]$ such that $v = (1 - h)u_1 + hu_2$. It then follows that $(1 - h)u_1 + hu_2 \leq (1 - h)y_1 + hy_2 = v$. Now since $Q$ is convex, we have $(1 - h)u_1 + hu_2 \in Q$, and by the triangle inequality it also holds that $\|(1 - h)u_1 + hu_2\|_1 \leq r$, so $(1 - h)u_1 + hu_2 \in B_r^1(0) \cap Q$. Therefore, we conclude that $B_r^1(0) \cap Q$ weakly dominates $Q$, and thus the proposition holds for $D$. This completes the proof by induction. $\square$

**Lemma D.15.** *For some $q \in [D]$, let $S$ be any $q$-dimensional subspace of $\mathbb{R}^D$ and $P = \{u \in \mathbb{R}^D \mid u_i \geq 0, \forall i \in [D]\}$. Let $u_\star$ be an arbitrary point in $P$ and $Q = P \cap (u_\star + S)$. Then there exists a constant $c \in (0, 1]$ such that for any sufficiently small radius $r > 0$, $c \cdot Q$ weakly dominates $P \cap (u_\star + S + B_r^2(0))$, where $B_r^2(0)$ is the $\ell_2$-norm ball of radius $r$ centered at $0$.*

*Proof of Lemma D.15.* We will prove by induction on the environment dimension $D$. For the base case of $D = 1$, either $S = \{0\}$ or $S = \mathbb{R}$. $S = \mathbb{R}$ is straight-forward; for the case $S = \{0\}$, we just need to ensure $c|u_\star| \leq |u_\star| - r$, and it suffices to pick $r = |u_\star|$ and $c = 0.5$.

Suppose the proposition holds for $D - 1$, below we show it holds for $D$. For each $i \in [D]$, we first consider the intersection between $P \cap (u_\star + S + B_r^2(0))$ and $H_i := \{u \in \mathbb{R}^D \mid u_i = 0\}$. Let $u_i$ be an arbitrary point in $P \cap (u_\star + S) \cap H_i$, then $P \cap (u_\star + S) \cap H_i = P \cap (u_i + S) \cap H_i = P \cap (u_i + S \cap H_i)$. Furthermore, there exists $\{\alpha_i\}_{i \in [D]}$ which only depends on $S$ and satisfies $P \cap (u^* + S + B_r^2(0)) \cap H_i \subset P \cap (u_i + S \cap H_i + B_{\alpha_i r}^2(0) \cap H_i)$. Applying the induction hypothesis to $P \cap (u_i + S \cap H_i + B_{\alpha_i r}^2(0) \cap H_i)$, we know there exists a $c > 0$ such that for sufficiently small $r$, $c(P \cap (u_\star + S) \cap H_i) = c(P \cap (u_i + S \cap H_i))$ weakly dominates $P \cap (u_i + S \cap H_i + B_{\alpha_i r}^2(0) \cap H_i)$.

For any point $v$ in $Q$ and any $z \in B_r^2(0)$, we take a random direction in $S$, denoted by $\omega$. If $\omega \geq 0$ or $\omega \leq 0$, we denote by $y$ the first intersection between $\{v + z - \lambda|\omega|\}_{\lambda \geq 0}$ and the boundary of $U$. Clearly $y \leq v$. Since $y \in P \cap (u_\star + S + B_r^2(0)) \cap H_i \subset P \cap (u_i + S \cap H_i + B_{\alpha_i r}^2(0) \cap H_i)$, by the induction hypothesis, there exists a $u \in c(P \cap (u_\star + S) \cap H_i)$ such that $u \leq y$. Thus $z \leq v + z$ and $z \in c(P \cap (u_\star + S)) = c \cdot Q$.

If $\omega$ has different signs across its coordinates, we take $y_1, y_2$ to be the first intersections of the line $\{v + z - \lambda|\omega|\}_{\lambda \in \mathbb{R}}$ and the boundary of $U$ in directions of $\lambda > 0$ and $\lambda < 0$, respectively. By the induction hypothesis, there exist $u_1, u_2 \in c \cdot Q$ such that $u_1 \leq y_1$ and $u_2 \leq y_2$. Since $v + z$ lies

in the line connecting $u_1$ and $u_2$, there exists some $h \in [0, 1]$ such that $v + z = (1 - h)y_1 + hy_2$. It then follows that $(1 - h)u_1 + hu_2 \leq (1 - h)y_1 + hy_2 = v + z$. Since $Q$ is convex, we have $(1-h)u_1 + hu_2 \in cQ$. Therefore, we conclude that $cQ \cap Q$ weakly dominates $P \cap (u_\star + S + B_r^2(0))$ for all sufficiently small $r$, and thus the proposition holds for $D$. This completes the proof by induction. $\qquad\square$

**Lemma D.16.** *(Polyak-Łojasiewicz condition for $F$.) For any $x^*$ such that $L(x^*) = 0$, i.e., $x^* \in \overline{\Gamma}$, there exist a neighbourhood $U'$ of $x^*$ and a constant $c > 0$, such that $\|F(x)\|_2^2 \geq c \cdot \max(R(x) - R(x^*), 0)$ for all $x \in U' \cap \overline{\Gamma}$. Note this requirement is only non-trivial when $\|F(x^*)\|_2 = 0$ since $F$ is continuous.*

*Proof of Lemma D.16.* It suffices to show the PL condition for $\{x \mid F(x) = 0\}$. We need to show for any $x^*$ satisfying $F(x^*) = 0$, there exist some $\epsilon > 0$ and $C > 0$, such that for all $x \in \overline{\Gamma} \cap B_\epsilon^2(x^*)$ with $R(x) > R(x^*)$, it holds that $\|F(x)\|_2^2 \geq C(R(x) - R(x^*))$.

**Case I.** We first prove the case where $x = \binom{u}{v}$ itself is a canonical parametrization of $w = u^{\odot 2} - v^{\odot 2}$, i.e., $u_j v_j = 0$ for all $j \in [d]$. Since $x^*$ satisfies $\nabla F(x^*) = 0$, by Lemma D.11, we have $x^* = \psi(w^*)$ where $w^* = (u^*)^{\odot 2} - (v^*)^{\odot 2}$. In this case, we can rewrite both $R$ and $F$ as functions of $w \in \mathbb{R}^d$. In detail, we define $R'(w) = R(\psi(w))$ and $F'(w) = F(\psi(w))$ for all $w \in \mathbb{R}^d$. For any $w$ in a sufficiently small neighbourhood of $w^*$, it holds that $\mathrm{sign}(w_j) = \mathrm{sign}(w_j^*)$ for all $j \in [q]$. Below we show that for each possible sign pattern of $w_{(q+1):d}$, there exists some constant $C$ which admits the PL condition in the corresponding orthant. Then we take the minimum of all $C$ from different orthant and the proof is completed. W.L.O.G., we assume that $w_j \geq 0$, for all $j = q + 1 \ldots, d$.

We temporarily reorder the coordinates as $x = (u_1, v_1, u_2, v_2, \ldots, u_d, v_d)^\top$. Recall that $Z = [z_1, \ldots, z_n]^\top$ is a $n$-by-$d$ matrix, and we have

$$\|F'(w)\|_2^2 = \min_{\lambda \in \mathbb{R}^n} \left\langle (a - \mathrm{sign}(w) \odot Z^\top \lambda)^{\odot 2}, |w| \right\rangle,$$

where $a = \frac{8}{n} \sum_{i=1}^n z_i^{\odot 2} \in \mathbb{R}^d$. Since $F(x^*) = 0$, there must exist $\lambda^* \in \mathbb{R}^n$, such that the first $2q$ coordinates of $\nabla R(x^*) + \sum_{i=1}^n \lambda_i^* \nabla f_i(x^*)$ are equal to 0. As argued in the proof of Lemma D.12, we can assume the first $q'$ rows of $Z$ are linear independent on the first $q$ coordinates for some $q' \in [q]$. In other words, $Z$ can be written as $\begin{bmatrix} Z_A & Z_B \\ 0 & Z_D \end{bmatrix}$ where $Z_A \in \mathbb{R}^{q' \times q}$. We further denote $\lambda_a := \lambda_{1:q'}$, $\lambda_b := \lambda_{(q'+1):n}$, $w_a := w_{1:q}$ and $w_b := w_{(q+1):d}$ for convenience, then we have

$$\|F'(w)\|_2^2 = \min_{\lambda \in \mathbb{R}^n} \left\langle (a_1 + \mathrm{sign}(w_a) \odot Z_A^\top \lambda_a)^{\odot 2}, |w_a| \right\rangle + \left\langle (a_2 + Z_B^\top \lambda_a + Z_D^\top \lambda_b)^{\odot 2}, w_b \right\rangle. \quad (57)$$

Since every $w$ in $\overline{\Gamma}$ is a global minimizer, $R'(w) = R'(w) + \sum_{i=1}^n \lambda_i^* (z_i^\top w - y_i) := g^\top w + R'(w_*)$, where $g = \mathrm{sign}(w) \odot a + Z^\top \lambda^*$. Similarly we define $g_a := g_{1:q}$ and $g_b := g_{(q+1):d}$. It holds that $g_a = 0$ and we assume $Z_D g_b = 0$ without loss of generality, because this can always be done by picking suitable $\lambda_i^*$ for $i = q' + 1, \ldots, n$. (We have such freedom on $\lambda_{q'+1:n}^*$ because they doesn't affect the first $2q$ coordinates.)

We denote $\lambda_a - \lambda_a^*$ by $\Delta\lambda_a$, then since $0 = g_a = \mathrm{sign}(w_a) \odot a_1 + Z_A^\top \lambda_a^*$, we further have

$$\left\langle (a_1 + \mathrm{sign}(w_a) \odot Z_A^\top \lambda_a)^{\odot 2}, |w_a| \right\rangle = \left\langle (a_1 + \mathrm{sign}(w_a) \odot Z_A^\top \lambda_a^* + \mathrm{sign}(w_a) \odot Z_A^\top \Delta\lambda_a)^{\odot 2}, |w_a| \right\rangle$$
$$= \left\langle (\mathrm{sign}(w_a) \odot Z_A^\top \Delta\lambda_a)^{\odot 2}, |w_a| \right\rangle.$$

On the other hand, we have $g_b = \mathrm{sign}(w_b) \odot a_2 + Z_B^\top \lambda_a^* + Z_D^\top \lambda_b^* = a_2 + Z_B^\top \lambda_a^* + Z_D^\top \lambda_b^*$ by the assumption that each coordinate of $w_b$ is non-negative. Combining this with the above identity, we can rewrite Equation (57) as:

$$\|F'(w)\|_2^2 = \min_{\lambda \in \mathbb{R}^D} \left\langle (Z_A^\top \Delta\lambda_a)^{\odot 2}, |w_a| \right\rangle + \left\langle (g_b + Z_B^\top \Delta\lambda_a + Z_D^\top \lambda_b)^{\odot 2}, w_b \right\rangle. \quad (58)$$

Now suppose $R'(w) - R'(w^*) = g_b^\top w_b = \delta$ for some sufficiently small $\delta$ (which can be controlled by $\epsilon$). We will proceed in the following two cases separately.

- **Case I.1**: $\|\Delta\lambda_a\|_2 = \Omega(\sqrt{\delta})$. Since $Z_A$ has full row rank, $\left\|(Z_A^\top\Delta\lambda_a)^{\odot 2}\right\|_1 = \left\|(Z_A^\top\Delta\lambda_a)\right\|_2^2 \geq \|\Delta\lambda_a\|_2^2 \lambda_{\min}^2(Z_A)$ is lower-bounded. On the other hand, we can choose $\epsilon$ small enough such that $\forall i \in [q]|(w_a)_i^2| \geq \frac{1}{2}(w_a^*)_i^2$. Thus the first term of Equation (58) is lower bounded by $\|\Delta\lambda_a\|_2^2 \lambda_{\min}^2(Z_A) \cdot \min_{i\in[q]} \frac{1}{2}(w_a^*)_i^2 = \Omega(\delta) = \Omega(R'(w) - R'(w^*))$.

- **Case I.2**: $\|\Delta\lambda_a\|_2 = O(\sqrt{\delta})$. Let $u = g_b + Z_B^\top\Delta\lambda_a + Z_D^\top\lambda_b$, then we have $u \in S + B^2_{c\sqrt{\delta}}(0)$ for some constant $c > 0$, where $S = \{g_b + Z_D^\top\lambda_b \mid \lambda_b \in \mathbb{R}^{n-q'}\}$. By Lemma D.14, there exists some constant $c_0 \geq 1$, such that $\frac{1}{c_0} \cdot S$ weakly dominates $S + B^2_{c\sqrt{\delta}}(0)$. Thus we have $\|F'(w)\|_2^2 \geq \inf_{u\in S+B_{c\sqrt{\delta}}(0)} \langle u^{\odot 2}, w_b \rangle \geq \inf_{u\in\frac{1}{c_0}\cdot S} \langle s^{\odot 2}, w_b \rangle$, where the last step is because each coordinate of $w_b$ is non-negative.

Let $A$ be the orthogonal complement of $\mathrm{span}(Z_D, g_b)$, i.e., the spanned space of columns of $Z_D$ and $g_b$, we know $w_b \in \frac{\delta}{\|g_b\|_2^2}g_b + A$, since $Z_D w_b = Z_D w_*^2 = 0$ and $g_b^\top w_b = \delta$. Therefore,

$$\inf_{w:R'(w)-R'(w^*)=\delta>0} \frac{\|F'(w)\|_2^2}{R'(w) - R'(w^*)} \geq \inf_{w_b:R'(w)-R'(w^*)=\delta>0} \inf_{u\in\frac{1}{c_0}\cdot S} \left\langle u^{\odot 2}, \frac{w_b}{\delta} \right\rangle$$

$$\geq \frac{1}{c_0^2} \inf_{w_b\in\frac{\delta}{\|g_b\|_2^2}g_b+A, w_b\geq 0, u\in S} \left\langle u^{\odot 2}, w_b \right\rangle. \quad (59)$$

Note $\langle u^{\odot 2}, w_b \rangle$ is a monotone non-decreasing function in the first joint orthant, i.e., $\{(u, w_b) \in \mathbb{R}^d \times \mathbb{R}^{d-q'} \mid u \geq 0, w_b \geq 0\}$, thus by Lemma D.15 the infimum can be achieved by some finite $(u, w_b)$ in the joint first orthant. Applying the same argument to each other orthant of $u \in \mathbb{R}^d$, we conclude that the right-hand-side of (59) can be achieved.

On the other hand, we have $u^\top w_b = \delta > 0$ for all $w_b \in \frac{\delta}{\|g_b\|_2^2}g_b + A$ and $u \in S$, by $Z_D g_b = 0$ and the definition of $A$. This implies there exists at least one $i \in [d - q']$ such that $w_{2,i}u_i > 0$, which further implies $\langle u^{\odot 2}, w_b \rangle > 0$. Therefore, we conclude that $\|F'(w)\|_2^2 = \Omega(R'(w) - R'(w_0))$.

**Case II.** Next, for any general $x = \binom{u}{v}$, we define $w = u^{\odot 2} - v^{\odot 2}$ and $m = \min\{u^{\odot 2}, v^{\odot 2}\}$, where $\min$ is taken coordinate-wise. Then we can rewrite $\|F(x)\|_2^2$ as

$$\|F(x)\|_2^2 = \min_{\lambda\in\mathbb{R}^n} \left\| \left( \begin{bmatrix} a \\ a \end{bmatrix} + \begin{bmatrix} Z \\ -Z \end{bmatrix}\lambda \right) \odot \begin{bmatrix} u \\ v \end{bmatrix} \right\|_2^2$$

$$= \min_{\lambda\in\mathbb{R}^n} \left\| \left( \begin{bmatrix} a \\ a \end{bmatrix} + \begin{bmatrix} Z \\ -Z \end{bmatrix}\lambda \right)^{\odot 2} \odot \begin{bmatrix} u^{\odot 2} \\ v^{\odot 2} \end{bmatrix} \right\|_1$$

$$= \min_{\lambda\in\mathbb{R}^n} \left\| \left( \begin{bmatrix} a \\ a \end{bmatrix} + \begin{bmatrix} Z \\ -Z \end{bmatrix}\lambda \right)^{\odot 2} \odot \left( \psi(w)^{\odot 2} + \begin{bmatrix} m \\ m \end{bmatrix} \right) \right\|_1$$

$$\geq \min_{\lambda\in\mathbb{R}^n} \left\| \left( \begin{bmatrix} a \\ a \end{bmatrix} + \begin{bmatrix} Z \\ -Z \end{bmatrix}\lambda \right)^{\odot 2} \odot \psi(w)^{\odot 2} \right\|_1 + \min_{\lambda\in\mathbb{R}^n} \left\| \left( \begin{bmatrix} a \\ a \end{bmatrix} + \begin{bmatrix} Z \\ -Z \end{bmatrix}\lambda \right)^{\odot 2} \odot \begin{bmatrix} m \\ m \end{bmatrix} \right\|_1$$

$$= \min_{\lambda\in\mathbb{R}^n} \left\| \left( \begin{bmatrix} a \\ a \end{bmatrix} + \begin{bmatrix} Z \\ -Z \end{bmatrix}\lambda \right) \odot \psi(w) \right\|_2^2 + \min_{\lambda\in\mathbb{R}^n} \left\| \left( \begin{bmatrix} a \\ a \end{bmatrix} + \begin{bmatrix} Z \\ -Z \end{bmatrix}\lambda \right) \odot \begin{bmatrix} \sqrt{m} \\ \sqrt{m} \end{bmatrix} \right\|_2^2.$$

Then applying the result for the previous case yields the following for some constant $C \in (0,1)$:

$$
\begin{aligned}
\|F(x)\|_2^2 &\geq C(R(\psi(w)) - R(\psi(w^*))) + \min_{\lambda \in \mathbb{R}^n} \left\| \left( \begin{bmatrix} a \\ a \end{bmatrix} + \begin{bmatrix} Z \\ -Z \end{bmatrix} \lambda \right) \odot \begin{bmatrix} \sqrt{m} \\ \sqrt{m} \end{bmatrix} \right\|_2^2 \\
&= C(R(\psi(w)) - R(x^*)) + 2 \left\langle a^{\odot 2}, m \right\rangle \\
&\geq C(R(\psi(w)) - R(x^*)) + 2 \min_{i \in [d]} a_i \left\langle a, m \right\rangle \\
&= C(R(\psi(w)) - R(x^*)) + \min_{i \in [d]} a_i (R(x) - R(\psi(w))) \\
&\geq \min \left\{ C, \min_{i \in [d]} a_i \right\} (R(x) - R(x^*)),
\end{aligned}
$$

where the first equality follows from the fact that $x^* = \psi(w^*)$ and the last inequality is due to the fact that both $R(\psi(w) - R(\psi(w^*)))$ and $R(x) - R(\psi(w))$ are non-negative. This completes the proof. $\qquad \square$

Now, based on the PL condition, we can show that (17) indeed converges.

**Lemma D.17.** *The trajectory of the flow defined in* (17) *has finite length, i.e., $\int_{t=0}^{\infty} \|\frac{\mathrm{d}x}{\mathrm{d}t}\|_2 \mathrm{d}t < \infty$ for any $x^* \in \Gamma$. Moreover, $x(t)$ converges to some $x(\infty)$ when $t \to \infty$ with $F(x(\infty)) = 0$.*

*Proof of Lemma D.17.* Note that along the Riemannian gradient flow, $R(x(t))$ is non-increasing, thus $\|x(t)\|_2$ is bounded over time and $\{x(t)\}_{t \geq 0}$ has at least one limit point, which we will call $x^*$. Therefore, $R(x^*)$ is a limit point of $R(x(t))$, and again since $R(x(t))$ is non-increasing, it follows that $R(x(t)) \geq R(x^*)$ and $\lim_{t \to \infty} R(x(t)) = R(x^*)$. Below we will show $\lim_{t \to \infty} x(t) = x^*$.

Note that $\frac{\mathrm{d}R(x(t))}{\mathrm{d}t} = \left\langle \nabla R(x(t)), \frac{\mathrm{d}x(t)}{\mathrm{d}t} \right\rangle = -\left\langle \nabla R(x(t)), \frac{1}{4}F(x(t)) \right\rangle = -\frac{1}{4} \|F(x(t))\|_2^2$ where the last equality applies Lemma D.10. By Lemma D.16, there exists a neighbourhood of $x^*$, $U'$, in which PL condition holds of $F$. Since $x^*$ is a limit point, there exists a time $T_0$, such that $x_{T_0} \in U$. Let $T_1 = \inf_{t \geq T_0} \{x(t) \notin U'\}$ (which is equal to $\infty$ if $x(t) \in U'$ for all $t \geq T_0$). Since $x(t)$ is continuous in $t$ and $U$ is open, we know $T_1 > T_0$ and for all $t \in [T_0, T_1)$, we have $\|F(x(t))\|_2 \geq \sqrt{c}(R(x(t)) - R(x^*))^{1/2}$.

Thus it holds that for $t \in [T_0, T_1)$,

$$
\frac{\mathrm{d}(R(x(t)) - R(x^*))}{\mathrm{d}t} \leq -\frac{\sqrt{c}}{4}(R(x(t)) - R(x^*))^{1/2} \|F(x(t))\|_2,
$$

that is,

$$
\frac{\mathrm{d}(R(x(t)) - R(x^*))^{1/2}}{\mathrm{d}t} \leq -\frac{\sqrt{c}}{8} \|F(x(t))\|_2.
$$

Therefore, we have

$$
\int_{t=T_0}^{T_1} \|F(x(t))\|_2 \, \mathrm{d}t \leq \frac{8}{\sqrt{c}}(R(x(T_0)) - R(x^*))^{1/2}. \tag{60}
$$

Thus if we pick $T_0$ such that $R(x(T_0)) - R(x^*)$ is sufficiently small, $R(T_1)$ will remain in $U$, which implies that $T_1$ cannot be finite and has to be $\infty$. Therefore, Equation (60) shows that the trajectory of $x(t)$ is of finite length, so $x(\infty) := \lim_{t \to \infty} x(t)$ exists and is equal to $x^*$. As a by-product, $F(x^*)$ must be 0. $\qquad \square$

Finally, collecting all the above lemmas, we are able to prove Lemma 6.5. In Lemma D.17 we already show the convergence of $x(t)$ as $t \to \infty$, the main part of the proof of Lemma 6.5 is to show the $x(\infty)$ cannot be sub-optimal stationary points of $R$ on $\overline{\Gamma}$, the closure of $\Gamma$. The key idea here is that we can construct a different potential $\phi$ for each such sub-optimal stationary point $x^*$, such that (1) $\phi(x_t)$ is locally increasing in a sufficiently neighborhood of $x^*$ and (2) $\lim_{x \to x^*} \phi(x) = -\infty$.

**Lemma 6.5.** *Let $\{x_t\}_{t \geq 0} \subseteq \mathbb{R}^D$ be generated by the flow defined in* (17) *with any initialization $x_0 \in \Gamma$. Then $x_\infty = \lim_{t \to \infty} x_t$ exists. Moreover, $x_\infty = x^*$ is the optimal solution of* (18).

*Proof of Lemma 6.5.* We will prove by contradiction. Suppose $x(\infty) = \binom{u(\infty)}{v(\infty)} = \lim_{t\to\infty} x(t)$ is not the optimal solution to (18). Denote $w(t) = (u(t))^{\odot 2} - (v(t))^{\odot 2}$, then $w(\infty) = \lim_{t\to\infty} w(t)$ is not the optimal solution to (36). Thus we have $R(w(t)) > R(w^*)$. Without loss of generality, suppose there is some $q \in [d]$ such that $(u_i(\infty))^2 + (v_i(\infty))^2 > 0$ for all $i = 1, \dots, q$ and $u_i(\infty) = v_i(\infty) = 0$ for all $i = q+1, \dots, d$. Again, as argued in the proof of Lemma D.12, we can assume that, for some $q' \in [q]$,

$$\{z_{i,1:q}\}_{i\in[q']} \text{ is linearly independent and } z_{i,1:q} = 0 \text{ for all } i = q'+1, \dots, n. \tag{61}$$

Since both $w(\infty)$ and $w^*$ satisfy the constraint that $Zw(\infty) = Zw^* = Y$, we further have

$$0 = \langle z_i, w(\infty) \rangle = \langle z_i, w^* \rangle = \langle z_{i,(q+1):d}, w^*_{(q+1):d} \rangle, \quad \text{for all } i = q'+1, \dots, n. \tag{62}$$

Consider a potential function $\varphi : U \to \mathbb{R}$ defined as

$$\varphi(x) = \varphi(u, v) = \sum_{j=q+1}^{d} w^*_j \left[ \ln(u_j)^2 \mathbb{1}\{w^*_j > 0\} - \ln(v_j)^2 \mathbb{1}\{w^*_j < 0\} \right].$$

Clearly $\lim_{t\to\infty} \varphi(x(t)) = -\infty$ if $\lim_{t\to\infty} x(t) = x(\infty)$. Below we will show contradiction if $x(\infty)$ is suboptimal. Consider the dynamics of $\varphi(x)$ along the Riemannian gradient flow:

$$\frac{\mathrm{d}\varphi}{\mathrm{d}t}(x(t)) = \left\langle \nabla\varphi(x(t)), \frac{\mathrm{d}x(t)}{\mathrm{d}t} \right\rangle = -\left\langle \nabla\varphi(x(t)), \frac{1}{4}F(x(t)) \right\rangle \tag{63}$$

where $F$ is defined previously in Lemma D.10. Recall the definition of $F$, and we have

$$\langle \nabla\varphi(x(t)), F(x(t)) \rangle = \underbrace{\left\langle \nabla\varphi(x(t)), \frac{1}{4}\nabla R(x(t)) + \frac{1}{4}\sum_{i=1}^{q'} \lambda_i(x(t))\nabla f_i(x(t)) \right\rangle}_{\mathcal{I}_1}$$

$$+ \underbrace{\left\langle \nabla\varphi(x(t)), \frac{1}{4}\sum_{i=q'+1}^{n} \lambda_i(x(t))\nabla f_i(x(t)) \right\rangle}_{\mathcal{I}_2}. \tag{64}$$

To show $\langle \nabla\varphi(x(t)), F(x(t)) \rangle < 0$, we analyze $\mathcal{I}_1$ and $\mathcal{I}_2$ separately. By the definition of $\varphi(x)$, we have

$$\nabla\varphi(x) = \sum_{j=q+1}^{d} 2w^*_j \left[ \frac{\mathbb{1}\{w^*_j > 0\}}{u_j} \cdot e_j - \frac{\mathbb{1}\{w^*_j < 0\}}{v_j} \cdot e_{D+j} \right]$$

where $e_j$ is the $j$-th canonical base of $\mathbb{R}^d$. Recall that $\nabla f_i(x) = 2\binom{z_i \odot u}{-z_i \odot v}$, and we further have

$$\mathcal{I}_2 = \sum_{i=q'+1}^{n} \lambda_i(x(t)) \sum_{j=q+1}^{d} w^*_j \left[ \frac{\mathbb{1}\{w^*_j > 0\}}{u_j} \langle e_j, z_i \odot u \rangle + \frac{\mathbb{1}\{w^*_j < 0\}}{v_j} \langle e_j, z_i \odot v \rangle \right]$$

$$= \sum_{i=q'+1}^{n} \lambda_i(x(t)) \sum_{j=q+1}^{d} w^*_j \left[ \frac{\mathbb{1}\{w^*_j > 0\}}{u_j} z_{i,j} u_j + \frac{\mathbb{1}\{w^*_j < 0\}}{v_j} z_{i,j} v_j \right]$$

$$= \sum_{i=q'+1}^{n} \lambda_i(x(t)) \sum_{j=q+1}^{d} w^*_j z_{i,j} = \sum_{i=q'+1}^{n} \lambda_i(x(t)) \langle z_{i,(q+1):d}, w^*_{(q+1):d} \rangle = 0 \tag{65}$$

where the last equality follows from (62).

Next, we show that $\mathcal{I}_1 < 0$ by utilizing the fact that $w^* - w(\infty)$ is a descent direction of $R'(w)$. For $w \in \mathbb{R}^d$, define $\tilde{f}_i(w) = z_i^\top w$ and

$$\tilde{R}(w) = R(w) + \sum_{i=1}^{q'} \lambda_i(x(\infty))(\tilde{f}_i(w) - y_i).$$

Clearly, for any $w \in \mathbb{R}^D$ satisfying $Zw = Y$, it holds that $\tilde{f}_i(w) - y_i = 0$ for each $i \in [n]$, and thus $R(w) = \tilde{R}(w)$. In particular, we have $\tilde{R}(w(\infty)) = R(w(\infty)) > R(w^*) = \tilde{R}(w^*)$. Since $\tilde{R}(w)$ is a convex function, it follows that $\tilde{R}(w(\infty) + s(w^* - w(\infty))) \leq s\tilde{R}(w^*) + (1-s)\tilde{R}(\infty) < \tilde{R}(w(\infty))$ for all $0 < s \leq 1$, which implies $\frac{d\tilde{R}}{dt}(w(\infty) + s(w^* - w(\infty)))|_{s=0} < -2c < 0^+$ for some constant $c > 0$. Note that, for small enough $s > 0$, we have

$$R(w(\infty) + s(w^* - w(\infty))) = \frac{4}{n} \sum_{j=1}^{d} \left( \sum_{i=1}^{n} z_{i,j}^2 \right) |w_j(\infty) + s(w_j^* - w_j(\infty))|$$

$$= \frac{4}{n} \sum_{j=1}^{q} \left( \sum_{i=1}^{n} z_{i,j}^2 \right) \text{sign}(w_j(\infty))(w_j(\infty) + s(w_j^* - w_j(\infty)))$$

$$+ \frac{4}{n} \sum_{j=q+1}^{d} \left( \sum_{i=1}^{n} z_{i,j}^2 \right) s|w_j^*|.$$

Therefore, we can compute the derivative with respect to $s$ at $s = 0$ as

$$-2c > \frac{d\tilde{R}}{dt}(w(\infty) + s(w^* - w(\infty)))\bigg|_{s=0}$$

$$= \frac{4}{n} \sum_{j=1}^{q} \left( \sum_{i=1}^{n} z_{i,j}^2 \right) \text{sign}(w_j(\infty))(w_j^* - w_j(\infty)) + \frac{4}{n} \sum_{j=q+1}^{d} \left( \sum_{i=1}^{n} z_{i,j}^2 \right) |w_j^*|$$

$$+ \sum_{i=1}^{q'} \lambda_i(x(\infty)) z_i^\top (w^* - w_j(\infty))$$

$$= \frac{4}{n} \sum_{j=1}^{q} \left( \sum_{i=1}^{n} z_{i,j}^2 \right) \text{sign}(w_j(\infty))(w_j^* - w(\infty)) + \frac{4}{n} \sum_{j=q+1}^{d} \left( \sum_{i=1}^{n} z_{i,j}^2 \right) |w_j^*|$$

$$+ \sum_{j=1}^{q}(w_j^* - w_j(\infty)) \sum_{i=1}^{q'} \lambda_i(x(\infty))z_{i,j} + \sum_{j=q+1}^{d} w_j^* \sum_{i=1}^{q'} \lambda_i(x(\infty))z_{i,j} \qquad (66)$$

where the second equality follows from the fact that $w_{(q+1):d}(\infty) = 0$. Since $x(t)$ converges to $x(\infty)$, we must have $F(x(\infty)) = 0$, which implies that for each $j \in \{1, \ldots, q\}$,

$$0 = \frac{\partial R}{\partial u_j}(x(\infty)) + \sum_{i=1}^{q'} \lambda_i(x(\infty)) \frac{\partial f_i}{\partial u_j}(x(\infty)) = 2u_j(\infty) \left[ \frac{4}{n} \sum_{i=1}^{n} z_{i,j}^2 + \sum_{i=1}^{q'} \lambda_i(x(\infty))z_{i,j} \right],$$

$$0 = \frac{\partial R}{\partial v_j}(x(\infty)) + \sum_{i=1}^{q'} \lambda_i(x(\infty)) \frac{\partial f_i}{\partial v_j}(x(\infty)) = 2v_j(\infty) \left[ \frac{4}{n} \sum_{i=1}^{n} z_{i,j}^2 - \sum_{i=1}^{q'} \lambda_i(x(\infty))z_{i,j} \right].$$

Combining the above two equalities yields

$$\frac{4}{n} \sum_{i=1}^{n} z_{i,j}^2 = -\text{sign}(w_j(\infty)) \sum_{i=1}^{q'} \lambda_i(x(\infty))z_{i,j}, \quad \text{for all } j \in [q].$$

Apply the above identity together with (66), and we obtain

$$-2c > \sum_{j=1}^{q} -\text{sign}(w_j(\infty))^2(w_j^* - w(\infty)) \sum_{i=1}^{q'} \lambda_i(x(\infty))z_{i,j} + \frac{4}{n} \sum_{j=q+1}^{d} \left( \sum_{i=1}^{n} z_{i,j}^2 \right) |w_j^*|$$

$$+ \sum_{j=1}^{q}(w_j^* - w_j(\infty)) \sum_{i=1}^{q'} \lambda_i(x(\infty))z_{i,j} + \sum_{j=q+1}^{d} w_j^* \sum_{i=1}^{q'} \lambda_i(x(\infty))z_{i,j}$$

$$= \frac{4}{n} \sum_{j=q+1}^{d} \left( \sum_{i=1}^{n} z_{i,j}^2 \right) |w_j^*| + \sum_{j=q+1}^{d} w_j^* \sum_{i=1}^{q'} \lambda_i(x(\infty))z_{i,j} \qquad (67)$$

On the other hand, by directly evaluating $\nabla R(x(t))$ and each $\nabla f_i(x(t))$, we can compute $\mathcal{I}_1$ as

$$
\begin{aligned}
\mathcal{I}_1 &= \sum_{j=q+1}^{d} \frac{w_j^* \mathbb{1}\{w_j^* > 0\}}{u_j(t)} \left[ \frac{2}{n} \sum_{i=1}^{n} z_{i,j}^2 u_j(t) + \frac{1}{2} \sum_{i=1}^{q'} \lambda_i(x(t)) z_{i,j} u_j(t) \right] \\
&\quad - \sum_{j=q+1}^{d} \frac{w_j^* \mathbb{1}\{w_j^* < 0\}}{v_j(t)} \left[ \frac{2}{n} \sum_{i=1}^{n} z_{i,j}^2 v_j(t) - \frac{1}{2} \sum_{i=1}^{q'} \lambda_i(x(t)) z_{i,j} v_j(t) \right] \\
&= \frac{2}{n} \sum_{j=q+1}^{d} \left( \sum_{i=1}^{n} z_{i,j}^2 \right) |w_j^*| + \frac{1}{2} \sum_{j=q+1}^{d} w_j^* \sum_{i=1}^{q'} \lambda_i(x(t)) z_{i,j} \\
&= \frac{2}{n} \sum_{j=q+1}^{d} \left( \sum_{i=1}^{n} z_{i,j}^2 \right) |w_j^*| + \frac{1}{2} \sum_{j=q+1}^{d} w_j^* \sum_{i=1}^{q'} \lambda_i(x(\infty)) z_{i,j} \\
&\quad + \frac{1}{2} \sum_{j=q+1}^{d} w_j^* \sum_{i=1}^{q'} \left( \lambda_i(x(t)) - \lambda_i(x(\infty)) \right) z_{i,j}.
\end{aligned}
$$

We already know that $\lambda_{1:q'}(x)$ is continuous at $x(\infty)$ by the proof of Lemma D.12, so the third term converges to 0 as $x(t)$ tends to $x(\infty)$. Now, applying (67), we immediately see that there exists some $\delta > 0$ such that $\mathcal{I}_1 < -c$ for $x(t) \in B_\delta(x(\infty))$. As we have shown in the above that $\mathcal{I}_2 = 0$, it then follows from (63) and (64) that

$$
\frac{\mathrm{d}\varphi}{\mathrm{d}t}(x(t)) > c, \quad \text{for all } x(t) \in B_\delta(x(\infty)). \tag{68}
$$

Since $\lim_{t\to\infty} x(t) = x(\infty)$, there exists some $T > 0$ such that $x(t) \in B_\delta(x(\infty))$ for all $t > T$. By the proof of Lemma D.13, we know that $\varphi(x(T)) > -\infty$, then it follows from (68) that

$$
\lim_{t\to\infty} \varphi(x(t)) = \varphi(x(T)) + \int_T^\infty \frac{\mathrm{d}\varphi(x(t))}{\mathrm{d}t} \mathrm{d}t > \varphi(x(T)) + \int_T^\infty c \, \mathrm{d}t = \infty
$$

which is a contradiction. This finishes the proof. $\qquad\square$

### D.6 PROOF OF THEOREM 6.7

Here we present the lower bound on the sample complexity of GD in the kernel regime.

**Theorem 6.7.** *Assume $z_1, \ldots, z_n \overset{i.i.d.}{\sim} \mathcal{N}(0, I_d)$ and $y_i = z_i^\top w^*$, for all $i \in [n]$. Define the loss with linearized model as $L(x) = \sum_{i=1}^{n} (f_i(x_0) + \langle \nabla f_i(x_0), x - x_0 \rangle - y_i)^2$, where $x = \binom{u}{v}$ and $x_0 = \binom{u_0}{v_0} = \alpha \binom{\mathbb{1}}{\mathbb{1}}$. Then for any groundtruth $w^*$, any learning rate schedule $\{\eta_t\}_{t \geq 1}$, and any fixed number of steps $T$, the expected $\ell_2$ loss of $x(T)$ is at least $\left(1 - \frac{n}{d}\right) \|w^*\|_2^2$, where $x(T)$ is the $T$-th iterate of GD on $L$, i.e., $x(t+1) = x(t) - \eta_t \nabla L(x(t))$, for all $t \geq 0$.*

*Proof of Theorem 6.7.* We first simplify the loss function by substituting $x' = x - x(0)$, so correspondingly $x_0' = 0$ and we consider $L'(x') := L(x) = (\langle \nabla f_i(x(0)), x' \rangle - y_i)^2$. We can think as if GD is performed on $L'(x')$. For simplicity, we still use the $x$ and $L(x)$ notation in below.

In order to show test loss lower bound against a single fixed target function, we must take the properties of the algorithm into account. The proof is based on the observation that GD is rotationally equivariant (Ng, 2004; Li et al., 2020c) as an iterative algorithm, i.e., if one rotates the entire data distribution (including both the training and test data), the expected loss of the learned function remains the same. Since the data distribution and initialization are invariant under any rotation, it means the expected loss of $x(T)$ with ground truth being $w^*$ is the same as the case where the ground truth is uniformly randomly sampled from all vectors of $\ell_2$-norm $\|w^*\|_2$.

Thus the test loss of $x(T)$ is

$$
\mathbb{E}_z \left[ (\langle \nabla f_z(x(0)), x(T) \rangle - \langle z, w^* \rangle)^2 \right] = \mathbb{E}_z \left[ (\langle z, w^* - (u(T) - v(T)) \rangle)^2 \right] = \|w^* - (u(T) - v(T))\|_2^2. \tag{69}
$$

Note $x(T) \in \text{span}\{\nabla f_x(x(0))\}$, which is at most an $n$-dimensional space spanned by the gradients of model output at $x(0)$, so is $u(T) - v(T)$. We denote the corresponding space for $u(T) - v(T)$ by $S$, so $\dim(S) \leq n$ and it holds that $\|w^* - (u(T) - v(T))\|_2^2 \geq \|(I_D - P_S)w^*\|_2^2$, where $P_S$ is projection matrix onto space $S$.

The expected test loss is lower bounded by

$$
\begin{aligned}
\mathbb{E}_{w^*}\left[\mathbb{E}_{z_i}\left[\|w^* - (u(T) - v(T))\|_2^2\right]\right] &= \mathbb{E}_{z_i}\left[\mathbb{E}_{w^*}\left[\|w^* - (u(T) - v(T))\|_2^2\right]\right] \\
&\geq \min_{\{z_i\}_{i\in[n]}} \mathbb{E}_{w^*}\left[\|(I_D - P_S)w^*\|_2^2\right] \\
&\geq \left(1 - \frac{n}{d}\right)\|w^*\|_2^2.
\end{aligned}
$$

$\square$

