# OpenReview forum: "What Happens after SGD Reaches Zero Loss? --A Mathematical Framework"
_ICLR.cc/2022/Conference — ICLR 2022 Spotlight_

### Official Review · Reviewer_Msxe · 2021-10-29

**Correctness:** 4
**Technical Novelty And Significance:** 4
**Empirical Novelty And Significance:** Not applicable
**Recommendation:** 10
**Confidence:** 5

**Main Review:**

First, I would like to thank the authors for the paper, the intuitive explanations, the clear writing and the proper referencing of the very nice Katzenberger article.

Both as a reviewer and a curious researcher I really enjoyed reading this article. I truly think it tackles a very nice question about the effect of the noise of SGD in minima selection. Remarkably, the paper introduces the very mathematical object that lacks in the Blanc et al. paper. Even if the limiting diffusion, in all its generality, is difficult to interpret, it shows properly some already known intuition in the case of SGD for machine learning problems with label noise. Finally, to give an explicit example where the limiting dynamics can be analysed thoroughly, the authors show optimality of label noise SGD for overparametrised linear network: what a nice result and application !

Obviously, as one can read, I am very enthusiastic about this paper, but let me try to help the authors with a few notes:

I would say that sometimes the writing is a little bit too direct and should be tempered a bit: the authors study the impact of the noise locally near $\Gamma$ with an effective dynamics. To « trap » the dynamics in $\Gamma$, the consider a regime for which the step-sizes go to zero (and along with it the effect of the noise itself). Hence I see this result as a limiting one: surely in practice, the dynamics is not exactly the one depicted: a first phase where the loss goes to zero and a second phase where the dynamics diffuses in $\Gamma$. Of course, it does not diminish the nice impression that the paper gives me, but it should be clearer that to clearly separate the two phases the authors consider a limiting regime.

Even if this work is already pretty complete:
- I would like to read more interpretations of the limiting process of Eq. (10). What happens, e.g., for other neural networks than the one depicted in Section 6 ?
- What happens if one does not add artificially label noise, e.g. with the model of Section 6 ? I expect that the analysis collapses because the noise itself is killed in $\Gamma$. If not how is the dynamics changed ?
- Can you give more intuition on the effective dynamics of equation (17) ? How are killed the directions for which the sparsest predictor is zero ?

For the modelling of machine learning-SGD with an SDE, authors should add the nice work of S.Wojtowytsch [ https://arxiv.org/abs/2106.02588 ]. Here, it is explained that the nature of SGD noise for machine learning problem is very degenerate. Another nice reference for your problem is the nice work of Pesme et al. [ https://arxiv.org/abs/2106.09524 ] on the implicit bias of the same model as the one studied in Section 6. It should also be pointed out that Lemma 6.3 is well known for these models: despite the fact that Woodworth et al., 2020; Azulay et al., 2021 assume convergence other works show it directly, e.g. in Pesme et al.. Hence I cannot consider it as a contribution (yet, it is in my opinion negligible w.r.t. the main message of the paper!).

**Summary Of The Paper:**

The authors of the paper under review carry a general analysis of the effect of the noise near the manifold of global minimum $\Gamma = [x, L(x) = 0]$. More precisely they show that in the limit of infinitesimal step-sizes, a time rescaled effective dynamics follows a diffusion in the manifold $\Gamma$ that mixes effects of the noise and the gradient flow. Leveraging this, they characterise explicitly the limiting point recovered by SGD on a sparse problem trained with label noise on a overparametrised linear network of depth 2.

**Summary Of The Review:**

As already expressed, I am very enthusiastic about the paper as they introduce the perfect mathematic tool and scale that characterizes nicely the effect of noise near the interpolation manifold. Furthermore they apply flawlessly and successfully their vision to a nice problem. In a word: Congrats !

---

> ### Author Response · Authors · 2021-11-18
> **Response to the Review by Reviewer Msxe**
>
> We thank the reviewer for the great appreciation and the expert advice!
>
> 1. **Clarification on the two phases of the dynamics:** Thank you for the suggestion! We modified our intro and added Remark B.10 accompanied by a graphical demonstration (Figure 2) in the appendix (both modifications are in purple fonts) to highlight the two-phase structure in the revision.
>
> 2. **More interpretation on the limiting diffusion, Eq. (10):** In the revision, we provide another interesting example with rigorous proof where the implicit bias induced by noise in the normal space cannot be characterized by a fixed regularizer, which was first discovered by [Damian et al. 2021] but was only verified via experiments.
>
> 	We do have some results for other architectures but they haven't been worked out completely. We will leave them for future works.
>
> 3. **What happens with mini-batch noise? Does the analysis collapse?** No, our analysis doesn't collapse. Indeed, when mini-batch SGD results in zero noise covariance on the manifold, e.g., mini-batch SGD achieving interpolation solutions, our result implies that the limiting diffusion degenerates, at least in the time scale of $O(1/\eta^{2})$. In other words, the iterate will stay at the landing point on the manifold, $\Phi(x(0))$, where $x(0)$ is the initialization. On the other hand, if one uses softmax loss and $\ell_2$ regularization, the noise covariance of mini-batch SGD won't vanish at the local minimizer, as the gradient of softmax is always non-zero. It's a more complicated case and is left for future work.
>
> 	Another interesting direction left as future work is to characterize the implicit regularization of SGD before reaching zero loss. In that case, the strength of the implicit regularization of noise will heavily depend on the initialization, the learning rate, and the magnitude of the noise covariance. This regime has been studied previously by [Pesme et al. 2021], but the authors approximate SGD by SDE with a slightly modified covariance, so it's still unclear how to precisely do such analysis on the original SGD and we leave this to future works.
>
> 4. **More intuition on the regularizer in the example towards sparse recovery:** Eq.(17) is the regularizer in the space of $x=\binom{u}{v}\in\mathbb{R}^{2d}$, and if we view it in the space of $w= u^{\odot 2}-v^{\odot 2}\in\mathbb{R}^{d}$ as we’ve done in Section D.4 in the appendix, then this is indeed a weighted $\ell_1$ regularizer. It has been shown by [Tropp 2015] that this type of regularizer will produce a sparse solution, i.e., killing the directions for which the sparsest predictor is zero. Compressed sensing.
>
> 5. **Lemma 6.3 is already known:** We kindly note that our result is not implied by [Pesme et al., 2021]  as we allow any non-zero initialization, while theirs require the initialization to be balanced and identity initialization, i.e., $u_i=v_i=\alpha$, $\forall 1\le i \le d$.
>
> 	Therefore, we found our proof still valuable. On the other hand, even if their proof (based on constructing a Lyapunov function) can be generalized to the general case, our proof takes a different strategy, i.e., we show the loss function satisfies the PL condition along its trajectory, which is a nice observation combining the properties of trajectory and landscape, and might be of independent interest to the community.
>
> 6. **Missing References**: Thank you for pointing us to these nice papers! We have added citations to them in the revision.
>
> **Reference**:
>
> Alex Damian, Tengyu Ma, and Jason Lee. Label noise SGD provably prefers flat global minimizers. arXiv preprint arXiv:2106.06530, 2021.
>
> Joel A Tropp. Convex recovery of a structured signal from independent random linear measurements. In Sampling Theory, a Renaissance, pp. 67-101. Springer, 2015.
>
> Pesme, Scott, Loucas Pillaud-Vivien, and Nicolas Flammarion. "Implicit Bias of SGD for Diagonal Linear Networks: a Provable Benefit of Stochasticity." arXiv preprint arXiv:2106.09524 (2021).

---

> > ### Comment · Reviewer_Msxe · 2021-11-19
> > **Thanks**
> >
> > Thank you for the answer. My score remains unchanged.

---

> > > ### Author Response · Authors · 2021-11-29
> > > **Thank you for your positive feedback!**
> > >
> > > We appreciate your support very much!

---

### Official Review · Reviewer_6sCC · 2021-10-31

**Correctness:** 4
**Technical Novelty And Significance:** 3
**Empirical Novelty And Significance:** 2
**Recommendation:** 8
**Confidence:** 3

**Main Review:**

This paper provides a very strong result for the implicit bias of SGD after it achieves zero training loss, i.e., capturing its behavior once it is in the manifold of the global minimizers. Based on Katzenberger (1991)’s result, they capture the global behavior, and their result covers the case that the stepsize $\eta\to 0$. In detail, via a projection operator, they kill the noise in the normal space and derive the limiting SDE via showing that the SGD will converge in distribution to this SDE. For each term of this SDE, they provide convincing explanation. Based on this result, they derive the limiting Flow for label noise. Furthermore, they study the overparameterized sparse linear regression model, showing that, in contrast to GD, SGD with label noise can recover the ground truth efficiently provided almost any initialization.

Overall, this paper is very solid, and its technique, though is adapted from Katzenberger (1991), shall have its own value to the community to understand the behavior and property of SGD. Its application to demonstrate the power of label noise SGD is also very interesting.

However, I have a few confusions about the label noise SGD. In the paragraph after the equation 12, the author mentioned that “Suppose the model can achieve the global minimum of the loss ...” Can you demonstrate that why the ground truth should be the global minimizer in the label noise setting? On the other hand, it seems that this analysis heavily relies on the label noise. Is it possible to extend this result to the mini-batch SGD?

**Summary Of The Paper:**

This paper provide a global analysis of the implicit bias of SGD after it achieves zero training loss by adapting ideas from Katzenberger (1991). Moreover, based on this analysis, they show that label noise SGD can avoid the kernel regime, i.e., its sample complexity is significantly less than $O(d)$ in the overparameterized sparse linear regression model.

**Summary Of The Review:**

This paper has very good theory and is well-structured. Also, its application to label noise SGD is also very interesting.

---

> ### Author Response · Authors · 2021-11-18
> **Response to the Review by Reviewer 6sCC**
>
> We first thank the reviewer for the appreciation. We address your questions below.
>
> 1. **Why the ground truth should be the global minimizer in the label noise setting?** After Eq.(12), we consider the loss $L(x) = \frac{1}{2} \mathbb{E}_{i,\delta_i}[ (f_i(x) - \tilde y_i)^2]$ where $\tilde y_i = y_i + \delta_i$, $i$ is uniformly random over $[n]$ and $\delta_i\overset{iid}{\sim}\textrm{Unif}\{-1,1\}$.
>   Note that this only differ by a constant from the regular $\ell_2$ loss, as one can verify that $ L(x) = \mathbb{E} _i$$ [ (f_i(x) - y_i)^2] +\mathbb{E} _{\delta_i} [\delta_i^2] $ by the independence between the label noise and the random index $i$. Therefore, the ground truth is still the global minimizer. From another perspective, in this setting, the label noise is equivalent to gradient noise with a certain structure. It doesn't change the expected gradient for any $x$, neither does the local/global minimizer.
>
> 2. **Is it possible to extend this result to the mini-batch SGD?** Thank you for pointing out this important direction. Indeed, when mini-batch SGD results in zero noise covariance on the manifold, e.g., mini-batch SGD achieving interpolation solutions, the limiting diffusion degenerates, at least in the time scale of $O(1/\eta^{2})$. In other words, the iterate will stay at the landing point on the manifold, $\Phi(x(0))$, where $x(0)$ is the initialization. On the other hand, if one uses softmax loss and $\ell_2$ regularization, the noise covariance of mini-batch SGD won't vanish at the local minimizer, as the gradient of softmax is always non-zero. It's a more complicated case and is left for future work.
>
> 	Another interesting direction left as future work is to characterize the implicit regularization of SGD before reaching zero loss. In that case, the strength of the implicit regularization of noise will heavily depend on the initialization, the learning rate, and the magnitude of the noise covariance. We refer the reviewer to [Pesme et al. 2021] for such analysis in a simplified setting, where the authors approximate SGD by SDE with a slightly modified covariance.
>
> **Reference**
>
> Pesme, Scott, Loucas Pillaud-Vivien, and Nicolas Flammarion. "Implicit Bias of SGD for Diagonal Linear Networks: a Provable Benefit of Stochasticity." arXiv preprint arXiv:2106.09524 (2021).

---

### Official Review · Reviewer_uj7w · 2021-11-02

**Correctness:** 4
**Technical Novelty And Significance:** 3
**Empirical Novelty And Significance:** 3
**Recommendation:** 8
**Confidence:** 3

**Main Review:**

Strength. The results are novel and the analysis is non-trivial. It introduces new ideas to the field by adapting the ideas from Katzenberger (1991). It allows the authors to obtain a global analysis of the implicit bias valid for $\eta^{-2}$ steps, as well as any arbitrary noise covariance structure.

Weakness. Some model assumptions, setups and the conditions in the main results might need more explanations.

**Summary Of The Paper:**

The authors propose an SDE approximation to study the implicit bias of SGD with infinitesimal learning rate. They show that given any noise covariance structure, after $\eta^{-2}$ steps, the SGD converges to an SGD on a certain manifold of local minimizer as the learning rate $\eta\rightarrow 0$. This recovers and strengthens some existing results in the literature. In particular, they show a sample complexity gap between label noise SGD and GD in the kernel regime for an overparametrized linear model, justifying the generalization benefit of SGD.

**Summary Of The Review:**

(1) It would be nice if the authors can explain more and justify why you can use the SDE approximation in equation (2). $\eta$ is the stepsize, and I can understand the stochastic modified equation in the literature, but somehow I don't see why SGD can be approximated by equation (2).

(2) In the drift term in equation (3), should it be $\nabla L$?

(3) In the paragraph after equation (3), you wrote that hopefully we can simplify the dynamics Equation (3) via choosing suitable $\Phi$. What I don't understand is that on the surface, it seems equation (3) is even simpler than equation (4) or (5). So what do you mean by simpler?

(4) On page 5, in the paragraph before Theorem 4.1., what is $\phi$?

(5) I find the assumptions in Theorem 4.6 quite strong. You need to assume that the SDE in equation (10) has a strong solution. Ideally, the assumption should be on the loss function $L$ instead of the equation (10). Another strong solution is that $Y$ never leaves $\Gamma$. This in my opinion is also quite strong. $Y(t)$ is an SDE with a tangent noise term, and because of the Brownian noise, I just don't see how this assumption can be satisfied. Even if you have degenerate noise in some direction, e.g. in the case of underdamped Langevin diffusion, it is still supported on the entire Euclidean space. If you have to assume such strong solutions for Theorem 4.6., it might be helpful to construct some toy examples to demonstrate that the limiting $Y$ is non-trivial and meaningful at least for some toy examples.

(6) I also find Remark 4.7. a bit puzzling. In Remark 4.7., you mentioned that the convergence in distribution result in Theorem 4.6. (at the time $T$) implies the convergence for the sample path on $[0,T]$. I don't understand this. Shouldn't it be the case that the latter implies the former?

---

> ### Author Response · Authors · 2021-11-18
> **Response to the Review by Reviewer uj7w**
>
> We thank the reviewer for the careful read. Your comments greatly help us improve the writing quality.
>
> 1. **Justification for SDE approxiamtion:** In [Li et al. 2017], the SDE approximation of SGD writes as $dX(t) = -\nabla L(X(t)) dt + \sqrt{\eta} \sigma(X(t)) dW(t)$ where the time correspondence is $t=k\eta$, i.e., $X(k\eta)\approx x_\eta(k)$ where $x_\eta(k)$ is the $k^{\text{th}}$ SGD iteration with learning rate $\eta$. Indeed, by letting $\tilde X(t) = X(t\eta)$, one can verify that $\tilde X(t)$ admits Eq.(2) in our paper, and unit continuous time correpsponds to one step for SGD. Moreover, by applying the rescaling of $\tilde X(t)=X(t/\eta)$, one can further recover Eq.(3) in our paper. See Section B.1 for a derivation.
>
> 2. **Typo:** Thanks for pointing out this typo! Yes, it should be $\nabla L$ instead, and we corrected it in the revision.
>
> 3. **Equation (4),(5) doesn't simplify (3) on the surface:** Here the goal is to understand when learning rate $\eta\to 0$, how does $x_\eta(t)$ in Eq (3) move along the manifold $\Gamma$. That is, we want to get some limiting dynamics on the manifold $\Gamma$, which is independent of $\eta$. Eq.(4) and (5) simplified Eq.(3) in this sense, as they get rid of the dependency on LR $\eta$ for their coefficients of the equation.
>
>
> 4. **Definition of $\phi$**: $\phi(x,t)$ is the solution found by gradient flow of $L$  initialized at $x$ after time $t$. See the notation section (Section 3) for a formal definition.
>
> 5. **Stong Assumptions in Thm 4.6. Ideally should only make assumptions on $L$:** We can’t only rely on assumptions of $L$. The structure of noise covariance matters, and actually the condition that (10) had a global solution made non-trivial requirements on both loss and covariance of noise. We don’t need to assume $Y(t)$ doesn’t leave $\Gamma$ and this is a typo. What we really need is that $Y(t)$ doesn’t leave $U$, which is much weaker. We've corrected this in the revision.
>
> 	Our full Theorem B.9 extends the convergence guarantee to the case where no global solution exists. In such case, Theorem B.9 says as long as the solution stays in some compact set $K\subset U$ for at least $T$ time w.p. $1-\delta$, then ignoring a small probability of $2\delta$, the trajectories still converges to the limiting diffusion under certain coupling between the probability spaces.
>
> 6. **Confusion for Remark 4.7:** Thank you for pointing out this question. In Remark 4.7, the starting words should be 'Theorem B.9' rather than 'Theorem 4.6', and we've corrected this in the revision.
>
> Again we thank the reviewer for the detailed comments. If our responses address your concerns, we kindly request the reviewer to reconsider the score.
>
> **Reference**:
>
> Qianxiao Li, Cheng Tai, and E Weinan. Stochastic modified equations and adaptive stochastic gradient algorithms. In International Conference on Machine Learning, pp. 2101-2110. PMLR, 2017.

---

> > ### Comment · Reviewer_uj7w · 2021-11-29
> > **response to authors response**
> >
> > The author(s) have carefully addressed the concerns I raised. I am satisfied with the response and have raised my score accordingly.

---

> > > ### Author Response · Authors · 2021-11-29
> > > **Thank you for your positive feedback!**
> > >
> > > We are very glad that our response has addressed your questions, and thank you for raising the score!

---

### Official Review · Reviewer_TEtd · 2021-11-03

**Correctness:** 4
**Technical Novelty And Significance:** 4
**Empirical Novelty And Significance:** Not applicable
**Recommendation:** 8
**Confidence:** 4

**Main Review:**

Pros: The analysis using $\eta^{-2}$ scaling to separate the fast and slow motion of SGD dynamics is novel, and this mathematical framework seems appropriate for analyzing the implicit regularization of SGD in the manifold of zero training loss.

Major comments:
- Previous papers, e.g. Ghorbani et al. (2019) [Sect. 4.1], Gur-Ari et al (2018), Li et al. (2020), empirically show that there is significant alignment in top eigenspace between the Hessian and gradient covariance matrix using various deep neural network architectures and image datasets. If this alignment were true, I assume that Eq. (13) could describe the implicit regularization of SGD for general deep neural network. It would be very interesting if Eq. (13) can be used to explain the empirical observation of the spectrum of Hessian around minimizer found by SGD, i.e. a large bulk of nearly 0 eigenvalues with some large outliers. It seems this can be answered by a closer look of the right-hand-side of Eq. (13) and see how the gradient of $\Phi$ interacts with the gradient of tr(Hessian).
- The definition of the manifold $\Gamma$ is a bit confusing. At first it is defined at the beginning of p.5 that looks general and is not specific to a model, but then in Theorem B.9 in the appendix, definition of $\Gamma$ in Eq. (15) specializes on overparametrized linear models. Does Theorem 4.6 only apply to settings in Sect. 6? Clarification in appropriate places may be needed.
- Given that the ground truth parameter $w^\ast$ is $\kappa$-sparse in Sect. 6, could some other types of regularization beside label noise better explore the manifold $\Gamma$ and recover $w^\ast$ with a lower sample complexity than $\kappa\log{d}$?  For example, if one is allowed to introduce a deterministic trend in the right-hand-side of Eq. (13) that can somehow lead to sparsity without leaving the manifold $\Gamma$, e.g. directional pruning (https://arxiv.org/abs/2006.09358), would it help reduce the sample complexity?

Minor comments:
- Typo: “Polyak-ojasiewicz”
- Fifth line below Eq. (6), no $n$ in $\lim_{n\rightarrow\infty}{X\left(t\right)}$, and something seems to be wrong in that sentence
- The $\dagger$ is undefined in Definition 4.4 where it firstly appears
- Cor 5.2, $\Sigma=c\nabla^2 L$, no trace
- Are $\sigma$ and $\Sigma$ the same thing in Eq. (8)? I assume the operator $\sigma$ in Sect. 4.1 is the same as the $\Sigma$ in later sections, but it is best to unify them

References:
- Ghorbani, B., Krishnan S. and Xiao, Y. (2019). An Investigation into Neural Net Optimization via Hessian Eigenvalue Density. ICML
- Guy Gur-Ari, Daniel A. Roberts and Ethan Dyer (2018). Gradient Descent Happens in a Tiny Subspace. Arxiv: 1812.04754
- Xinyan Li, Qilong Gu, Yingxue Zhou, Tiancong Chen and Arindam Banerjee (2020). Hessian based analysis of SGD for Deep Nets: Dynamics and Generalization. Arxiv: 1907.10732.




**Summary Of The Paper:**

This paper proposes a new mathematical framework for explaining implicit regularization of SGD after attaining zero training loss. Technical novelty includes a different time scaling of the SGD that leads to the Katzenberger process, and then applying weak convergence of the Katzenberger process to obtain an elegant characterization of the SGD dynamics on zero loss manifold. This new technical device leads to a new proof of the sample complexity bound on the overparametrized linear model with label noise regularization.

**Summary Of The Review:**

This paper provides a new mathematical framework for analyzing the implicit regularization of SGD after attaining zero training loss. The framework is novel, and it leads to a simpler analysis of sample complexity under noisy label regularization. I am in favor of accepting this paper.

---

> ### Author Response · Authors · 2021-11-18
> **Response to the Review by Reviewer TEtd**
>
> We first thank the reviewer for the appreciation. We address your questions one by one below.
>
> 1. **Connection to empirical finding on hessian spectrum: a large bulk of nearly 0 eigenvalues with some large outliers:**  We thank the reviewer for this thoughtful comment. Indeed, in the case of $\ell_2$ loss, every point on the manifold of global minima satisfies that the hessian at $x$, $\nabla^2 L(x)$, is at most of rank $n$, where $n$ is the number of the data points. Therefore, the number of positive eigenvalues of the hessian is already much smaller than the number of parameters in the overparametrized setting. (See the discussion at the end of Section 3.1)
>     In our example of overparametrized linear model (Section 6), we show the limiting flow can further decrease the sparsity of hessian from $n$ to $\kappa$, though this relies on the special structure of the problem. It’s left as future work to investigate in what sense similar conclusions can be achieved for realistic models and datasets.
>     On the other hand, being on the manifold of global minima already means hessian cannot have negative eigenvalues.
>
>
> 2. **Confusion on the definition of $\Gamma$:** To clarify, our definition of $\Gamma$ in Assumption 3.1 is not specific to any model and is general. When we apply our main theorem to the overparametrized linear models in Section 6, we fixed Gamma as Eq (15). Also, the manifold $\Gamma$ in Theorem B.9 is fully general and is independent of the overparametrized linear model (there was a typo in the previous statement of Theorem B.8, and we only need $\Gamma$ and $U$ to satisfy Assumption 3.1 and 3.2, but not defined as Eq (15)).
>
> 3. **Can we do better than $\kappa\ln d$ for sparse recovery?:** Raskutti et al. (2012) proved that the $\tilde{O}(\kappa \ln d)$ sample complexity is minimax optimal (information-theoretically), so we don’t think it’s possible to further improve it by applying other regularization methods.

---

### Author Response · Authors · 2021-11-18
**Summary for Revision**

In the revision, we made the following major changes (rendered in purple in .pdf) regarding the reviewers' concerns:

1. A new subsection (Section B.1 in appendix) explaining how different time scalings could affect the coefficients in SDE (2) and (3). (In response to Comment (1) by **Reviewer uj7w**);
2. A new example, *k-phase motor*, for the limiting process of Eq. (10), showing that even when the limiting process is deterministic, the resulting flow is not necessarily a potential flow. In detail, we show the flow moves in a cycle at a constant speed, thus there is no potential or regularization term which monotone decreases along with the limiting flow (In response to the request of more interpretation by **Reviewer Msxe**);
3. Clarification on the two-phase SGD dynamics with modification in the third paragraph in the introduction, the new Remark B.10, and Figure 2 in the appendix. (As suggested by **Reviewer Msxe**))
4. Unifying the notation about $\Xi$, $\sigma$ and $\Sigma$. Since we only consider SGD with finite data in this paper, instead of using $\Xi$ as an index set in the original manuscript, we use $\Xi$ to denote the number of training data and use $\sigma(x)\in\mathbb{R}^{D\times \Xi}$ to denote the noise function. The covariance matrix $\Sigma\in\mathbb{R}^{D\times D}$ is defined to be $\sigma\sigma^T$. (In response to **Reviewer TEtd**);
5. Fixing important typos in Theorem 4.6 and Remark 4.7. (In response to **Reviewer uj7w**);
6. Fixing important typos in Theorem B.8. (In response to **Reviewer TEtd**);
7. Two additional references as suggested by **Reviewer Msxe**;

Besides the above major changes, we also fixed other minor typos and polished our proofs.

---

### Comment · Area_Chair_ZTM1 · 2021-11-27
**Clarification of the proof**

In order to further access the correctness of the proof, could the authors elaborate:

(1). In Theorem 4.6, why the only stochastic part is on the tangent space? On the manifold, why would the process become completely deterministic?

(2). Why would Y stays at the manifold given in Theorem 4.1?

(3). Why is the function $\phi$ differentiable? Moreover, if the manifold is the zero-loss manifold, wouldn't $\phi(x) = x$ since gradient flow would not move?

(4). Why would the noise of on the tangent space becomes zero as in Corollary 5.2?

(5). What if a weight decay regularizer is added to the objective? Is the proposed regularizer different from a standard weight decay?

(6). Would the theorem still be valid with only the $u_i^2 $ part instead of $u_i^2 - v_i^2$? This choice looks really peculiar to me. It seems like any power > 2 would also make the claim invaliad.

**The author need to change the wording of "sparse linear regression". The model is not a linear model but a peculiar one as pointed in (6).**


(7). Possibility of extending beyond current peculiar example: I dont see any further application of this approach. In order to use the general framework, one has to show:
(a). Being able to characterize the manifold of solutions where gradient flow would converge to, moreover, does the current theorem require that this is the zero-loss manifold? If so, this even further limits the applicability of the framework.
(b). **Only applicable to label noise with L2 regression**. Otherwise I am not sure how the stochastic term in Theorem 4.1 can be removed.
(c). Being able to show that one can minimize the regularizer continuously along the zero-loss manifold. Besides the peculiar choice of $u_i^2 - v_i^2$, I dont see how this can be done in other cases.


(8). What if the learning rate $\eta$ is not infinitesimal? The author assumes that gradient descent trajectory follows tightly with the gradient flow trajectory. However, this is **known to be not true in deep learning, even with very small learning rate ** (see e.g. "Gradient descent happens at the edge of stability in deep learning"). Can the framework be used in the actual practical regime (i.e. not relying on the exact solution of Theorem 4.1, but only relying on some properties that can hold for large $\eta$)?

---

> ### Author Response · Authors · 2021-11-29
> **Response to the questions by Area Chair ZTM1 (1)**
>
> We suspect that AC seeks a high-level intuition behind the math in the paper. Our analysis applies to the manifold of local minimizers (which need not be zero loss). In this manifold, the gradient is zero for the training dataset but is in general nonzero for individual training data points. Then noise in stochastic gradient doesn't vanish.
>
> Dynamics of SGD with small LR $\eta$ has two phases, where the first one lasts for $\Theta(1/\eta)$ steps and tracks gradient flow while the second one lasts for $\Theta(1/\eta^2)$ steps and tracks the limiting diffusion on the manifold. (See Figure 2 for a demonstration) When LR $\eta$ goes to $0$, $x_\eta(t/\eta^2)$ converges to some point on the manifold of local minimizers, for any fixed $t>0$. Thus $Y(\cdot)$, the limiting process of $x_\eta(\cdot/\eta^2)$, must be completely on the manifold, as shown by [Katzenberger, 1991] \(also see our Theorem 4.1). Our approach to understand the limiting diffusion is to track the 'slow' dynamics of $x_\eta(t/\eta^2)$, which is defined as $\Phi(x_\eta(t/\eta^2))$, the projection onto the manifold using gradient flow starting from $x_\eta(t/\eta^2)$. The limiting dynamics (as intuitively sketched in Section 1.2) is a diffusion on the manifold of local minimizers. It becomes a deterministic flow when the original SGD noise is always in the normal space. This happens for instance in the case of label noise which was studied in earlier papers as well.

---

> ### Author Response · Authors · 2021-11-29
> **Response to the questions by Area Chair ZTM1 (2)**
>
> Below we give clarifications of the proofs towards questions (1) to (8).
>
> (1). **Why the only stochastic part is on the tangent space?** This is due to our Lemma 4.3, showing for any $x$ on manifold that $\partial \Phi(x)$ is the projection matrix onto the tangent space $T_x(\Gamma)$. So it kills all the normal space noise in Theorem 4.1.
>
> **On the manifold, why would the process become completely deterministic?**
> In some special cases where the tangent noise part vanishes (e.g., SGD with label noise), the limiting process would become completely deterministic. Yet, generally speaking, non-vanishing tangent noise (the first term in Eq. (10)) would induce a stochastic process on the manifold.
>
> (2). **Y stays at the manifold in Thm 4.1** is proved by [Katzenberger, 1991]. Intuitively, we can think $Y_n$ as accelerated SGD or SDE described in the introduction with vanishing LR $\eta_n$. $Y_n$ gets increasingly closer to the manifold as the LR decreases, and therefore the corresponding limiting process must be on the manifold. (See texts above Eq (4))
>
> (3). **Differentialbility of $\Phi(x)$:**
> [Falconer, 1983] shows that $\Phi(x)$ is twice continuously differentiable in any open set $U$ if for all $x\in U$, $\Phi(x)$ belongs to the manifold. (See Assumption 3.2 and Lemma B.1 in our manuscript)
>
> You're correct that **$\Phi(x)\equiv x$** on the manifold, which directly implies  $\partial\Phi(x)$ is equal to identity when restricted in tangent space.
>
> (4). **Why would the noise on the tangent space become zero as in Corollary 5.2?**
> It's a direct consequence of Lemma C.3, which says $\partial \Phi(x) \nabla^2 L(x)=0$. We further note that label noise implies the noise covariance $\Sigma = c\nabla^2 L$ for some $c>0$, and thus the tangent noise part (the first term in Eq. (10)) is zero. We will polish the proof of Corollary 5.2 and refer to Lemma C.3 explicitly. Thanks for the question!
>
> The intuition behind Lemma C.3 is that the hessian at local minimizers only has non-zero eigenvalues in the normal space.
>
> (5). **What if a weight decay regularizer is added to the objective?** Adding explicit regularization (e.g., weight decay) changes the loss landscape and could make the manifold of local minima disappear, and thus the implicit regularization effect of SGD could become vacuous, e.g., standard linear regression with $\ell_2$ regularization admits unique local minimizer and thus SGD with all types of noise and small LR converges to that. On the other hand, adding weight decay regularizer on weights before normalization layers do not form any explicit regularization, which allows limiting diffusion/flow on the manifold. In summary, there is no simple and universal answer to this question and we need to analyze it on a case-by-case basis.
>
> **Is the proposed regularizer different from a standard weight decay?** In this manuscript we didn't propose any regularizer for practical usage. For the trace of hessian regularizer induced by label noise, in general, it is not equivalent to weight decay, though they are equivalent in our example -- overparametrized linear models with boolean data. To be more specific, for overparametrized linear models with boolean data, the implicit bias of label noise SGD with infinitesimal LR is equivalent to gradient flow with $\ell_2$ regularizer of infinitesimal strength.
>
> (6). **Would the theorem still be valid with only the $u_i^2$ part instead of $u_i^2-v_i^2$ ?**
>
> The sparse recovery result will still be valid for using $u_i^2$ part only, as long as the groundtruth is element-wise positive. The corresponding implicit regularizer of label noise SGD would be the squared (weighted) $\ell_2$ norm. The convergence proof of gradient flow will be slightly different because it doesn't have the conservation of $u_i(t)v_i(t)$ along gradient flow trajectory and we have to take into account the positiveness of the groundtruth.

---

> ### Author Response · Authors · 2021-11-29
> **Response to the questions by Area Chair ZTM1 (3)**
>
> ((6) continued)
>
> **''$u_i^2-v_i^2$ parametrization looks peculiar'':** The main reason that we chose $u_i^2-v_i^2$ is to simplify the theorem statement by removing the expressiveness constraint, as mentioned above. The $u_i^2-v_i^2$ model is quite standard and has been extensively studied by the community, e.g. [Woodworth et al., 2020, Pesme et al., 2021, Azulay et al., 2021, Li et al., 2021].
>
> We also note that the $u_i^2-v_i^2$ parametrization is actually equivalent to $2a_ib_i$ parametrization, which might seem more natural.  This is because $u_i^2-v_i^2 = 2 \frac{u_i-v_i}{\sqrt{2}}\frac{u_i+v_i}{\sqrt{2}}$ and that the trajectory of Gradient Descent is invariant under orthogonal reparametrization, which comes from the fact that for any orthogonal matrix $U$, function $f$ and any vector $x$, it  always holds that $\nabla_{Ux} f(Ux) = U\nabla_x f(Ux)$. So we can simply set $a_i(0) =\frac{u_i(0)-v_i(0)}{\sqrt{2}} $ and $b_i(0)= \frac{u_i(0)+v_i(0)}{\sqrt{2}}$, replace $u_i^2-v_i^2$ by $2a_ib_i$ and do SGD/GD on $\{a_i,b_i\}$ instead. A straightforward induction shows that $a_i(t) =\frac{u_i(t)-v_i(t)}{\sqrt{2}} $ and $b_i(t)= \frac{u_i(t)+v_i(t)}{\sqrt{2}}$ holds for any $t>0$ and any LR, which implies $2a_i(t)b_i(t)=u_i^2(t)-v_i^2(t)$ for every $t>0$.
>
>
>
> **Will our analyses be valid when power>3?** Yes, our analysis can be extended to power>3 with the almost same proof. Of course, the induced regularizer by label noise changes with the architecture, and in this case, it becomes $\ell_{4-4/p}$ norm regularizer (weighted by data), where $p\ge 1$ is the power and the current regularizer is $\ell_2$ on $u_i,v_i$ space. (Note when $p=1$ the regularizer becomes constant, meaning the implicit bias of label noise SGD vanishes for standard linear regression) Since power>3 has a different implicit regularizer, we don't expect them to enjoy the same guarantee for sparse recovery under the same setting in Section 6, especially for large $p$.
>
> **'' The author needs to change the wording of "sparse linear regression'':** We kindly note that we didn't use the term **sparse linear regression** in the current manuscript. We fully agree that our specific parametrization of the model is non-linear, though the problem itself is a linear regression with a sparse groundtruth.
>
>
> (7). **Possibility of extending beyond current peculiar example**:
>
> **Does the current theorem require that this is the zero-loss manifold?** No. The theorem only requires it to be a manifold of local minimizers.
>
> **The approach is only applicable to label noise with L2 regression:** First, our approach could be easily applied to other losses, e.g. $\ell_p$ loss, cross-entropy loss, as long as Assumption 3.1 and 3.2 are satisfied.  [[Demian et al., 2021]](https://arxiv.org/pdf/2106.06530.pdf) discussed the application of label noise on losses beyond $\ell_2$ loss in their Section D. Second, our approach can also be applied to other noise, e.g. the isotropic noise (Corollary 5.1). The $k$-phase motor example in section C.4 is another example that gives interesting implicit bias without using label noise. We may need more advanced math tools to understand the limit diffusion on manifold when it is not a deterministic flow. However, the punchline is that, if certain implicit bias is experimentally verified to hold for arbitrarily small LR $\eta$ in $\Theta(1/\eta^2)$ steps, then the limiting diffusion will be the exact and right tool to understand the implicit bias. Our framework will be a powerful tool to extract conjectures with clean math formulation from experiments.
>
> **Besides $u_i^2-v_i^2$, where can the approach be applied?** As mentioned above, our approach easily generalizes to $u_i^p-v_i^p$ or $u_i^p$. It can also be applied to general linear networks, including matrix factorization problems, like $UU^\top$, where $U\in \mathbb{R}^{d\times m}, m\ge d$. The loss function can have form of $L(U) = \sum_{i=1}^n (\textrm{Tr}[X_i UU^\top] -y_i)^2$, where data $X_i\in \mathbb{R}^{d\times d}$. The implicit regularizer for label noise would be $R(U) = \textrm{Tr}[\sum_{i=1}^n (X_i+X_i^\top)^2, UU^\top]$, which can be viewed as a weighted nuclear norm regularizer. Particularly, for matrix completion, when $X_i = e_{j_i}e_{k_i}^\top$, where $e_{j_i}, e_{k_i}$ are one-hot $d$-dimensional vectors with $1$ at $j_i$ and $k_i$. Then $(X_i+X_i^\top)^2 = e_{j_i}e_{j_i}^\top + e_{k_i}e_{k_i}^\top$. Thus with $\omega(d)$ samples, $\sum_{i=1}^n (X_i+X_i^\top)^2$ will concerntrate around $\frac{2nI}{d}$, assuming $j_i$ and $k_i$ are uniformly distributed, which implies label noise SGD will approximately minimize the nuclear norm.
>
> We agree our approach cannot be applied to ReLU networks as the smoothness assumptions are not satisfied, but besides that, we don't see particular barriers for applying it to networks with smooth activations.

---

> ### Author Response · Authors · 2021-11-29
> **Response to the questions by Area Chair ZTM1 (4)**
>
> (8). **Does our result still hold if the learning rate $\eta$ is not infinitesimal?** Our current approach is based on the analysis of [Katzenberger, 1991] which only ensures asymptotic convergence. However, given a desired accuracy/precision, as noted in our Theorem 6.1, there is a sufficiently small LR satisfying the guarantee, which is much more realistic than those results directly analyzing gradient flow or taking SDE approximation. Though it's not clear the dependence of the LR threshold on the dimension and other Lipschitz constants, our best guess is that the dependence is polynomial and a non-asymptotic rate can likely be achieved by a more fine-grained version of analysis in [Demian et al., 21], at least for label noise SGD. We will leave it for future work.
>
> **Does our result contradict with 'Edge of stability' paper?** The 'edge of stability' paper [Cohen et al., 20] actually focuses on GD and their empirical finding that sharpness will increase until $2/\eta$ doesn't apply to SGD. It's very likely the noise will prevent the sharpness from further increasing when SGD gets close to the manifold. In other words, their empirical findings don't directly contradict our theory, because we didn't really require SGD to follow gradient flow until convergence, rather our theory only asserts it will track gradient flow until the distance to the manifold is $\Theta(\sqrt{\eta})$, and since then the noise kicks in, which prevents further convergence.
>
> **Can the framework be used in the actual practical regime?** Such implicit bias of minimizing trace of hessian exists for practical settings. [[Demian et al., 2021]](https://arxiv.org/pdf/2106.06530.pdf) experimentally verified such implicit bias at constant scale LR with ResNet18 on CIFAR10 in their Figure 4. Still, we think it is unlikely to get a non-vacuous error bound if we plug in all the lipschitzness constant from realistic settings into the bound, even with an improved non-asymptotic analysis. The better way to understand our results is that it gives a clear description of the implicit bias in the limiting case.
>
>
>
> **Reference**
>
> Gary Shon Katzenberger. Solutions of a stochastic differential equation forced onto a manifold by a
> large drift. The Annals of Probability, pp. 1587–1628, 1991.
>
> K. J. Falconer. Differentiation of the limit mapping in a dynamical system. Journal of the London
> Mathematical Society, s2-27(2):356372, 1983. ISSN 0024-6107. doi: 10.1112/jlms/s2-27.2.356.
>
> Blake Woodworth, Suriya Gunasekar, Jason D Lee, Edward Moroshko, Pedro Savarese, Itay Golan,
> Daniel Soudry, and Nathan Srebro. Kernel and rich regimes in overparametrized models. In
> Conference on Learning Theory, pp. 3635–3673. PMLR, 2020.
>
> Scott Pesme, Loucas Pillaud-Vivien, and Nicolas Flammarion. Implicit bias of sgd for diagonal linear
> networks: a provable benefit of stochasticity. Advances in Neural Information Processing Systems 34 (2021).
>
> Shahar Azulay, Edward Moroshko, Mor Shpigel Nacson, Blake Woodworth, Nathan Srebro, Amir
> Globerson, and Daniel Soudry. On the implicit bias of initialization shape: Beyond infinitesimal
> mirror descent. Proceedings of the 38 th International Conference on Machine
> Learning, PMLR 139, 2021.
>
> Jiangyuan Li, Thanh Nguyen, Chinmay Hegde, and Ka Wai Wong. Implicit Sparse Regularization: The Impact of Depth and Early Stopping. Advances in Neural Information Processing Systems 34 (2021).
>
> Alex Damian, Tengyu Ma, and Jason Lee. Label noise sgd provably prefers flat global minimizers. Advances in Neural Information Processing Systems 34 (2021).
>
> Jeremy Cohen, Simran Kaur, Yuanzhi Li, J. Zico Kolter, and Ameet Talwalkar. "Gradient Descent on Neural Networks Typically Occurs at the Edge of Stability." In International Conference on Learning Representations. 2020.

---

> ### Comment · Area_Chair_ZTM1 · 2021-11-29
> **Thanks for the clarification**
>
> Thanks a lot for the classification.
>
> I am not sure about how can the current approach extend to cross entropy loss, since the minimizer is actually at infinite. Could the authors elaborate more on this point?
>
>
> Moreover, why would the regularizer derived in this paper for label noise case differs from the trace of log of negative hessian one from [Damian et al 2021]? According to you paper, the regularizer is sum of the norm of stochastic gradient square using label noise.

---

> > ### Author Response · Authors · 2021-11-29
> > **Response to the questions by Area Chair ZTM1**
> >
> >
> > **How does the current approach extend to cross-entropy loss when the minimizer doesn't exist?**
> >
> > You're correct that when the global minimum cannot be attained, our theorem will be vacuous, as there is no such manifold of minimizers. However, we kindly note that **this is not an inherent issue of cross-entropy loss**. For example, the global minimum of cross-entropy loss can be attained with soft labels. It also holds for hard labels which are not separable, e.g., [[Demian et al., 21]](https://arxiv.org/pdf/2106.06530.pdf) demonstrated the implicit regularization when the binary label of data is randomly flipped with some probability $p$ in Section D of their paper.
> >
> > **Is our regularizer different from that in [[Demian et al., 21]](https://arxiv.org/pdf/2106.06530.pdf)?**
> >
> > No, they are the same. We kindly note that in the beginning of the last paragraph on page 4 of [[Demian et al., 21]](https://arxiv.org/pdf/2106.06530.pdf), they wrote "In the limit as $\eta\to 0$, $R(\theta)\to \frac{1}{4} \textrm{tr}\nabla^2 L (\theta)$, ...". So their regularizer indeed agrees with our trace of hessian regularizer for infinitesimal LR (See Corollary 5.2).
> >
> > You're correct that the sum of the squared norm of stochastic gradient using label noise is equal to the trace of hessian (after some scaling) for any $x$ on the manifold of global minimizers.

---

### Decision · Program_Chairs · 2022-01-20

**Decision:**

Accept (Spotlight)

**Comment:**

All the reviewers agree that this paper made a solid contribution of understanding the algorithmic regularization of SGD noise  (in particular the label noise for regression) after reaching zero loss. The framework is novel and has the potential to extend to other settings.